# Active disambiguation guides inferring controllability and cause in social interactions

Lisa Spiering [1] ✉, Hailey A. Trier[1], Jill X. O'Reilly[1], Nils Kolling [2,3], Marco K. Wittmann [4,5], Matthew F. S. Rushworth [1,7] & Jacqueline Scholl [3,6,7]

There has been considerable interest in how we ascertain whether an environment is controllable and the neural mechanisms mediating this process. An especially acute version of this problem occurs when multiple people work together. Using a new task, fMRI and computational modelling, we demonstrate that in such ambiguous social contexts, people engage in specific behaviours that we refer to as active disambiguation. This process helps individuals establish what they themselves, as opposed to others, control and what consequence they themselves cause or that another person causes. People identify when active disambiguation is needed and engage in it at that time. A pattern of activity in the supramarginal gyrus that emerges during and after active disambiguation is linked to tracking uncertainty and establishing controllability. We show that activity in this brain region signals a second learning mechanism, by which individuals attribute outcomes to themselves versus others, proportional to their perceived control.

Humans are inherently social and collaborate towards shared goals. Achievements such as the moon landing were only made possible by people working together. To foster effective collaboration, it is therefore important to establish how much each individual contributes. In other words, it is essential to identify the causes behind shared outcomes. For instance, when a student receives mediocre feedback on a research project, she must determine the reason for it in order to improve. Whenever there are multiple antecedents to an event, credit assignment is needed to determine which factors are causally important and to what extent. It is especially important to determine the extent to which you, as an individual, as opposed to other people, caused a particular outcome to occur. For example, if the student took on a greater share of the project workload than her collaborators, then the outcome reveals something important about her ability. However, the outcome reveals little about her ability if she had only limited control over the project's conduct. Therefore, credit

assignment also requires assessing the level of responsibility, or control each agent has. Previous studies have examined how people attribute outcomes to one choice or another[1–4]. However, it remains an open question how people attribute ambiguous feedback to themselves or others under varying degrees of controllability, and how they infer their control over outcomes. Understanding how we ascertain control levels is important not just because it is a fundamental prerequisite for learning in humans and other animals but because alterations in perception of control have been identified in psychological illnesses such as depression[5,6] and schizophrenia[7,8].

It is clear that establishing the causes of outcomes is difficult; an outcome may be assigned not just to the action that led to it, but also to one's overall skill[9–11], or to other choices[2,4,12–14] and stimuli[1,3,15,16] that occurred close in time. Moreover, the problem is acute in social contexts in which multiple agents interact; in such situations people frequently confuse their own performance with that of others they

[1]Oxford University Centre for Integrative Neuroimaging, Department of Experimental Psychology, University of Oxford, Oxford, UK. [2]Université Lyon 1, Inserm, Stem Cell and Brain Research Institute U1208, Bron, France. [3]Le Vinatier Psychiatrie Universitaire Lyon Métropole, Bron, France. [4]Department of Experimental Psychology, University College London, London, UK. [5]Max Planck UCL Centre for Computational Psychiatry and Ageing Research, University College London, London, UK. [6]Centre de Recherche en Neurosciences de Lyon CRNL U1028 UMR5292, Université Claude Bernard Lyon 1, CNRS, INSERM, Bron, France. [7]These authors contributed equally: Matthew F. S. Rushworth, Jacqueline Scholl. ✉e-mail: lisa.spiering@psy.ox.ac.uk

cooperate with or compete against[17,18]. Theories of causal reasoning posit that causality can be established by actively intervening in the environment and observing the resulting changes[19,20]. It might initially seem difficult to imagine how such a principle might operate in social contexts but one possibility is that people effectively remove their own contribution by performing badly on purpose to determine what, if any, are the consequences. This might appear self-defeating in typical goal-directed behaviour, and indeed it would be were such an approach employed all the time. But, employed strategically at the right time, it would allow people to disambiguate the level of control they exert over events from the level of control exerted by others. To return to the example of the student with the mediocre feedback, she can ascertain her control by observing how the feedback changes when she is an active member of the research team or when she does not bother to turn up.

Controllability is closely linked to constructs such as sense of agency[21–24] and perceived self-efficacy[25,26]. Here, we examine interpersonal control, that is how people come to understand their level of responsibility over outcomes in social interactions. A key aspect of establishing control is establishing whether outcomes are contingent on our actions over and above any contingency that already exists between outcomes and states of the environment[27,28]. Analogously, establishing control in social situations requires the establishment of each individual's degree of agency relative to one another. The supramarginal gyrus (SMG) in the inferior parietal lobe has been linked to sense of agency. Apraxic patients with lesions in the left SMG confuse whether a hand movement is their own or someone else's[29] and SMG activity has been linked to realizing whether an action was one's own or not[22,23]. More generally, the SMG has been linked to directing attention to movement[30–32]. The SMG is, therefore, a candidate brain area for interpreting the consequences of any active intervention strategy that a person might employ to disambiguate their own or another's level of control over a situation.

This study investigates how people attribute ambiguous feedback to their own and another's performance, and their control over the feedback. For this, we present a new task on social credit assignment in combination with behavioural analyses, computational modelling of behaviour, and functional magnetic resonance imaging (fMRI). In this task, participants played games together with fictive other players. They had to infer from ambiguous feedback how well they did in the games, how well the other did and how much control they exerted over feedback. First, we test how participants assign feedback to themselves and others under varying levels of control. We show that participants adjust their beliefs about themselves and others by assigning prediction errors in relation to their control. Second, we examined whether participants engage in active exploratory interventions to help them disambiguate their control level from that of others. We found that participants employ certain behavioural patterns during the games which we term "active disambiguation" (AD). AD was driven by uncertainty, and in turn reduced uncertainty and led to better credit assignment. Thirdly, we established that when controllability was uncertain, AD mediated inference about control. Finally, we established the neural correlates of these behavioural patterns using fMRI and identified a key role for SMG.

## Results
### Experimental structure

31 participants completed one fMRI session, and 36 participants completed one online session. In subsequent analyses we refer to these as the fMRI and online samples. Participants were asked to assign a single, overall outcome (feedback) to three hidden latent causes: their own performance, another fictive player's performance, and their own control over the feedback (Fig. 1a). Here, performance refers to the ability or how well the participant and the other player do. In contrast, control refers to each agent's influence or responsibility, i.e., how

much each agent's performance matters for the feedback. The feedback varied parametrically over trials and was the sum of the participant's own and the other player's performance, weighted by their relative control levels over the outcome. To obtain estimates of each of the three beliefs, participants rated themselves, the other player, and their control at the start of each trial (Fig. 1b left). Then, they played one round of a simple game (Fig. 1b middle, see Supplementary Fig. 1 for illustrations of all games). Participants were told that another player would play the same game at the same time hidden from the participants' view. The other player was described to participants as computer-generated but based on real people's previous performances in the game. We designed the games so that participants could not estimate their own performance from introspection with complete accuracy and instead had to rely on the outcome feedback they were shown (see Fig. 1b and Methods). To incentivise participants to assign credit as accurately as possible, they received a bonus payment for rating accuracy.

As the credit assignment problem consisted of attributing a unitary outcome to three hidden latent causes (own/other performance and control), we systematically varied the information that participants were equipped with at different times in the block. This allowed us to precisely capture how they inferred different component aspects of the credit assignment problem. Participants completed four blocks, each with the same structure, totalling 144 trials. Each block was divided into distinct phases in which different latent causes had to be inferred (Fig. 1c, d): In the first phase, they started by playing a new game together with a new other player ('Self-Other phase', 16 trials, Fig. 1c). Here, their true level of control was shown to them so that they only had to learn about performance levels for themselves and the other player. In the second phase, they played a few trials alone so that the feedback only reflected their own performance ('Self-only trials', 4 trials). Therefore, during this phase, they could confirm and if appropriate correct their beliefs about their own performance in the game. In the third phase, they were paired with a new other player, and told that their control had been reset and was unknown ('Control-Other phase', 16 trials, Fig. 1c). This means that while knowing how good they themselves were at the game, they had to infer their control and the performance of the new other person. In summary, in the first phase, participants knew their control level but had to infer their own and the other player's performance levels, while in the third phase, participants knew their own performance level but had to infer their control level and the other player's performance level. The brief second phase served as a manipulation to ensure the efficacy of the third phase and so all the key analyses were conducted on behavioural and neural data from the first and third task phases. Note that control was a single parameter reflecting the balance between participants' own and the other's influence over outcomes (other control=1-self control). In the following, when we refer to control, we refer to the level of control the participant exerts (self control). For example, a level of control of 20% means that the participant's performance determines 20% of the feedback, while the remaining 80% of the feedback is determined by the other player's performance.

We used a pre-determined schedule of the true values of self, other and control that had to be learnt across the task (Fig. 1d). Each task block corresponded to the participants playing one game type and the games switched between each successive block. On each trial the participants' objective performances in a game were transformed with a logistic function into a self-performance score (Supplementary Fig. 2). The latter modification was crucial, because it meant that self-performance was primarily based on the pre-determined schedule, and approximately similar across trials and participants. The only exception was if participants performed so badly that their performance was counted as zero. This becomes crucial in the definition and deployment of AD trials below.

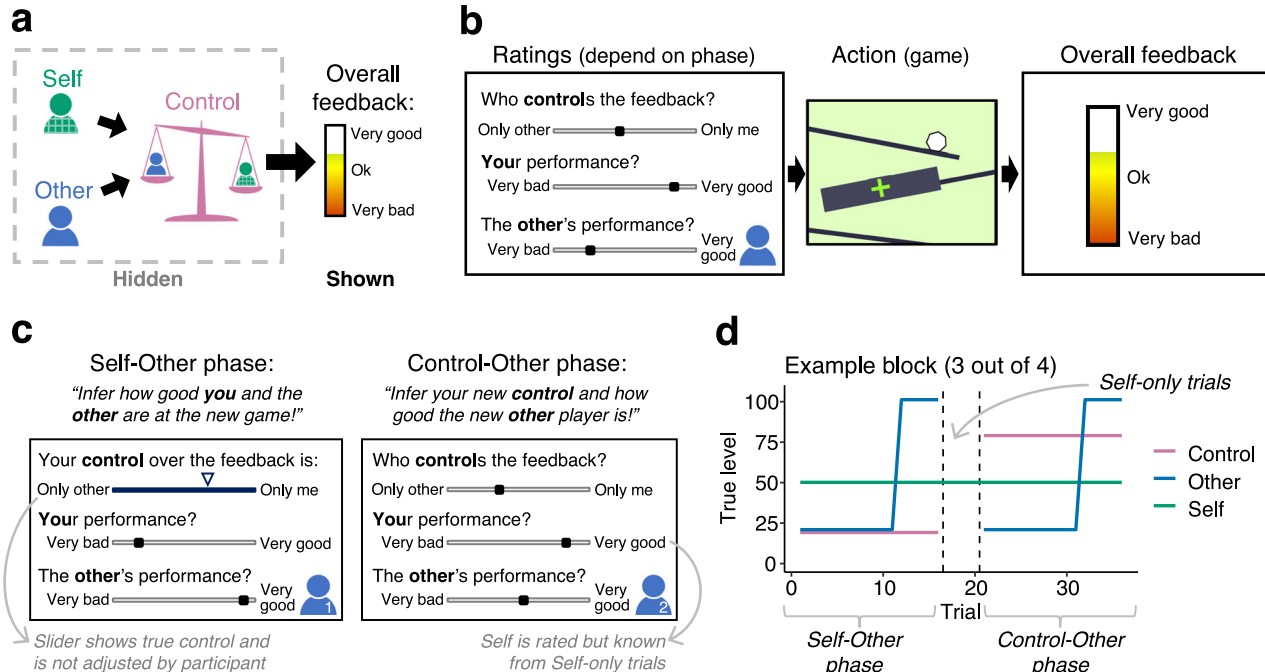

**Fig. 1 | Structure of the credit assignment task. a** In our task, participants assigned outcomes (overall feedback) to three causes: their own performance, another fictive player's performance, and their respective control over the feedback. These causes were hidden and latent, and there was only one piece of feedback on each trial. The feedback parametrically varied over trials. **b** Each trial started with a rating screen on which participants rated their beliefs about their own control, their own performance, and the other player's performance. Then, they played one round of a simple game (action phase): In the example here, a ball rolled down a slide and they had to press a button when they believed it had passed the green target cross. A large black box hid the ball's path which made it impossible to know exactly when the ball passed the cross. This meant that participants had to rely on the feedback shown to them to infer their own performance, alongside the other player's performance and their own control. At the end of the trial, the overall feedback was shown which was the sum of the participants' own and the other player's performance, weighted by their respective control over the feedback. After participants pressed a button to continue, an ITI followed. **c** Which ratings participants completed depended on the phase: In the beginning of each task block, participants entered the 'Self-Other phase' (left). This task phase allowed us to determine how participants assign credit to themselves and a new other player while knowing their control. Note that in the task, real faces were shown for the other players on each rating screen (here substituted with icons). In the Control-Other phase (right), participants inferred the new other player's performance and their new control over the feedback. In this phase, participants knew their own true performance level because of the preceding Self-only trials they had played between Self-Other and Control-Other phase. **d** Participants completed four blocks, each of which was divided into three phases, signalled to participants (example block is shown here). Each block started with the Self-Other phase, after which followed Self-only trials, ending with the Control-Other phase.

## Estimates of controllability modulate social prediction errors

A common finding across many social and non-social learning studies is that people change their beliefs based on prediction errors (PEs, differences between actual outcomes and the outcomes that had been expected)[17,33–37]. We hypothesized that in our task, participants also learn via a "total PE" (tPE) between the observed feedback and their feedback expectation (Fig. 2a). However, crucially, this tPE should then be used to assign credit to the appropriate latent causes, and to derive PEs relating to either themselves (and concerning their own performance level), the other player (and concerning the other's performance level), or the control they have over the task.

We estimated participants' feedback expectations ('estimated feedback', see Methods) from the ratings that participants provided about their self, other and control beliefs at the start of each trial. We used a Bayesian observer model ("Active learner") to extract trial-by-trial estimates of the optimal beliefs about self-performance, other performance, and control, and the uncertainty around these beliefs. The simulated data from this computational model allowed us 1) to assess whether participants' credit assignment behaviour was optimal, and 2) to derive trial-wise uncertainty estimates for participants' beliefs about self-performance, other-performance, and control.

We indeed found that participants' estimated feedback tracked the feedback shown (Fig. 2b). In contrast, their ratings of self-performance, other-performance, and control followed the hidden true levels less closely (Fig. 2c). While the accuracies of ratings and estimated feedback improved over time (ANOVA on rating accuracies with sample [fMRI or online] as between- and Time in phase [start or end] as within-participant factor; Time in phase effect: estimated feedback, $F(1,65) = 150.67$, $p < 0.001$, $\eta^2 = 0.54$; control, $F(1,65) = 38.37$, $p < 0.001$, $\eta^2 = 0.25$; other, $F(1,65) = 18.55$, $p < 0.001$, $\eta^2 = 0.16$; self, $F(1,65) = 11.04$, $p = 0.001$, $\eta^2 = 0.06$; Fig. 2d), the accuracies of estimated feedback improved more from the start to the end of a phase than the ratings did (ANOVA on rating accuracy change with rating type [estimated feedback or averaged individual ratings] as within- and sample with between-participant factor; rating type effect: $F(1,65) = 32.39$, $p < 0.001$, $\eta^2 = 0.08$; Fig. 2d). This suggests that while participants could track the overall feedback, they inferred the latent causes less well. This is because the observed feedback could be attributed to a multitude of possible combinations of latent causes. In other words, different combinations of self, other and control can mathematically give the same overall estimated feedback. That is if one holds a wrong belief about self, there can be a corresponding wrong belief about other that together would make the observed feedback possible.

Next, we tested whether people assigned credit appropriately. A key feature of credit assignment in our task was that in the first, Self-Other phase, participants knew their control level (they were directly instructed) and so tPEs should be assigned to self vs. other in relation to how much control they had over the outcome (see red text in Fig. 2a). Again, we found this to be true: overall, participants updated their beliefs about self and other more positively for positive tPEs (tPE effect on rating updates: self, $F(1,65) = 419.09$, $p < 0.001$, $\eta^2 = 0.87$; other, $F(1,65) = 310.16$, $p < 0.001$, $\eta^2 = 0.83$; Fig. 2e), meaning they rated

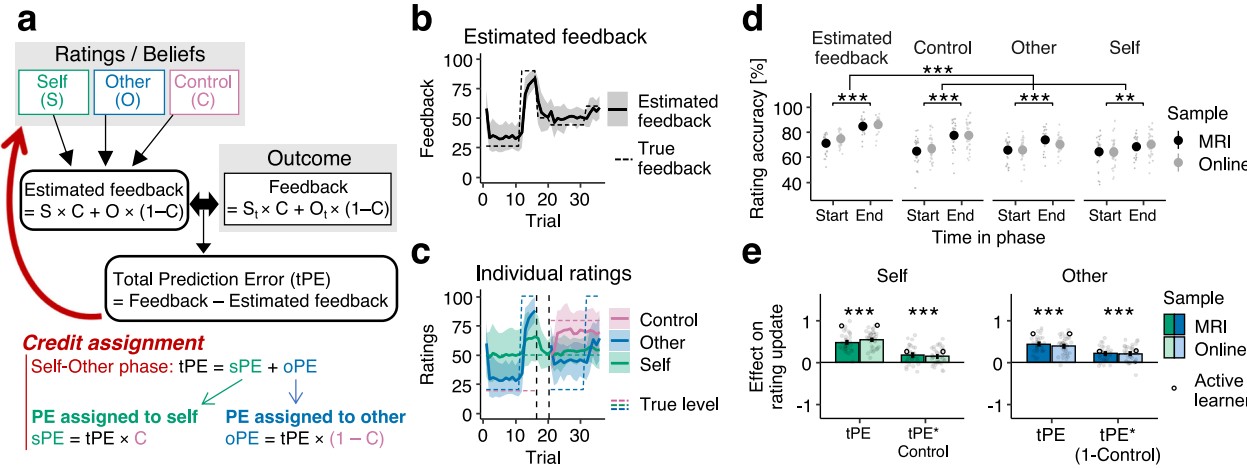

**Fig. 2 | Credit assignment behaviour. a** Hypothesized cognitive process and behaviour. Participants' beliefs (measured as their ratings) together form an estimate of the feedback they expect to receive (estimated feedback). At the time of the outcome, the discrepancy between their estimated feedback and the actual feedback results in a total prediction error (tPE). In the Self-Other phase, they use their knowledge about their control to split up the tPE into a PE assigned to themselves (sPE) and the other (oPE). **b** Participants' estimated feedback followed the observed mean feedback throughout the task (example block 3 out of 4, mean ± SD plotted as bold lines with shaded intervals). **c** Participants' mean ratings (bold lines) and their SD (shaded interval) for the same example block as in b. Supplementary Fig. 3 shows all blocks. **d** Over time, participants improved the accuracy, both of the estimated feedback and the individual component ratings of self-performance, other performance, and control, from observing the feedback (ANOVA: estimated feedback, $F_{(1,65)} = 150.67$, $p < 0.001$, $\eta^2 = 0.54$; control, $F_{(1,65)} = 38.37$, $p < 0.001$, $\eta^2 = 0.25$; other, $F_{(1,65)} = 18.55$, $p < 0.001$, $\eta^2 = 0.16$; self, $F_{(1,65)} = 11.04$, $p = 0.001$, $\eta^2 = 0.06$). Overall, the estimated feedback improved more from the start of a phase to the end than did the individual component ratings (ANOVA: rating type, $F_{(1,65)} = 32.39$, $p < 0.001$, $\eta^2 = 0.08$). Mean ± s.e.m. accuracy, dots show participants. Accuracy is the inverse of the absolute rating errors, normalised by the highest error possible. **e** Participants updated their beliefs about self and other based on the previous trial's tPE (ANOVA, tPE: self, $F_{(1,65)} = 419.09$, $p < 0.001$, $\eta^2 = 0.87$; other, $F_{(1,65)} = 310.16$, $p < 0.001$, $\eta^2 = 0.83$). As expected, if participants assign credit appropriately, they attribute more of the tPE to themselves (and less to the other) if they themselves have more control (ANOVA, tPE*Control: self, $F_{(1,65)} = 42.56$, $p < 0.001$, $\eta^2 = 0.40$; tPE*(1-Control): other, $F_{(1,65)} = 86.44$, $p < 0.001$, $\eta^2 = 0.57$). Simulated data from a Bayesian observer model ("Active learner") show the same result pattern as participants. We used ANOVAs (sample, [fMRI/online] as between-participant factor) and evaluated the $p$-value of the intercept term to assess the significance of participants' beta estimates (standardized regression coefficients) irrespective of sample. Bars show mean beta estimates, error bars indicate s.e.m., grey dots show individual participants. $n = 31$ MRI, $n = 36$ online; *$p < 0.05$; **$p < 0.01$; ***$p < 0.001$.

the performances higher if the overall feedback was better than expected. However, crucially, this effect was modulated by their level of control; they also assigned more of the tPE to themselves or the other player depending on who had more control (tPE*Control: self, $F_{(1,65)} = 42.56$, $p < 0.001$, $\eta^2 = 0.40$; tPE*(1-Control): other, $F_{(1,65)} = 86.44$, $p < 0.001$, $\eta^2 = 0.57$). We found consistent effects in simulated data from the Active learning model (Self: tPE, $F_{(1,65)} = 2203.97$, $p < 0.001$, $\eta^2 = 0.97$; tPE*Control, $F_{(1,65)} = 150.66$, $p < 0.001$, $\eta^2 = 0.70$; Other: tPE, $F_{(1,65)} = 1872.74$, $p < 0.001$, $\eta^2 = 0.97$; tPE*(1-Control), $F_{(1,65)} = 151.40$, $p < 0.001$, $\eta^2 = 0.70$). This suggests that participants, consistent with an Active learning model, attribute social PEs in accordance with their controllability.

Taken together, participants tracked the feedback they observed in the task but attributed it to the correct causes less well. To assign social PEs, they adjusted their beliefs about themselves and others in relation to their controllability beliefs. This behaviour was similar to that of the Active learning model.

## Active disambiguation as a behavioural strategy

In the task, the feedback received on each trial was ambiguous because it could be attributed to three potential causes: one's own performance, the other player's performance, and one's own control. This made the credit assignment task inherently difficult to solve. However, we hypothesized that participants employed a behavioural strategy to help with this, which we call "active disambiguation" (AD). We hypothesized that AD is a form of purposeful intervention[19,20] whereby the person aims to remove their own contribution from the outcome. In our task, this is possible via something that appears drastic and self-sabotaging in the context of standard goal-directed behaviour: to play the game (very) badly on purpose (e.g., pressing the button too late, Fig. 3a, see Supplementary Fig. 4 and "Methods" for more details on AD

detection). However, this allowed them to observe how much is due to internal causes (themselves) and how much is due to external causes (others), thereby simplifying the credit assignment task. If this reasoning is correct, then we should observe some critical features of AD trials. We hypothesized that participants make AD trials when the feedback is uncertain (when it is unclear how it reflects one or other player's performance levels, or both performance levels, and/or the control level), and that this helps them reduce uncertainty and their rating errors (Fig. 3b).

As hypothesised, we found evidence of AD; participants sometimes made errors in the games so extreme and obvious that their objective performance was transformed into a self-performance score of zero on that trial (see Methods and Supplementary Fig. 4 for details). For example, in the rockslide game, this meant responding while the ball was still visible, either before or after it entered the black box and was close to the target cross. On those trials, participants' objective game performance was transformed into a self-performance of zero, meaning they did not contribute to the feedback. Participants' behaviour exhibited high levels of AD under high uncertainty when feedback was caused by multiple sources, and AD was therefore informative (Self-Other phase: MRI, mean $(M) = 25.45\%$; online, $M = 20.16\%$; Control-Other phase: MRI, $M = 29.33\%$; online, $M = 25.40\%$; yellow bars in Fig. 3c). Whenever the feedback was not ambiguous and only reflected one's own performance, the proportion of AD showed a striking drop (Self-only trials: MRI, $M = 3.63\%$; online, $M = 2.42\%$; blue bars in Fig. 3c). We found that participants who objectively generated more AD trials also reported having done so after the experiment (MRI, Spearman's $\rho = 0.81$, $p < 0.001$, two-sided, 95%-CI = (0.63, 0.91); online, Spearman's $\rho = 0.85$, $p < 0.001$, two-sided, 95%-CI = (0.76, 0.90); Fig. 3d), further suggesting that participants used AD trials in a deliberate way and that they were not unintentional slips.

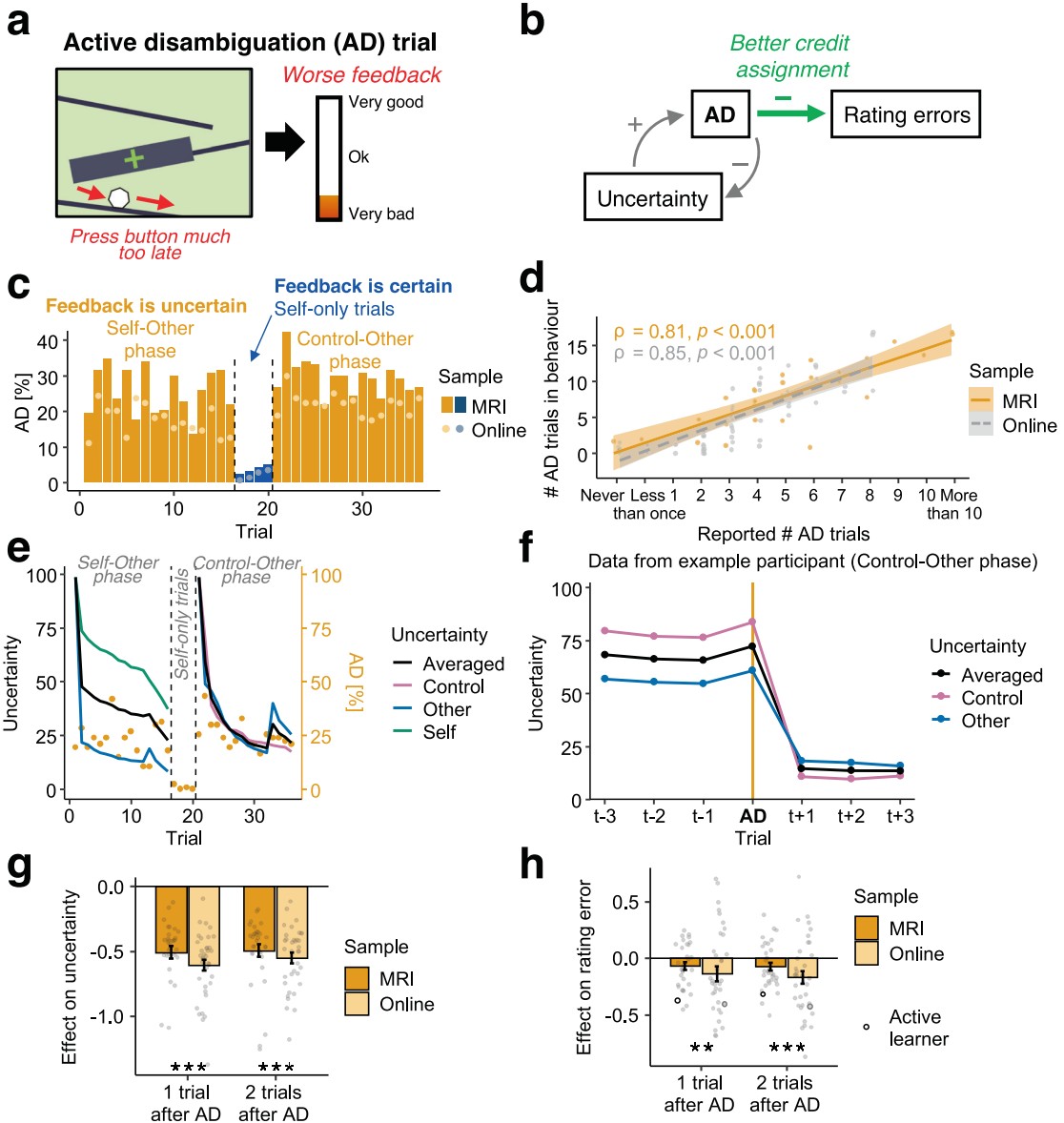

**Fig. 3 | Active disambiguation (AD). a** We hypothesized that participants made deliberate mistakes in the games to assign credit better. In the game shown here, they could press the button too late to see the difference this makes to the feedback. Whenever participants made such extreme errors, these were transformed into a zero self-performance and detected as AD trials (see also Methods and Supplementary Fig. 4). **b** We hypothesized that participants generated more AD trials under high uncertainty, and that AD trials helped them reduce uncertainty and assign credit better. **c** Participants indeed generated AD when the feedback was ambiguous. Whenever the feedback was certain, AD dropped strikingly. We found a similar pattern in another online sample (Supplementary Fig. 5a). Bars and dots show the mean AD proportion. **d** After the task, participants reported how many AD trials they generated per game ("In the study, how many times per game did you do badly on purpose?"). Their subjective reports of AD trials were significantly correlated with the objective frequency of AD trials identified in their behaviour (MRI, $\rho = 0.81$, $p < 0.001$, 95%-CI = (0.63, 0.91); online, $\rho = 0.85$, $p < 0.001$, 95%-CI =

(0.76, 0.90); two-sided). Again, these results replicate in another online sample (Supplementary Fig. 5b). Dots show individual participants, line ± shaded intervals are fitted regression curves with 95%-CIs. **e** Uncertainty is highest when a new phase starts (example block 3). Within each phase, uncertainty increases when the other's performance changes. **f**, **g** Trial-wise uncertainty is reduced if the previous trial or the one before was an AD trial (ANOVAs: 1 trial after AD, $F(1,65) = 301.54$, $p < 0.001$, $\eta^2 = 0.82$; 2 trials after AD, $F(1,65) = 282.17$, $p < 0.001$, $\eta^2 = 0.81$; see also Supplementary Fig. 8c). **h** Crucially, participants' rating errors are reduced after AD (1 trial after AD, $F(1,65) = 7.26$, $p = 0.009$, $\eta^2 = 0.10$; 2 trials after AD, $F(1,65) = 12.56$, $p < 0.001$, $\eta^2 = 0.16$), showing that they can use the additional information afforded by AD (similar to the Active learner). In **g**, **h**, we did not find significant group differences between samples (all $p > 0.05$). **g**, **h** Bars and error bars indicate mean ± s.e.m. beta estimates, grey dots for individual participants. **d** $n = 29$ MRI and $n = 64$ online (see Methods); **c**, **e**, **g**, **h** $n = 31$ MRI, $n = 36$ online; n.s., not significant; **$p < 0.01$; ***$p < 0.001$ (uncorrected).

Next, we tested whether in the trials immediately following AD, AD had reduced uncertainty and led to better credit assignment (measured as lower rating errors). Since participants did not self-report uncertainty in our task, we extracted from the Bayesian model ("Active learner", introduced before) a trial-wise measure of the overall uncertainty across self, other, and control (Fig. 3e). Previous work

showed that uncertainty estimates derived from such a Bayesian model are both good estimates of people's uncertainty (confidence) reports and they drive exploration[38]. We found that uncertainty was indeed reduced in the trials directly after AD (effect on uncertainty: 1 trial after AD, $F(1,65) = 301.54$, $p < 0.001$, $\eta^2 = 0.82$; 2 trials after AD, $F(1,65) = 282.17$, $p < 0.001$, $\eta^2 = 0.81$; Fig. 3f-g), and AD led to a

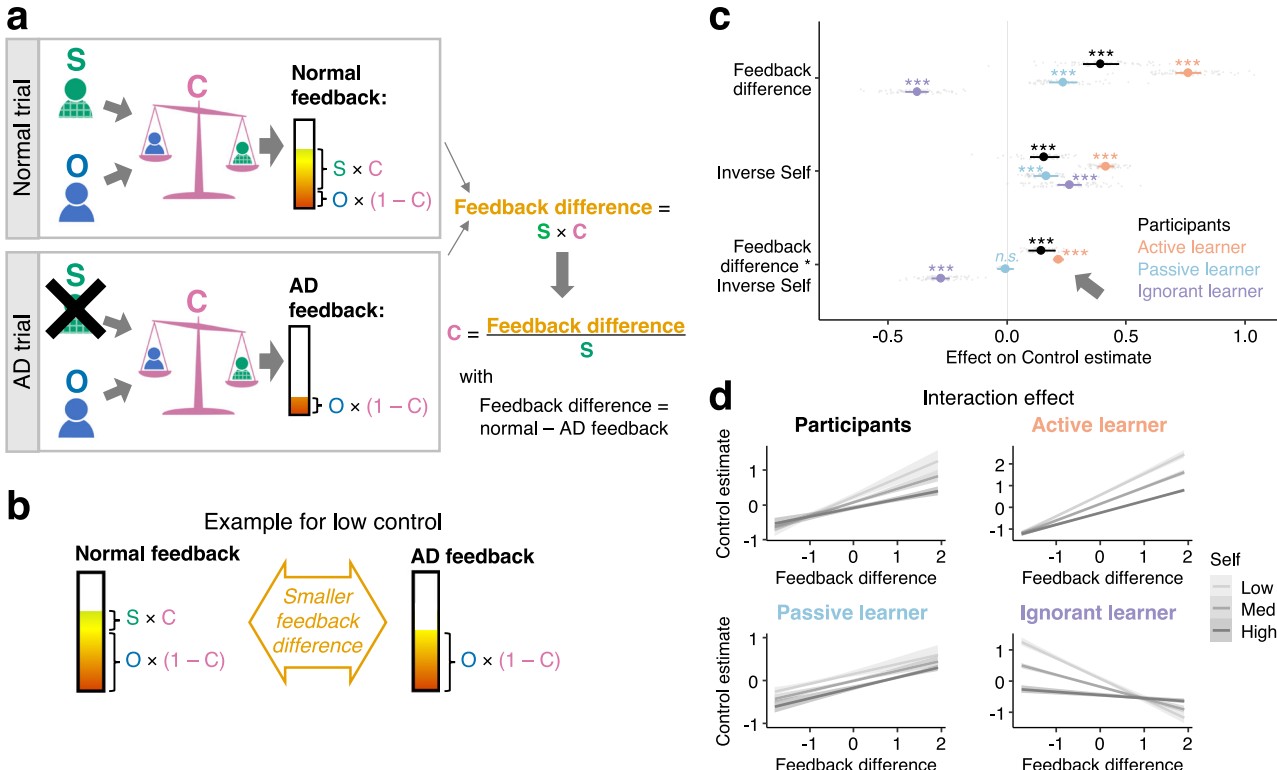

**Fig. 4 | Inferring control. a** On normal trials, Feedback=Self×Control+Other×(1-Control). On AD trials, the participant plays so badly that Self=0, so that the feedback only reflects the other player, weighted by their control. Therefore, control can optimally be inferred by observing the difference between normal and AD trials, accounting for self-performance (Control = Feedback difference/Self). **b** Intuitively, low control can be inferred if there is barely any difference between feedback on a normal versus an AD trial. **c** As expected and like the Active learner, after switch trials (either AD following normal trials, or vice-versa), participants inferred their control from the feedback difference between AD and normal trials, accounting for their own performance (Feedback difference*Inverse Self, β = 0.14, 95%-CI = (0.09, 0.20)). We ran the same Bayesian regression analyses on simulated data from two computational models that made wrong assumptions about AD trials. Importantly, all agents (participants, Active, Passive and Ignorant Learners) observed the same feedback participants received. We found that indeed the passive and ignorant learner showed different learning patterns: the interaction effect, Feedback difference*Inverse Self, differentiated between, on the one hand,

participants and the Active learner, and, on the other hand, the Passive and Ignorant learners. While participants and the Active learner had a significant positive interaction effect, the Passive learner had no significant effect, and the Ignorant learner had a significantly negative effect (Feedback difference*Inverse Self: Active learner, β = 0.22, 95%-CI = (0.19,0.24); Passive learner, β = −0.01, 95%-CI = (−0.04,0.03); Ignorant learner, β = −0.28, 95%-CI = (−0.32,−0.24)). Dots with error bars show mean ± CI beta estimates. Grey dots are individual participants. **d** Both participants and the Active learner infer a low control when there is a lower difference between normal and AD feedback. If there is a large difference, they infer a high control−but this depends on the level of their own performance. If low self-performance is assumed and there is still a high feedback difference, then a higher control is inferred. The ignorant and passive learner shows markedly different behaviour. For easier interpretation, we relabeled "inverse self" as "self" and flipped its levels accordingly (e.g., "low self" is "high inverse self"). Lines ± shading show mean estimates ± CI. n = 31 MRI, n = 36 online; n.s., 95%-CI includes 0; ***99.9%-CI excludes 0.

reduction in the model's rating errors (effect on errors of Active learner: 1 trial after AD, $F_{(1,65)}$ = 152.08, $p < 0.001$, $\eta^2 = 0.70$; 2 trials after AD, $F_{(1,65)}$ = 123.24, $p < 0.001$, $\eta^2 = 0.65$; black open dots in Fig. 3h). This shows that a Bayesian learner benefits from AD trials. Crucially, participants' actual rating errors were similarly also reduced after AD (effect on participants' rating errors: 1 trial after AD, $F_{(1,65)}$ = 7.26, $p = 0.009$, $\eta^2 = 0.10$; 2 trials after AD, $F_{(1,65)}$ = 12.56, $p < 0.001$, $\eta^2 = 0.16$; Fig. 3h). Note that we detected AD in all 31 MRI participants but only in a subset of the online sample ($n = 36$ out of 69 had usable AD data), suggesting differences between lab-based and online data collection.

Taken together, participants used AD when there was uncertainty to be resolved. After AD, both their uncertainty and their rating errors were decreased. This indicates that participants had a good intuition about when AD was useful for them, and that they used the additional information provided by AD in order to change their beliefs.

### Active disambiguation supports inference of controllability
We have shown that participants purposefully use ADs when they are informative, reduce uncertainty and enable better credit assignment.

Next, we analysed the mechanism by which AD trials made this possible. We reasoned that AD trials provided information important for inferring controllability when it is ambiguous, in the following way. Crucially, participants might compare feedback from normal and AD trials (Fig. 4a). From this difference, integrated across trials, they could infer their control as long as they also took into account their own overall performance level (which was known in this task phase). Intuitively, if making a deliberate mistake (AD) makes hardly any difference to the feedback compared to when doing one's best (normal trial), then a low control can be inferred (Fig. 4b). Conversely, if the feedback difference is high, then a higher control level can be inferred. While this is true even when high self-performance is assumed, it is especially the case when low self-performance is assumed; if low self-performance is assumed and there is a high feedback difference then a very high control level should be inferred. In summary, control estimates should be proportional to the feedback difference (normal − AD trials) divided by the self-performance level. This can also be rendered as the feedback difference multiplied by the inverse of the self-performance level (feedback difference * inverse self). We tested whether participants used this estimate in the Control-Other phase

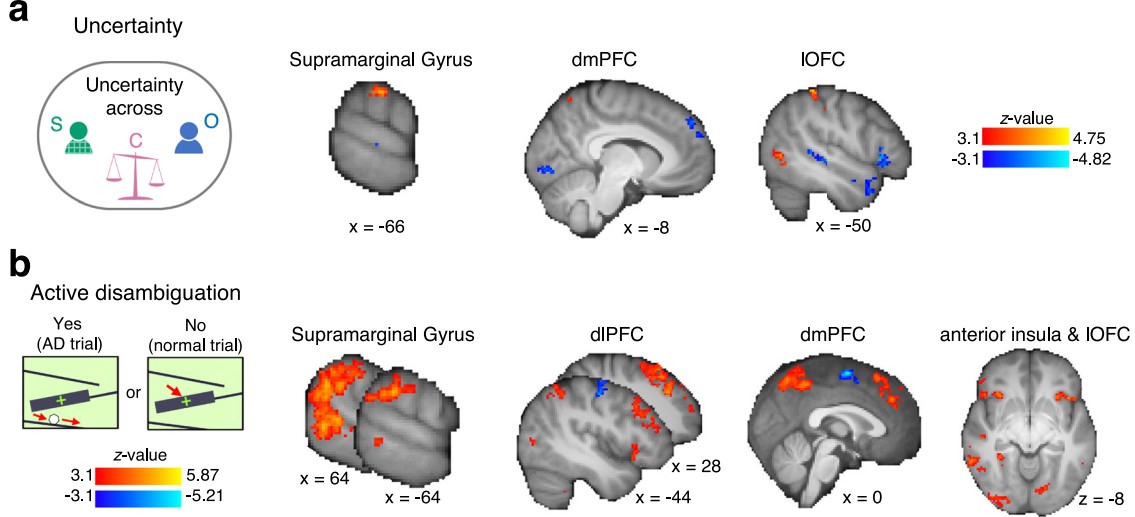

**Fig. 5 | Whole-brain maps for uncertainty and AD at the time of action.** Since we found that participants generated AD trials under uncertainty in both task phases, we collapsed the data across both Self-Other and Control-Other phases in this neural analysis (see "Methods" for full details). **a** Activity in the left SMG related to higher uncertainty. Lower uncertainty (higher certainty) was reflected in dmPFC and left lOFC. **b** We also found bilateral activation related to AD in SMG (with a peak in TPJ in the right hemisphere). Additionally, we found AD-related activity in bilateral dorsolateral prefrontal cortex (dlPFC), extending to dorsomedial PFC (dmPFC), and adjacent dACC, bilateral anterior insula, and lOFC. The peak coordinates of significant clusters can be found in Supplementary Table 1. $n = 31$ MRI, whole-brain effects are cluster corrected with z > 3.1 and $p < 0.05$; effects are time-locked to the time of action, when participants played the games, and are across Self-Other and Control-Other phase.

when control was unknown and had to be inferred. We found that participants indeed estimated their control by combining the information from normal and AD trials, taking into account their own performance (interaction Feedback difference * Inverse Self, $\beta = 0.14$, 95%-CI = (0.09,0.20); Fig. 4c).

To test whether the behaviour observed is indeed the behaviour expected from a learner that does perform AD trials and that knows that AD trials were performed (rather than assumes that a trial reflects its best performance), we compared participants' behaviour to the Active learner (introduced before) and two alternative Bayesian learning models we refer to as the "Ignorant" and the "Passive" learners (Supplementary Fig. 6). All learning models were given the same data from participants. However, the second and third models made wrong assumptions about the feedback: The Ignorant learner assumed that AD trials are not AD but normal trials. This mimics someone who does not realize that these trials are intentionally performed to achieve AD, and instead assumed the outcome reflects the best performance the agent was capable of, as was the case on all other trials. The Passive learner does not learn from AD trials at all, but only from normal trials. This is like someone who gets confused by the AD trials and discards that information from learning. Participants' behaviour was consistent with the Active learner model, but importantly, different from the two alternative learners (interaction Feedback difference * Inverse Self: Active learner, $\beta = 0.22$, 95%-CI = (0.19,0.24); Passive learner, $\beta = -0.01$, 95%-CI = (−0.04,0.03); Ignorant learner, $\beta = -0.28$, 95%-CI = (−0.32,−0.24); Fig. 4c). Both the Active learner and participants inferred high control when the feedback difference was high, and even more so if the self-performance was low, because then a large feedback difference must be explained by higher control (Fig. 4D, top panel). In contrast, if the feedback difference is low, the Active learner and the participants inferred a low control level regardless of the level of self-performance. The ignorant and passive learner show a very different result pattern for the interaction effect (Fig. 4D, bottom panel), suggesting that AD, and insight into when AD occurs, are essential to infer controllability.

We considered other ways in which participants might have used equivalent strategies to infer self and other, and show that participants did not use these strategies to the same degree to infer control

(Supplementary Fig. 9 and 10). After finishing the task, participants also reported that they found AD trials most helpful for inferring their control (Supplementary Fig. 9f). This suggests that when control was uncertain, participants used AD efficiently for learning about their control and that they had insight into this.

In sum, when control was ambiguous, participants used the information afforded from AD in order to infer their level of control over the feedback. They did so by varying their behaviour and observing the resulting feedback changes. Comparing their behaviour with three computational models suggests that this was a deliberate strategy which requires insight into the nature of AD.

## Supramarginal gyrus tracks uncertainty and active disambiguation at the action

Next, we analysed the neural data to identify the neural correlates of AD and AD-guided controllability estimation. We first examined neural activity when participants engaged in AD to resolve uncertainty. For this, we ran a whole-brain analysis to identify the neural correlates of AD and trial-wise uncertainty (from the Active learner model, as before) at the time when participants initiated action during the games.

The left anterior supramarginal gyrus (SMG) was sensitive to both uncertainty (Fig. 5a) and AD (Fig. 5b). For AD, this activation was bilateral (extending to and peaking in the temporoparietal junction [TPJ] in the right hemisphere). This means that activity in the SMG signals uncertainty as well as AD generation at the time of the action. While uncertainty-related activations were in the SMG, we also found some uncertainty-linked deactivations (Fig. 5a). While activations for AD overlapped with uncertainty-related activity in SMG, they were more widespread (Fig. 5b).

Our analysis incorporated three important controls. First, the activations emerged after controlling for the variety of game types participants played during the course of the experiment. Second, analyses controlled for variation in reaction time. Third, although ADs occurred intermittently amongst other normal trials, AD-associated activity reflects AD per se rather than the more general process of switching between AD and normal trials; a control regressor in the analysis indexed switches (a trial on which participants switched from a normal to an AD trial or vice-versa).

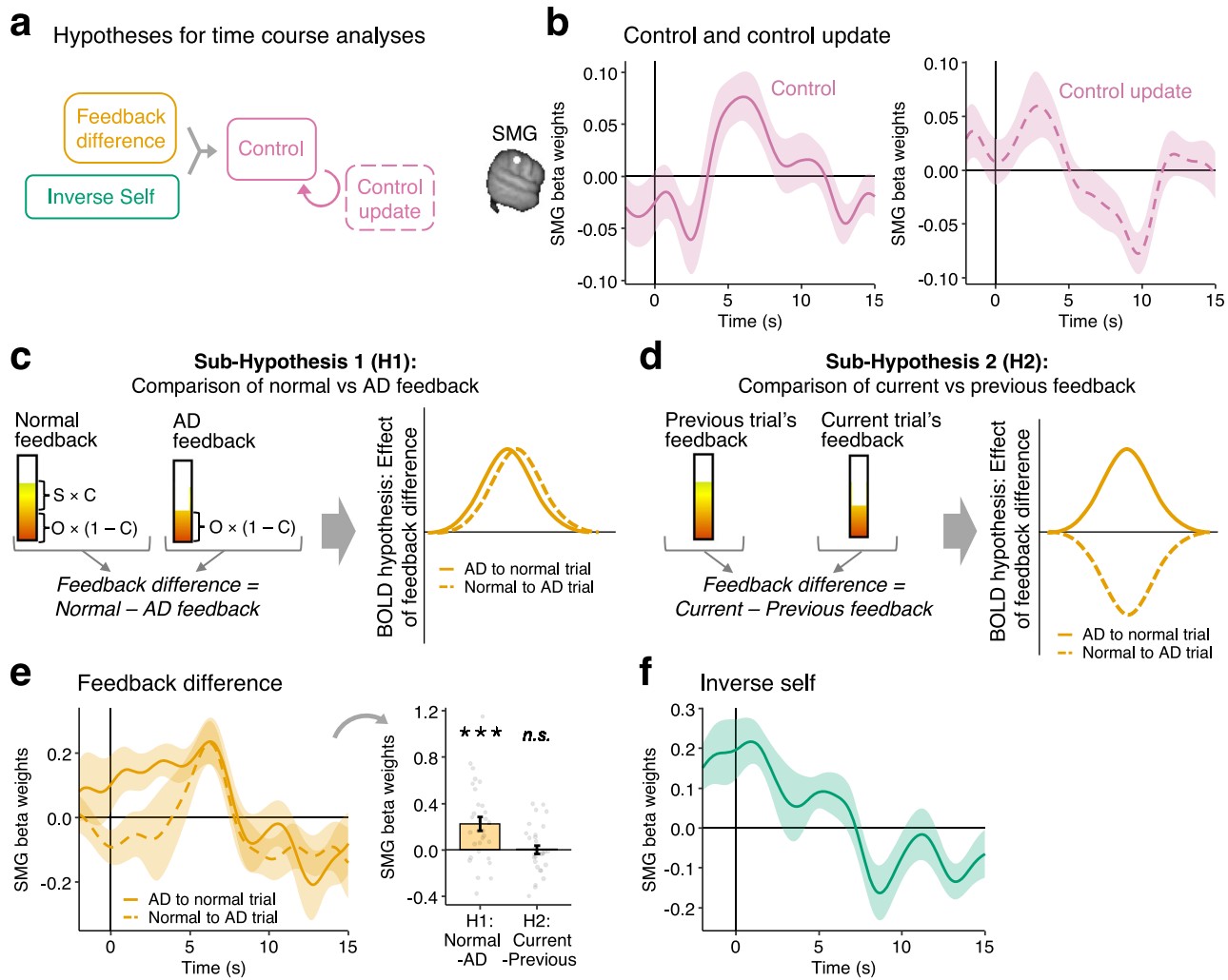

**Fig. 6 | Representations of control, feedback difference and inverse self in the SMG at the time of outcome in the Control-Other phase. a** We used time-course analyses to identify the neural correlates of how participants infer their control from feedback difference and the inverse self. **b** We indeed found that SMG tracks the inferred control estimate and the degree of control level updating at the outcome (ROI-GLM1). These activity patterns remained unchanged whether control and control updates were examined in the same or separate GLMs. **c** We next tested whether SMG also represents the components necessary for inferring control. We compared two neural hypotheses for how neural activity can track the difference in feedback between trials. Our first hypothesis (H1) is that neural activity tracks the difference between normal and AD trials. This means that the neural activity does not depend on order: whether a normal trial followed an AD trial, or vice-versa. **d** Alternatively (H2), neural activity might track the feedback difference in a simpler manner, by comparing the currently observed feedback to the feedback from the previous trial. We used a single GLM (ROI-GLM2) to test these two hypotheses. For

this, we computed feedback difference as normal vs AD (according to H1) in all panels. However, we split trial transitions between AD to normal trial transitions and vice versa, because H1 and H2 make different predictions between those trial types. **e** At the outcome (ROI-GLM2), we found a significant effect of feedback difference in accordance with H1 rather than H2, suggesting that the SMG represents the feedback difference as normal vs AD feedback. The right panel shows participants' aggregated peak amplitudes using a LOO procedure (H1: $t(30) = 3.77$, $p < 0.001$; H2: $t(30) = 0.06$, $p = 0.952$; two-sided one-sample t-test). Feedback difference and control (**e-b** left) have low regressor correlations ($r < 0.3$, Supplementary Fig. 9A), and are run on different trial selections (control on all trials, feedback difference on switch trials only). **f** In the same GLM (ROI-GLM2), we found that SMG also tracks participants' prior inverse self-estimate. $n = 31$ MRI; lines/bars show mean, error bars/shaded intervals show s.e.m.; n.s., not significant; ***$p < 0.001$; effects are time-locked to the outcome phase; **b** includes all trials from the Control-Other phase; **e, f** include only switch trials in the Control-Other phase.

## Supramarginal gyrus carries neural signals for learning controllability

Our behavioural results showed that participants inferred their control from the observed feedback difference between AD and normal trials. Next, we established the neural mechanisms mediating the inference process at the time of outcome when the feedback is shown (Fig. 6a). We tested this in the SMG as the area sensitive to both uncertainty and AD at the time of action. As in the behavioural analysis, we focused on the Control-Other phase when participants learnt their control level and again focused on how participants compared AD and normal trials. First, we tested if, at the time of outcome, the SMG represented the control estimate and the updating of control (ROI-GLM1). We indeed

found that the SMG represented the inferred control level ($t(30) = 2.34$, $p = 0.026$, Fig. 6b left). Additionally, SMG activity was modulated by the control update from current to the next trial ($t(30) = -3.79$, $p < 0.001$, Fig. 6b right).

Next, we tested in a separate GLM whether neural activity reflected the components necessary for control inference. We contrasted two competing hypotheses for how neural activity might reflect the feedback difference between trials: First, neural activity might represent the feedback difference computed as the difference between normal minus AD trial (hypothesis 1 [H1], Fig. 6c). This coding scheme encapsulates a comparison that directly leads to control inference (see also Fig. 4a). Alternatively, neural activity might track the feedback

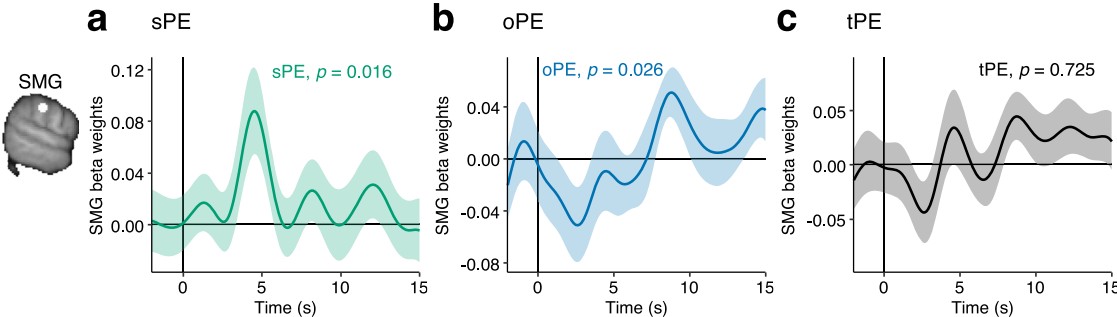

**Fig. 7 | Representations of sPE and oPE in the SMG during the outcome phase in the Self-Other phase.** In the Self-Other phase, the key credit assignment problem that participants face is to assign credit to themselves and other people according to their control. On normal trials, they split up the tPE into a sPE and oPE according to their known control (see also Fig. 2a, e). We therefore tested whether BOLD activity in the left SMG is modulated by sPE and oPE. Since sPE and oPE are correlated with $r = 0.55$, we tested this both with two separate GLMs containing either sPE but not oPE and vice versa, and a GLM in which both variables were included. All the analyses, however, led to the same conclusion illustrated in the next two panels. **a** We found that at the time of the outcome, activation of the left SMG is significantly related to the sPE ($t(30) = 2.55$, $p = 0.016$, ROI-GLM3). **b** In a separate GLM, we found that SMG BOLD activity is significantly modulated by the oPE ($t(30) = 2.34$, $p = 0.026$, ROI-GLM4). **c** In a separate analysis, we found that the SMG does not represent the tPE which is the sum of sPE and oPE ($t(30) = -0.36$, $p = 0.725$, ROI-GLM5). This is an interesting feature and together with panels a and b suggests that the SMG is involved in belief updating in an agent-based manner, rather than the total outcome per se. $n = 31$ MRI; lines show mean, shaded intervals show s.e.m. across participants; effects are time locked to the outcome phase and analogously to our behavioural analysis, only normal trials in the Self-Other phase are included. Reported $p$-values originate from two-tailed one-sample t-tests against zero, applied to the peak values from the LOO procedure.

difference by comparing the current trial's feedback to the previous trial's feedback (hypothesis 2 [H2], Fig. 6d). Such a contrast might be simpler to perform as each new piece of feedback is obtained and it resembles the types of contrasts at the heart of prediction error learning mechanisms when the current trial outcome is compared with weighted estimates of previous trial outcomes. However, despite its simplicity this contrast does not lead directly to control inference.

In summary, we found that at the time of outcome, the SMG activity reflected a comparison between normal and AD feedback in support of H1, rather than H2 (H1: $t(30) = 3.77$, $p < 0.001$; H2: $t(30) = 0.06$, $p = 0.952$; ROI-GLM2, Fig. 6e). We ruled out that this signal reflected AD per se by including a categorical regressor for whether the current trial was AD. In the same GLM, we tested for the inverse of participants' prior estimate of self-performance (inverse self), the other component mediating control inference. We found that SMG activity was also sensitive to the inverse self-estimate (i.e., activation with high self-rating, $t(30) = -2.27$, $p = 0.031$; ROI-GLM2, Fig. 6f). We also observed a relatively early peak linked to the inverse self. While it lay outside our pre-defined time window of interest, a leave-one-out (LOO) procedure identified the effect as significant (−2 to 2 s time window: $t(30) = 3.69$, $p < 0.001$). This early peak could result from the self being a prior estimate that is known ahead of the feedback. In Supplementary Fig. 14 we consider another brain region where uncertainty and AD activation clusters overlap.

In summary, when control was ambiguous, SMG signalled all component variables needed for control inference, including the observed feedback difference between AD and normal trials, the self-performance estimate, and the updating of the control estimate at the time of the outcome. Crucially, it also represented the inferred control level at the same time. In addition, SMG activity reflected the initiation of the AD process and the uncertainty levels that, when high, preceded AD.

### Supramarginal gyrus tracks prediction errors assigned to self and others

Our behavioural analyses showed that when participants know the level of their control over outcomes (Self-Other phase), they assign prediction errors to themselves (sPE) and others (oPE) by taking into account control (Fig. 2e). Next, we examined the neural correlates of this process time-locked to the time of credit assignment (i.e., outcome phase, see also Fig. 2A).

Given the activity patterns we had already identified in SMG, we again used the SMG as an ROI. We indeed found that SMG activity was modulated by PEs assigned to self and other (sPE, $t(30) = 2.55$, $p = 0.016$, ROI-GLM3, Fig. 7a; oPE, $t(30) = 2.34$, $p = 0.026$, ROI-GLM4, Fig. 7b). Due to the correlation between oPE and sPE ($r = 0.55$, Supplementary Fig. 13b), we tested their effects in separate GLMs (as reported before, Fig. 7a, b), and also together in one GLM (sPE, $t(30) = 3.84$, $p < 0.001$; oPE, $t(30) = -2.41$, $p = 0.022$; Supplementary Fig. 15a, b, ROI-GLM5). Each time, oPE and sPE effects remained significant in the SMG. Alternatively, we tested whether the SMG might track the tPE (the total prediction error or the sum of sPE and oPE). However, we found no evidence that the SMG tracked the overall tPE nor did it track control in this task phase, when participants had already been instructed about its level and it did not need to be inferred (control, $t(30) = 1.43$, $p = 0.163$, Supplementary Fig. 15c; tPE, $t(30) = -0.36$, $p = 0.725$, Fig. 7c; ROI-GLM6). We consider another brain region where uncertainty and AD activation clusters overlap in Supplementary Fig. 16. Together, our results suggest that in contexts when people already know their control, the SMG is involved in processing outcomes in an agent-based manner, attributed to either self or others.

Finally, we tested whether neural activity tracked reward in our task, irrespective of any inference process. The aim of this analysis was to check whether, in our task, we can find more classic reward signals. The ventral striatum has been implicated in representing reward in past studies. In an anatomically defined ROI of the ventral striatum, we found it indeed tracked the shown feedback, as well as trial-wise changes of the feedback at the time of the outcome (feedback, $t(30) = 2.23$, $p = 0.034$; feedback change, $t(30) = 2.48$, $p = 0.019$; ROI-GLM7, Supplementary Fig. 17). This suggests that our task also allows investigating more traditional reward-related signals in the human brain.

Overall, in situations where control is explicit, SMG tracked outcomes as social PEs, which are modulated by individuals' control beliefs. In contrast, the ventral striatum signalled the observed feedback. Figure 8 shows a result summary of this study.

## Discussion

A long-standing issue of interest in psychology and cognitive neuroscience, and one that has influenced the understanding of psychological illness, has been how humans and other animals learn which aspects of their environment are controllable[39]. Recently there has

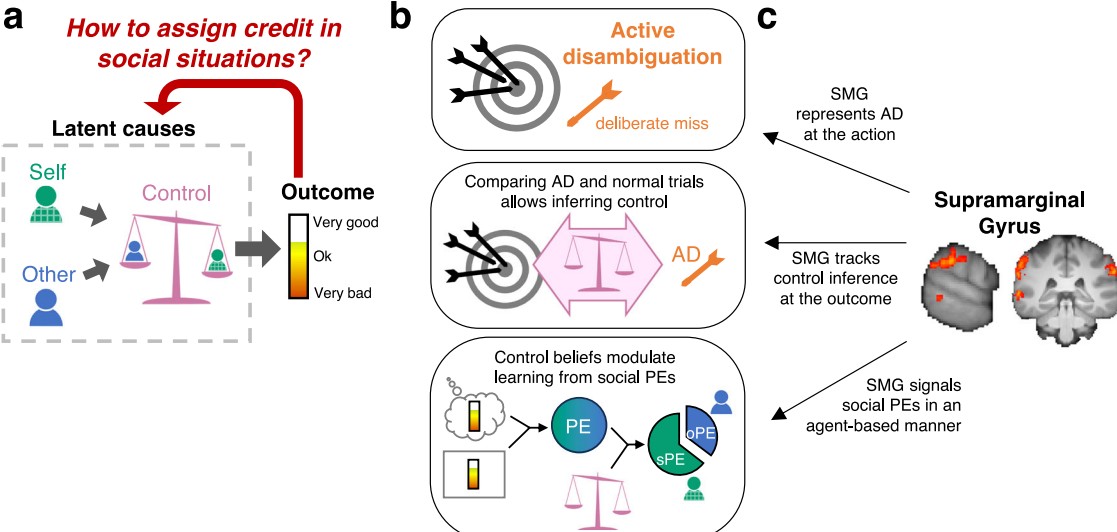

**Fig. 8 | Controllability in social credit assignment and the role of the SMG.**
**a** People often work together to achieve their joint goals. Therefore, whenever they experience an outcome for their joint efforts, they face the fundamental credit assignment problem of knowing how much to attribute ambiguous outcomes to themselves, others and their level of control. **b** Participants varied their behaviour by making deliberate mistakes ("active disambiguation", AD) when it was useful for credit assignment. They had insight into AD, and it allowed them to reduce uncertainty and assign credit more accurately. When control was unknown, they employed this behavioural strategy to inform their control estimates, by comparing outcomes obtained through AD and normal performance. When participants knew their level of control, they used this knowledge to assign ambiguous PEs to themselves and others appropriately. **c** Neural activity in the supramarginal gyrus tracked each of these processes. At the time of action, it signalled AD (shown are whole-brain cluster-corrected maps of the effect of AD at the time of action, see also Fig. 5b). At the time of outcome, it tracked both the inferred level of control and the components necessary for this learning process. Finally, SMG activity represented the portions of the PE attributed to oneself versus another person.

been renewed interest in this topic inspired by new computational approaches to learning[24,27,28,40]. Here we examined what might be considered an instance of this problem that is prominent for a social species such as humans: how we learn the extent to which we ourselves, as opposed to others, have responsibility or agency over events. Estimating control in this way goes hand-in-hand with estimating performance levels – how good at a task are we ourselves and how good is the other person we are working with.

We investigated how in cooperative two-person games, individuals assign credit to either themselves or others, while also estimating their control over the outcomes (Fig. 1). Participants appropriately assigned credit by attributing prediction errors to either themselves or the other person after taking into account their control (Fig. 2). The credit assignment problem in the games was inherently ambiguous and difficult to solve. To overcome this, participants used a behavioural strategy we refer to as "active disambiguation" (AD). They employed this tactic especially when there was high uncertainty about who was responsible for what (Fig. 3). AD enabled participants to more accurately estimate their control by comparing the outcomes observed during AD with those on non-AD trials (Fig. 4). Activity in SMG, but no other motor association areas, reflected a comparison of AD and non-AD trials (Fig. 6e). This comparison leads directly to an estimate of the participant's control and SMG also held control-related activity (Fig. 6b). Finally, identifying control from AD is intertwined with estimating one's own performance level (Fig. 6a) and the third activity pattern carried by SMG reflected the participant's own performance level (Fig. 6f). The SMG also tracked prediction errors in an agent-based manner, in relation to control (Fig. 7).

Our findings suggest that AD is recruited as a general strategy to resolve uncertainty and aid credit assignment, and that SMG activity reflects this process. In the current task, AD supported inference of controllability specifically when control was ambiguous, but it was also active when there was uncertainty about self- or other-performance levels. Thus, our results point to a broader contribution of both AD and

SMG to uncertainty resolution across different dimensions of social credit assignment.

A key learning mechanism exhibited by participants was one of active information seeking. People inferred their control by observing how outcomes change when they varied their behaviour, to the extent of even performing badly on purpose during AD. AD resembled other forms of exploratory behaviours that are guided not simply by the immediate prospect of reward, but also by the potential for information acquisition[38,41–46]. Importantly, in our task, participants do not choose among options that simply differ in information content, but instead they actively participate in creating the information they seek by performing exploratory actions that test specific hypotheses about levels of agency. There is increasing interest in how information seeking is tied to active hypothesis testing[1,47–50]. A similarly active process may be at work when inferences about an environment's level of controllability are made[27,28]. Crucially, in our task, the key question does not concern establishing the extent to which outcomes are contingent on our actions beyond existing contingencies between outcomes and states of the environment. Instead, the focus is on comparison of the degree of action-dependent contingency exhibited by each actor – the participant themselves and the other person who they work together with in the task.

At the core of control inference studied here is a comparison process and prior work highlighted that control can be inferred through comparing either observed and predicted action outcomes[51] or the likelihoods of outcomes in the presence or absence of one's actions[27,28,52,53]. Ligneul et al.[27] studied how people track their control over categorical outcomes by comparing a "spectator" and "actor" model in non-social contexts. Here, we show that a similar comparison process involving exploratory actions is at play in social situations. Several differences likely account for our divergent neural findings, with our results implicating the SMG, in contrast to the prefrontal regions reported by Ligneul et al. First, Ligneul et al. studied controllability inference in non-social settings, where the challenge was to

establish whether one's own actions influenced state transitions. In contrast, our paradigm involved social contexts, in which participants had to disambiguate their own versus another person's contribution to shared outcomes. Second, Ligneul et al. examined control over categorical, geometric stimuli and their transitions, whereas our task focused on control over parametric feedback reflecting self- and other-performance that also varied in a similarly continuous manner. Third, in Ligneul's study, exploration meant choosing between options to generate informative transitions, while in our task it consisted of deliberately reducing one's own performance to reveal one's own versus someone else's contribution. Taken together, the two lines of work suggest that exploratory behaviour is a general mechanism supporting controllability inference, but that the neural substrates recruited may differ depending on whether the challenge is arbitration between actor-spectator models in non-social contexts or disambiguating of self-other agency in social contexts. Engaging in exploratory actions also allows sensing one's control over different objects in non-social contexts[54]. On an abstract level, there are some similarities of this AD-guided control inference to prior work showing that people take advice despite already being highly confident[55]. While initially appearing suboptimal, both behaviours are beneficial for long-term learning about the quality of advice[55] or controllability.

In the current task, participants' metacognitive introspection was disrupted by the manner in which objective performance was mapped to a point score and by features of the stimulus display that obscured aspects of the ball's path. However, in more natural settings it is likely that metacognitive assessment of performance might also contribute to a person's estimate of control.

More generally, AD-guided learning can be considered a form of intervention-based causal learning[56,57]. Sometimes the contingencies between an agent and the environment can only be ascertained by actively intervening in their environment and observing the consequences. Learning in development may also depend on interventions[58]. From childhood to adulthood, people get better at estimating controllability and change how they explore to assess control[59]. AD is also, arguably, the target of some cognitive therapies for depression ("behavioural experiments")[60–62]. The current results suggest that a similar process of inference and information seeking is inherent in social contexts.

In the current task, SMG appeared to be central to credit assignment and identifying agency. Other brain regions, such as area 47/12o on the border between ventrolateral and orbitofrontal cortex, have been linked to credit assignment in non-social contexts[2,4,13,63]. However, monitoring the consequences of one's own and others' actions may depend on SMG[22,23,29], and therefore, make SMG important in establishing agents' relative levels of control—their degrees of agency—during social interaction.

We found widespread neural activation related to AD and to uncertainty about control and performance levels prior to AD at the time that participants generated actions (Fig. 5). While activity in some components of the distributed network, such as SMG, may also be closely linked to credit assignment and agency identification processes, activity in dorsomedial prefrontal and adjacent cingulate cortex and in dorsolateral prefrontal cortex may be more closely linked to the process of exploration itself[38,45,64].

In this study, we studied interpersonal control of joint task outcomes—how well is a task performed together by two people. A complementary approach has been to ask how people coordinate the actions they each make with one another in order to move a particular physical object[65]. Typically, the focus in such studies has been on how people learn to predict each others' actions so that they can be coordinated together. However, a crucial element of such coordination is ascertaining whether joint goals are accomplished and the focus of our current study can be thought of as understanding how we learn the

nature and degree of our contribution to the accomplishment of a joint goal. It has been noted that the sense of agency or control can be understood on different levels such as the sensorimotor and the cognitive level (referred to as implicit "feeling of agency" and explicit "judgement of agency", respectively[66,67]) and this can be translated from individual to joint actions in social interactions[68]. Again, different research strands have tended to focus on either the cognitive mechanisms of feeling in control[24,27] or control at a more sensorimotor level[69,70]. In psychiatric conditions, control is also used as a term to study how responsible people feel for outcomes. For example, negative attributional styles in depression describe how depressed people tend to feel a heightened sense of control over negative life events, but no or a low level of control over neutral events (depressive realism[6,71]).

Here, we studied how humans assign credit and infer their control in social settings. We rewarded participants only for how well they assigned credit. This allowed us to delineate the process of credit assignment from that of maximising collaborative outcomes. It is interesting that nevertheless, we found feedback-related signals in the ventral striatum reflecting how well the games were performed on each trial. Previous work, however, has investigated how humans decide to collaborate to do better[72]. Future work can investigate the degree to which our findings change when participants are rewarded not just for assigning credit correctly, but also for maximising the joint feedback they receive as a result of game performance. This might lead to a sparser but more efficient use of AD to guide learning. Furthermore, future studies should test whether our findings on AD behaviour generalise beyond dyadic interactions to larger social groups. It might be that AD-guided control inference is even more prevalent in larger groups, where the diffusion of responsibility can be greater and disentangling one's own contribution from that of others is more difficult.

While all our MRI participants exhibited AD behaviour, only about half of the online sample did so. This could point towards differences between lab-based and online data collection. Alternatively, it could indicate that some individuals may rely on more passive observation to disambiguate causes. It would be an interesting future avenue to characterise the factors influencing why some individuals adopt AD readily while others do not.

While our study focused on healthy adults, understanding control and agency estimation may inform our understanding of psychological illness. For example, depressed people estimate their level of control over uncontrollable outcomes differently ("depressive realism")[5,6]. Cognitive therapies of depression target aberrant attributional styles (attributing negative outcomes to self)[73–76]. Psychosis has also been linked to difficulties in attributing experiences to oneself or external sources[77–79]. Individuals with schizophrenia or schizophrenia-like traits show delusions of control[7,8,80]. Understanding better how we come to understand our own levels of agency in a social context may, therefore, shed light on how this estimation of ability, perhaps even self-worth, is altered in psychological illness and suggest new methods of treatment.

## Methods
### Participants
36 healthy participants took part in the fMRI experiment (sample taken from a larger study, see supplementary methods), and 69 participants took part in the online experiment. 2 MRI participants were excluded because they misunderstood how they should rate their own performance in the games (they rated their previous trial's performance rather than their overall best attempt, meaning that they rated their performance lower after an active disambiguation (AD) trial; one of these two participants reported having changed how they did their self-ratings midway through the experiment). These misunderstandings were prevented in the participants collected afterwards by an additional task-understanding question administered by the experimenter. 3 additional MRI participants were excluded because they

performed fewer than 3 AD trials in the Control-Other phase. For the online sample, 3 participants were excluded because they took breaks longer than 5 min during the task blocks, and 9 participants were excluded because they misunderstood how to rate their own performance. In addition and for almost all of our analyses (except the correlation analysis shown in Fig. 3d), we excluded 30 online participants who did not perform AD in the task or reported never having done AD in the post-task debrief questionnaire. The final samples included 31 MRI participants (self-reported gender: 22 female, aged 18–33 years), and 36 online participants (self-reported gender: 21 female, 1 diverse, aged 18–40 years). No statistical methods were used to pre-determine sample sizes. Our sample size was chosen as larger than those reported in previous publications[38]. Participants were not allocated into experimental groups. We do not include further analysis of gender because it was not applicable to our research questions. MRI participants were paid £50, plus a bonus based on task performance (range: £2–4). Online participants received £7.5 per hour plus a performance-dependent bonus (range: £0.5–2). The study was approved by the Medical Sciences Inter-Divisional Research Ethics Committee (MSD-IDREC, R40628/RE005 for MRI, R54722/RE008 for online) at the University of Oxford. Details on recruitment and criteria can be found in the supplementary methods.

## Experimental procedure

MRI participants took part in a single testing session (full details on recruitment in supplementary methods, Participants and recruitment). After providing written informed consent, they were instructed about the task and performed a task practice outside of the scanner (details can be found in the supplementary methods, Task instructions). During the training, participants practised each of the games. They also practised one trial with ratings, game and example feedback. Afterwards, they filled in a questionnaire assessing their task understanding. They then completed the main task inside the scanner. Afterwards, outside of the scanner, they filled in a task debrief questionnaire and psychiatric questionnaires (see supplementary methods, not analysed here).

Online participants also took part in a single session. After providing electronic informed consent (by ticking a box on the study website), they followed a similar experimental procedure as MRI participants (full details in supplementary methods, Participants and recruitment). The experimental task paradigm was created using JavaScript. JATOS[81] was used for experiment hosting and data collection.

## Experimental design

**Trial structure.** In the task, participants had to infer, from feedback, their own performance level in a game, another player's performance level in the game, and their control over the feedback. To measure participants' current estimates of these three variables, each trial started with a rating phase (reaction time, RT-paced). Depending on the current task phase, participants rated different combinations of their own performance, the other's performance, and their control level on every trial (details below). After completing the ratings, participants played one round of a short and simple game (action phase, RT-paced). For example, in the rockslide game, they pressed a button to indicate when they thought a rolling ball would hit a target cross. After the game, they were shown a single piece of feedback. The feedback consisted of a parametrically varying score reflecting the aggregate of the participant's own performance, the other's performance and their control over the feedback (outcome phase, RT-paced). Participants pressed a button to continue and move on to the next trial. After the outcome phase, a short interval was included (inter-trial interval, ITI), during which a fixation cross was shown. For the MRI sample, the duration of the ITI was drawn from a Poisson distribution with a range of 2–7 s, in order to decorrelate variables of interest

between trial phases. For the online sample, the ITI had a fixed duration of 500 ms.

**Task phases.** Participants completed 144 trials, divided into four blocks. In each task block, participants played one game. Each block was further sub-divided into three distinct phases during which participants played the current game together with different players and different control level settings (Fig. 1d). The beginning of each phase was explicitly signalled to participants with an instruction screen. The first 16 trials of each block belonged to the Self-Other phase. Here, participants were informed of the true control level and so only had to infer their own and the new other's performance. This allowed us to examine how participants assigned credit to themselves and others under varying levels of control. After completing the Self-Other phase, four trials followed which were the Self-only trials. In the Self-only trials, there was no other player and participants had full control over the feedback. This meant that the feedback directly reflected participant's self-performance on the given trial. Therefore, participants could double-check and correct their self-estimates during those trials. Afterwards, the third task phase followed, the Control-Other phase. During this phase, participants played with a new other player and their control level reset and was unknown. This meant that while they now knew their own performance level (self-performance) for the game, participants now had to infer from the feedback the new other's performance and their control level. Overall, participants were incentivised to provide ratings as accurately as possible, rather than playing the games as well as they could. For each task block, they received a bonus that depended only on how accurate their ratings were, and not on their actual game performance. Additionally, the cover story of the task was that they were team captains and had to assess other players' performances in order to select teammates and assign control for each game. Therefore, at the end of each task block, they selected one of the two players they had played the current game with, and also, they chose how to distribute the control over the feedback from the two control levels they experienced.

**Ratings.** Which of the three ratings (self, partner, control) were completed on every trial depended on the current task phase (Fig. 1c). In the Self-Other phase, participants only rated their own and the other's performance. Here, the control rating slider indicated the true control level and could not be moved by participants (see also Fig. 1c). In the Self-only trials, participants only rated their own performance, and the control and other rating slider were not shown. In the Control-Other phase, participants rated their control level, self and other performance (Fig. 1c). On the first trial of each phase, participants provided their ratings according to their 'gut feeling' which gave us a measure of their prior expectations about self, other, and control, before any feedback observations. For both self and other performance, participants were instructed to rate their best past attempt at the game, rather than the last trial (example rating questions: "On your best attempt, how good are you at the rockslide game?", "On their best attempt, how good is the other player at the rockslide game?"). This ensured that participants rated their overall abilities on normal trials as opposed to active disambiguation (AD) trials. Self and other were rated between 0 and 100, labelled as 'very bad' and 'very good' respectively. Control was rated (or, in the Self-Other phase, shown) on a scale between 0 and 1, ranging from 'only other' to 'only me'. All three rating sliders were continuous with no numerical values shown on the scales. Participants could position the rating handle on any value between 'very bad' and 'very good' for self and other, and 'only other' and 'only me' for control. To reduce motor confounds, the position of the rating pointer was initialised to a random position for the MRI sample. MRI participants then used the left and right buttons to move the pointer to the desired position, and pressed the middle button to submit their rating. Online participants used their mouse cursor to rate and clicked

on a 'Continue' button to submit their ratings (i.e., there was no initial rating pointer position).

**Games.** Each block corresponded to one game so that participants played four different games throughout the whole study. The exact nature of these games is less important than the fact that they were a vehicle to investigate how participants form beliefs about their own and others' performances (for a previous example of this strategy, see refs. 17,18). Supplementary Fig. 1 shows all four games. For example, in the Rockslide game (Fig. 1b), a ball rolled down a set of slides and participants pressed a button when they thought that the ball passed a target cross. What made the game difficult to play was that the ball's path was hidden behind a large black box that was centred around the target cross. This meant the participants had to guess when the ball passed by the target cross. On every trial, the position of the target cross on the middle slide changed slightly. The participant was told that the fictive other player would play the games at the same time but that they would be unable to see them play. This means that participants could not directly and accurately perceive how well they themselves had performed nor could they directly perceive how well the other player had performed in the current game. Therefore, they had to infer their own and the other's performance from the feedback shown to them after the game phase. While they could not directly perceive their performance level accurately, they were able to perceive large, purposefully made errors that were made on AD trials.

**Feedback.** After playing the game, participants were shown an overall feedback (outcome phase). This feedback reflected their own performance in the game on this trial, the other's performance on the current trial, weighted by their respective control levels. The following formula was used to compute the parametric feedback value on each trial:

$$\text{feedback}_t = \text{self}_t * \text{control}_t + \text{other}_t * (1 - \text{control}_t) \quad (1)$$

where *feedback* is the parametric feedback value on the current trial $t$. *Self* is the self-performance on the current trial, and depends on how well participants played the game just before.

Importantly, note that we used veridical self-performance feedback, but nonetheless precisely controlled participants' self-performance in our experiment. We did this, essentially, by manipulating the mapping between the objective game performance to self-performance score. We used a logistic function to transform participants' objective game performance into a self-performance score that is manipulated according to our pre-determined schedule (see next section for details). *Control* is the control level the participant exerts over the feedback in the current phase, while *(1-control)* is the control level the other player exerts. *Other* is the other's performance level on the current trial, pre-determined according to the experimental schedule (see section on schedules). On Self-only trials, the feedback was also computed according to this formula, but here *control_t = 1* so that the *feedback_t = self_t*, meaning that the feedback directly reflects the self-performance.

**Mapping of objective game performance to self-performance scores.** We used a logistic function to transform participants' objective performance (absolute distance from target in pixels or ms, depending on game) on a given game round into a point score (*self-performance*) that then contributed to the overall feedback. This mapping of objective performance error to self-performance was precisely calibrated to participants' objective performance in the games. The goals of this mapping were 1) to manipulate the mean self-performance of each participant according to our pre-defined schedule, while 2) ensuring that, even after mapping, self-performance still reflected variations in the participant's true objective performance. This means that if on a given trial participants did worse than in the last

trial, the self-performance value was lower (and vice-versa for a more accurate objective performance). This mapping procedure means that we disrupted participants' metacognitive assessment of their performance. This mirrors real-world situations where, for example, taking a certain number of seconds to run 50 m can only be interpreted as "good" or "bad" performance when compared to others (or one's own previous performance). An illustration of the mapping of objective onto self-performance can be found in Supplementary Fig. 2, and details on the exact mapping procedure are given in the Supplementary Methods.

**Active disambiguation.** Participants could change their behaviour in the games in order to observe its effect on the feedback. Specifically, they could on purpose play badly in the games. An objective performance error was detected as an AD trial, and therefore yielded a self-performance of 0 on that trial, if it fulfilled specific criteria. The detection criteria for AD were specific to the current game that participants played. Supplementary Fig. 4 illustrates the criteria for the rockslide game. In essence, trials were categorised as AD trials if the participant made an obvious and extreme mistake in the game. For example, in the rockslide game, this meant responding while the ball was still visible and not hidden behind the black box that hides the target cross. Panel c in Supplementary Fig. 4 shows data from an example participant, illustrating that the ball position at response is distinctly different between normal and AD trials. For the other games, the criteria for AD trials were similar as for the rockslide game. To ensure that participants generated enough AD trials in the task that we could analyse neural correlates of AD, we took a series of measures in our experimental design. Firstly, participants were not rewarded with a bonus for how well they played the games, but for the accuracy of their ratings. Secondly, in the instructions participants received a hint that they can "try to change how they play the game to see how this affects the feedback". This ensured that participants were aware that they could alter their game performance. Note that we have several indications that AD trials are an intentional behavioural strategy rather than just accidental mistakes. Firstly, the proportion of AD trials strongly depends on the uncertainty of the feedback, that is, participants employ AD when it can help them assign credit (Fig. 3c). Secondly, participants reported using this strategy in the debrief questionnaire (Fig. 3d).

**Schedule.** We used a pre-determined schedule to manipulate the true self-performance level, other performance level and control levels per task block and phase. We used a within-participant design and each participant experienced the same schedule of self, other and control levels. Supplementary Fig. 3a, b shows the schedule of true values participants had to learn in the four task blocks. Within each task block, the true self level stayed constant and only changed between blocks. On each trial within each task block, participants played one game. Each task block was divided into distinct task phases signalled to participants (see more information above). In both Self-Other and Control-Other phases, participants played 16 trials together with new other players and new control levels. Changing the true self level from task block to task block, and the true control level and other performance level between phases allowed us to decorrelate our variables of interest. To further decorrelate them, on the 12th trial of each phase, the other's performance level changed unannounced. The trial-wise performance of the other player was drawn from a normal distribution with a mean of the true other level and an SD of 6.5. To highlight the differences between the phases, the games had different background colours. The pictures that were shown for the other player were female neutral face stimuli taken from the Chicago face database[82]. We created three schedule variants for the MRI sample to counterbalance the game order, game colours and face stimuli for the other players. For the online sample, four schedule variants were used.

## Bayesian observer models

We compared participants' behaviour to Bayesian observer models. For this, we fit three different models to the sequence of feedbacks observed by participants. These models thus provide the beliefs participants could optimally hold at any point in the task (under different assumptions about ADs, see below).

**Model types**. We fit three different learners to participants' data. Crucially, all three models were Bayesian observer models that were applied to the feedback observed by participants. The models differed, however, in how they treated AD trials. Specifically, the "Active learner" model was given accurate information about when participants performed AD trials and when they performed normal trials. The "Ignorant learner" model was a similar, Bayesian learner but did not know when an AD had occurred and instead assumed that each trial was normal, i.e., feedback on every trial (including the de facto AD trials) always reflected participants' best possible performance on the game. The third model, the "Passive learner" was again a Bayesian learner, moreover it correctly knew when an AD as opposed to a normal trial had occurred. However, the Passive learner did not learn from the AD trials; it did not use them for updating its beliefs. In other words, the model knew when AD trials occurred and knew that they were different but simply discarded them from the learning process. Full details on how the three models were implemented are given in the next section (Modelling details). By comparing the three models, it was possible to show that learning and credit assignment depended on using the difference between the feedback on the AD and normal trials in the manner of the Active learner. Supplementary Fig. 6 shows the parameter/trial-to-trial estimates from the models.

**Modelling details**. Models were fit using cmdstanr (v. 0.5.3)[83]. They were run with four chains and 10,000 iterations. Convergence was checked using the Rhat<1.1 criterion and the absence of divergent samples[84]. Iterations were increased in the case of divergent samples and until the Rhat criterion was fulfilled.

The models were fit separately to each participant, each game block and each Self-Other and Control-Other phase. Each model comprised five fitted parameters: mean and standard deviation of the self-performance, mean and standard deviation of other performance, and mean of control level. For instance, the estimated mean self-performance corresponded to the average self-performance, while the standard deviation corresponded to the variability of self-performance. On every trial, the models updated these parameters based on the feedback observed by participants. To obtain beliefs for each trial, the models were fitted repeatedly (i.e., first fitted only to data from trial 1, then data from trials 1 and 2, then from 1, 2 and 3, etc.). While the three models differed in the assumptions they held about the AD and normal trial types (see end of this section for more details), in other respects they were identical; they updated their beliefs in the same manner and all followed Bayes' rule. Before the very first feedback observation (i.e., phase trial 1), the models' priors were defined as follows:

$$\text{self}_{prior} \sim \mathcal{N}\left(\text{self}_{\text{rating t1}}, 30\right) \tag{2}$$

$$\text{other}_{prior} \sim \mathcal{N}\left(\text{other}_{\text{rating t1}}, 30\right) \tag{3}$$

$$\text{control}_{prior} \sim \mathcal{N}\left(\text{control}_{\text{rating t1}}, 0.3\right) \tag{4}$$

where $\text{self}_{prior}$, $\text{other}_{prior}$ and $\text{control}_{prior}$ are the priors for the mean parameters of self, other and control, respectively. These are modelled as Gaussians, centred around the participant's ratings on the first phase trial ($\text{self}_{\text{rating t1}}$, $\text{other}_{\text{rating t1}}$, $\text{control}_{\text{rating t1}}$) and with a wide

standard deviation (30 for self and other, 0.3 for control). The latter led to a wide initial 90% credible interval of 98.70, spanning almost the whole range of possible beliefs if the prior mean is 50. The priors for the standard deviation parameters of self and other performances were defined as follows:

$$\sigma_{\text{self}} \sim \mathcal{N}(0, 5) \tag{5}$$

$$\sigma_{\text{other}} \sim \mathcal{N}(0, 5) \tag{6}$$

where $\sigma_{\text{self}}$ and $\sigma_{\text{other}}$ are the priors for the standard deviation parameters of self and other performance (i.e., the variabilities of these performances).

The models then updated the prior parameters of self, other and control (both means and standard deviations) based on the observed feedback. The way in which this happened depended on the model type (Active, Ignorant or Passive learner). Crucially, only the Active learner model was set up with the correct assumptions about when AD trials happened and how the feedback is different on AD trials and normal trials. In the Active learner model, the priors were updated depending on whether the current trial was a normal trial or an AD trial:

$$\text{feedback(normal)} \sim$$
$$\mathcal{N}(\text{self} * \text{control} + \text{other} * (1 - \text{control}), \sqrt{\sigma_{\text{self}}^2 * \text{control}^2 + \sigma_{\text{other}}^2 * (1 - \text{control})^2}) \tag{7}$$

$$\text{feedback(AD)} \sim \mathcal{N}\left(\text{other} * (1 - \text{control}), \sqrt{\sigma_{\text{other}}^2 * (1 - \text{control})^2}\right) \tag{8}$$

where on both normal and AD trials, the feedback is assumed to be drawn from a normal distribution. In the Self-Other phase of the task in which participants were given accurate information about the true control level, *control* simply corresponded to this true control level. In the Control-Other phase of the task in which participants had been given accurate information about the true self level, *control* was estimated but the models were given the true *self* level and $\sigma_{\text{self}} = 5$. In the last few trials of both Self-Other and Control-Other phases of the task, the other player changed performance in a manner that was unannounced to participants. On those trials, the models estimated the other as a new other player to allow for flexible learning so late in the task phase. All model estimates were constrained between 0 and 100 (1 for control), similar to the possible rating range for participants.

In the Ignorant learner model, all trials (including AD trials) were treated as normal trials. This means that on every trial, the estimates of self, other and control were updated as follows (similar to Eq. 7 in the Active learner model but applied to all trials):

$$\text{feedback(all trials, normal and AD)} \sim$$
$$\mathcal{N}\left(\text{self} * \text{control} + \text{other} * (1 - \text{control}), \sqrt{\sigma_{\text{self}}^2 * \text{control}^2 + \sigma_{\text{other}}^2 * (1 - \text{control})^2}\right) \tag{9}$$

In the Passive learner model, our goal was to mimic someone who gets confused by AD trials and discards them from learning. Therefore, the Passive learner model was fit to normal trials only and AD trials were removed from the data. Crucially, this meant that the parameters about self, other and control were only updated on the remaining normal trials (and not on AD trials), as follows (same as Eq. 7 in Active learner model):

$$\text{feedback(normal)} \sim$$
$$\mathcal{N}\left(\text{self} * \text{control} + \text{other} * (1 - \text{control}), \sqrt{\sigma_{\text{self}}^2 * \text{control}^2 + \sigma_{\text{other}}^2 * (1 - \text{control})^2}\right) \tag{10}$$

**Model parameters.** We extracted the mean parameters for self, other and control for all three computational models; these are referred to as estimates throughout the manuscript. Additionally, we used features of the Active learner model to extract a trial-wise measure of uncertainty in the task. For this, we first extracted the 90% credible intervals around the self, other, and control mean estimates on every trial. Trial-wise uncertainty was then defined as the average across these individual uncertainties on each trial, according to the task phase:

$$\text{uncertainty} = \begin{cases} \frac{\text{uncertainty}_{self} + \text{uncertainty}_{other}}{2}, & \text{Self} - \text{Other phase} \\ \frac{100 * \text{uncertainty}_{control} + \text{uncertainty}_{other}}{2}, & \text{Control} - \text{Other phase} \end{cases} \tag{11}$$

where the trial-wise *uncertainty* was the average across self and other uncertainties in the Self-Other phase, and across control and other uncertainties in the Control-Other phase. For reference, Supplementary Fig. 7 shows the correlation of the extracted uncertainties of the Active learner model with the task schedule.

## Behavioural analyses

We applied a set of Bayesian regression models to the behavioural data, using the brms package (v. 2.17.0)[85] built on Stan[84]. For all non-hierarchical regressions, we ran the GLMs per participant and normalised all regressors to a mean of 0 and SD of 1. Four chains were run with 8000 iterations and delta sampling parameter of 0.8. Model fit was checked with Rhat < 1.1 and the absence of divergent samples[84]. If these criteria were not fulfilled, the number of iterations and delta were increased. Weak priors were used (normal(0,5) for parameters). We tested the resulting beta estimates for significance using ANOVAs with sample (MRI or online sample) as a between-factor, and evaluated the *p*-value of the intercept to evaluate the significance of the beta estimates (irrespective of sample). In addition, we report effect size as Cohen's *d* (*d*) for t-tests and Eta-squared ($\eta^2$) for ANOVAs. We included 95% confidence intervals for frequentist statistics, and 95% credible intervals for Bayesian statistics.

**Rating accuracy.** To test whether participants learnt from the feedback, we compared the accuracies of the estimated feedback to each of the individual ratings provided by the participants (i.e., the ratings for self, other, and control, Fig. 2d). For this, we computed participants' feedback estimation on every trial of the Self-Other and Control-Other phase as follows:

$$\begin{aligned}\text{estimated feedback} = &\text{ self rating} * \text{control rating} \\ &+ \text{other rating} * (1 - \text{control rating})\end{aligned} \tag{12}$$

where the estimated feedback was the sum of participants' beliefs about themselves (*self rating*) and the other player (*other rating*), weighted by their belief about their respective control (*control rating*). In the Self-Other phase, *control rating* was substituted with the true control level shown to participants, as this was participants' control estimate in this task phase.

Then, on every trial we extracted their rating accuracies for self, other and control ratings, and estimated feedback respectively:

$$\text{rating accuracy} = 1 - \frac{\text{rating error}}{\text{maximum possible rating error}} \tag{13}$$

with

$$\text{rating error} = |\text{rating} - \text{true level}| \tag{14}$$

$$\text{maximum rating error} = \max(100 - \text{true level}, \text{true level}) \tag{15}$$

For example, the accuracy of participants' self-ratings was computed based on the difference between the participants' self-ratings on a given trial and the participants' true self-level on the same trial. For the estimated feedback, this variable was used instead of the *rating*, and the observed feedback, averaged across all non-AD trials of the current phase, was used as the true level.

Next, we tested whether the accuracy of the individual ratings and the estimated feedback improved over time. For this, we extracted the rating accuracies on the first ("Start") and last trial (11, the last trial before the other player changed performance unexpectedly, "End") in each phase. For self and control, rating accuracies were only extracted from the Self-Other phase or Control-Other phase respectively because self and control were only learnt in one phase. For each participant, the rating accuracies were then averaged across the four task blocks. For each type of accuracy (self, other, control ratings or estimated feedback) as dependent variable, we then ran an ANOVA with *time in phase* ("Start" or "End") as within, and *sample* (MRI or online) as between-participant factors.

Finally, we examined whether the accuracy of the estimated feedback improved more over time than the ratings individually. Here, we extracted the difference of the accuracies from start to the end of a phase, and averaged the accuracy difference of the individual ratings per participant. Then, we compared the accuracy change (from start to end) of the estimated feedback and the average individual ratings using an ANOVA with *sample* as between- and *rating type* (binary, estimated feedback or individual ratings) as within-participant factor. We elaborate further on how the accuracy of the estimated feedback can be higher than the accuracies of the individual ratings in the supplementary methods (section 'Accuracies of individual ratings and estimated feedback').

**Credit assignment in the Self-Other phase.** To examine how participants assign credit in the Self-Other phase, we examined whether participants split up the signed total prediction error (tPE) into a self and other-assigned PE (sPE, oPE) according to their control (Fig. 2e). For this, we computed the trial-wise tPE in this way:

$$\text{tPE} = \text{feedback} - \text{estimated feedback} \tag{16}$$

where the tPE of a current trial is the signed difference between the observed feedback and the estimated feedback from the current trial. Then, we fit these regression models:

$$\text{rating update} \sim \text{tPE} + \text{control} + \text{tPE} * \text{control} + \text{trial number} \tag{17}$$

where *rating update* was the signed difference between the current and previous trials' rating. This was fit separately to the updates of self and other ratings of the participant, as well as the respective estimates from the Active learning model. *tPE* is the total prediction error observed on the previous trial; for the active learning model it was computed based on its estimates. *Control* is the known control level. For the regression models predicting other rating updates, the standard control estimate was substituted with the control estimate for the other player (1-control). *Trial number* was included as a control regressor indexing time within a given task phase. Only normal trials during the Self-Other phase were included because only on those trials can the tPE be split up according to the known control level. The results of this regression analysis are shown in Fig. 2e. For completeness, effects of Control on self and 1-Control on other are described in the supplementary notes (Control effect on self and other ratings).

**Active disambiguation as a behavioural strategy.** We hypothesized that participants used a specific behavioural strategy (AD) to assign credit more accurately and reduce uncertainty. For this, we first established that participants generated more AD trials in uncertain

contexts than certain contexts (Fig. 3c). We report the proportions of AD trials out of all trials per phase, averaged across participants.

Next, we validated the detection of AD trials in our task against participants' self-reports (Fig. 3d, see also Supplementary Fig. 4 for AD detection criteria and example data). Our goal here was to show that our analysis methods detect more AD trials in the data from participants who reported having performed more AD trials. For this, we extracted participants' responses to the debrief question "In the study, how many times per game did you do badly on purpose?" (see supplementary methods, Experimental design, post-task debrief questionnaire for more details). Then, we computed each participant's average number of AD trials across game blocks, and correlated this number ('# AD trials in behaviour') with the debrief responses ("Reported # AD trials") using Spearman's correlation ($\rho$). In this correlation analysis, we included participants that were otherwise excluded because of the low number of detected or reported AD trials (see information on sample selection). Therefore, in the online sample, an additional 28 participants were included so that the total online sample was $n = 64$ for this analysis. For the MRI sample, 29 participants were included in this analysis because from the total sample of $n = 34$, five participants did not fill in the debrief questionnaire.

Next, we tested whether AD reduces the trial-wise uncertainty (Fig. 3g). The following regression model was fit:

$$\text{uncertainty} \sim AD_{t-1} + AD_{t-2} + \text{trial number} + \text{other changed} + \text{phase} \tag{18}$$

where *uncertainty* is the current trial's average uncertainty. $AD_{t-1}$ and $AD_{t-2}$ are binary indicator variables for whether there was an AD one or two trials before the current trial. *Trial number* is time in the current phase, and *phase* is a binary variable for whether the current trial is in the Self-Other phase or Control-Other phase. *Other changed* is a binary variable for whether the other player's performance has changed (0 until 11th trial, 1 from 12th to 16th trial, see also "Schedule") and accounts for the uncertainty changing when the other player's performance changed unannounced during a phase.

To test the effect of AD on rating errors (Fig. 3h), we computed the sum of all rating errors on every trial (i.e., combining errors across self, other and control) and fit a regression model as follows:

$$\text{summed rating errors} =$$
$$\begin{cases} \text{self rating error} + \text{other rating error}, & \text{Self} - \text{Other phase} \\ \text{self rating error} + \text{other rating error} + 100 * \text{control rating error}, & \text{Control} - \text{Other phase} \end{cases} \tag{19}$$

$$\text{summed rating errors} \sim AD_{t-1} + AD_{t-2} + \text{trial number} + \text{phase} \tag{20}$$

where *summed rating errors* are computed separately for participants' ratings and the Active learner's estimates. The regression model was fit separately to participants' and Active learner's rating errors, with the same regressors as for the regression model predicting uncertainty. During the Control-Other phase, the Active learner's summed rating errors were based on the sum of control and other rating errors. This is because in this task phase, participants had been informed of the veridical self-performance level, and accordingly, the Active learner assumed the true self-performance level in this task phase.

**Inferring control from active disambiguation.** We hypothesized that participants inferred their controllability over the feedback from comparing normal and AD feedback, taking into account their overall self-performance. Analytically, this follows from the equations of how

the feedback is computed on normal and AD trials:

$$\text{normal feedback} = \text{Self} * \text{Control} + \text{Other} * (1 - \text{Control}) \tag{21}$$

$$\text{AD feedback} = \text{Other} * (1 - \text{Control}) \tag{22}$$

where the AD feedback is only based on the other's performance, weighted by their control. This is because on an AD trial, participants play the games so badly that on that trial $t$, their objective performance is transformed into Self($t$)=0. From these equations, it then follows:

$$\text{Control} = \frac{\text{feedback difference}}{\text{Self}} \tag{23}$$

with

$$\text{feedback difference} = \text{normal feedback} - \text{AD feedback} \tag{24}$$

Intuitively, let's assume that overall, a participant performs at a medium level in a certain game. On a normal trial, when the participant does their best in the game, the overall feedback shown is around a medium or average level. If they engage in AD, that is they deliberately do badly in the game, then the feedback they receive will be very bad. This means that the participant must have substantial control over the feedback to account for the big feedback difference. However, if they do badly on purpose and the feedback is not changed much (Fig. 4b), then the participant can infer that they have only very little control over the feedback. In short, given that the participant knows their own performance, they can change their behaviour (or performance) in the game to see how this affects the feedback, and from the difference between the feedback on the two occasions they can infer how much control they have over the feedback. To translate this into a numerical example: The participant knows that their mean Self-performance level on a normal trial is *Self* = 50, and has observed a *normal feedback* = 44 and an *AD feedback* = 4 (all on a scale from 0 to 100). From this, a control of 0.8 can be inferred because *control = (44 − 4)/50 = 0.8*.

We used regression models to test if participants used this intuition to learn about their control over the feedback (Fig. 4c, d). We ran these regression models only on switch trials during the Control-Other phase because this is when participants can infer their control if they are comparing feedback on an AD and an adjacent (prior or subsequent) normal trial. Switch trials were defined as trials that are either a normal trial following an AD trial, or vice-versa. Since some participants had a low number of switch trials, hierarchical regression models were used and run across MRI and online samples. Models were run with weak priors (normal(0,5) for parameters), 4 chains, 6000 iterations and a delta sampling parameter of 0.9. Model convergence was checked as before. All regressors were normalised to a mean of 0 and SD of 1 across participants. The following hierarchical regression model was fit:

$$\begin{aligned} \text{control} \sim\ & \text{feedback difference} + \text{inverse self} + \text{feedback difference} \\ & * \text{inverse self} + \text{previous control} + (\text{feedback difference} + \text{inverse self} \\ & + \text{feedback difference} * \text{inverse self} + \text{previous control} | \text{ID}) \end{aligned} \tag{25}$$

where *control* is the next trial's control rating. *Feedback difference* is the difference between normal and AD feedback, as observed on the current and previous trial. *Inverse self* is ln((current trial's self rating)$^{-1}$), and *previous control* is the current trial's control rating. The hierarchical regression model was separately fit to participants' control ratings, and to the respective control estimates provided by the three Bayesian models (active, ignorant, and passive). In the Supplementary Information, we consider the ways in which participants learnt about

self and others following a similar logic to infer these variables from AD and normal feedback (Supplementary Methods, section Inferring self and other from active disambiguation, Supplementary Fig. 9). We examined factors driving AD trials and relative control vs other updating in the supplementary information (Supplementary Methods, section Effects driving AD trials and learning in the Control-Other phase, Supplementary Fig. 8).

### Imaging data acquisition and preprocessing

Imaging data were acquired with a Siemens Prisma 3 T MRI using a 32-channel head-coil. T1 weighted structural scans were obtained with repetition time (TR) = 1900 ms, echo time (TE) = 3.96 ms and $1 \times 1 \times 1$ mm voxel size. Functional images were acquired using a multiband T2*-weighted echo planar imaging sequence with acceleration factor of two, and with an oblique angle of 30° to the posterior commissure (PC)– anterior commissure (AC) line to reduce signal dropout in orbitofrontal regions[86]. Other acquisition parameters included $2.4 \times 2.4 \times 2.4$ mm voxel size, TE = 30 ms, TR = 1,360 ms, 60° flip angle, a 240 mm field of view and 66 slices per volume. For each session, a fieldmap (voxel size = $2.4 \times 2.4 \times 2.4$ mm, TR = 482.0 ms, TE1 = 4.92 ms, TE2 = 7.38 ms) was acquired to reduce spatial distortions. Bias correction was applied directly to the scan. We took physiological recordings during the task to measure pulse, breathing and skin conductance of participants.

Imaging data was analysed using FMRIB's Software Library (FSL)[87]. Preprocessing stages included motion correction, correction for spatial distortion by applying the fieldmap, brain extraction, spatial smoothing (using a full-width at half-maximum of 3 mm), and high-pass temporal filtering. Functional images were first co-registered to an individuals' T1 structural image and then nonlinearly registered to the Montreal Neurological Institute (MNI) template using 12 degrees of freedom.

### fMRI whole brain analysis

First-level analysis was carried out using FSL FEAT (version 6.00). First, data were pre-whitened with FSL FILM to account for temporal auto-correlations. Temporal derivatives, standard motion correction parameters and physiological noise regressors (using the FSL PNM toolbox[88]) were included in the model. Results were obtained with automatic outlier-deweighting, and using FSL FLAME1 with cluster-correction threshold of $z > 3.1$ and $p < 0.05$[89].

We used a single GLM for whole-brain analysis, and included all three phases of a trial (rating, action, outcome) in the GLM. Constant and parametric regressors were time-locked to the onsets of the relevant trial event, and had the duration of the respective event. All parametric regressors were normalized to a mean of zero and standard deviation of 1. All regressors were convolved with the double-gamma HRF in FEAT. Correlations between regressors are shown in Supplementary Fig. 12.

**Rating phase.** The rating phase began at the time the rating screen appeared and lasted until the participant finished all ratings. To account for differences between Self-Other and Control-Other phases, the regressors related to control and self ratings were defined separately for each task phase. The regressors for the other rating were defined across both task phases because participants needed to learn about the other's performance level in a similar manner in both task phases.

- Control rating related regressors, time locked to rating screen onset:

  - constant for control rating
  - parametric control rating
- Self rating related regressors, time locked to self rating onset (in the Self-Other phase, when the true control was shown, this was

time locked to the rating screen onset + 1.23 sec, the mean RT for making a rating (computed from the Self-only trials)):

  - constant for self rating
  - parametric self rating
- Other rating related regressors, time locked to other rating onset

  - constant for other rating
  - parametric other rating

Parametric regressors of no interest were included for the reaction times (RT) of each rating (separate by task phase, control rating RT only for Control-Other phase).

**Action phase.** The action phase began when participants played the games and lasted until the game finished. Regressors were defined across Self-Other and Control-Other phases because the action phase did not differ between task phases.
- game-type specific constants (4 in total, 1 for each of the 4 games)
- active disambiguation (binary, prior to normalizing: 1 for AD trial, 0 for normal trial)
- parametric uncertainty (extracted from Active learning model)
- switch trial (binary, prior to normalizing: 1 for switch trial, 0 for stay trial)

In addition, a parametric regressor of no interest was included for the RT of the action phase to control for RT differences on AD trials. Another parametric regressor of no interest was included for the trial number across the game block to control for the effect of trial number (as a measure of time in block) on uncertainty.

**Outcome phase.** The outcome phase started when the feedback appeared on the screen and lasted until the participant pressed the 'Continue' button. Regressors were defined depending on the task phase because we hypothesized different learning mechanisms per task phase.
- Constant for outcome phase
- Parametric regressors in Self-Only phase:

  - sPE (on normal trials only)
  - oPE (on normal trials only)
  - signed uncertainty change (from the current to the next trial)
- Parametric regressors in the Control-Other phase:

  - Feedback difference (defined as normal – AD feedback, on switch trials only)
  - signed uncertainty change (from the current to the next trial)

We included regressors of no interest in the outcome phase. To account for motor-related brain activity, we included a constant time-locked to when participants pressed the 'Continue' button and with a duration of 1 s. For each task phase separately, we included binary regressors for switch trial and active disambiguation (both regressors were defined similarly to the corresponding regressors in the action phase).

### fMRI time course analysis on ROIs

**ROI selection and time series extraction.** Similar to previous work[3,17,38,90,91], spherical ROIs with a radius of three voxels (5 mm) were constructed based on significant activation clusters in the whole-brain analysis. ROIs were centred on the peak voxels of the effect of uncertainty at the time of action, and where uncertainty-related clusters overlapped with the effect of AD at the time of action. This was the case for three brain regions, from which we discarded one for our analyses

because it was in the occipital cortex. The two remaining ROIs were in the left SMG (MNI: $x = -62$, $y = -24$, $z = 36$) and left area 7 in the superior parietal lobule (MNI: x = −12, y = −64, z = 64). To ensure that only brain was included in the ROIs, we centred them two voxels more medially for the left SMG, and one voxel more inferiorly for the left area 7.

Spherical ROI masks were transformed from standard space to each participant's structural space using a standard-to-structural warp field, and from structural to functional space with a structural-to-functional affine matrix. The resulting transformed masks were thresholded and binarized. Then, motion and physiological noise were partialled out from the filtered functional data. The time series of each ROI was extracted from the residual filtered functional data, and averaged across voxels. Timeseries data were normalised, and up-sampled 20 times using cubic spline interpolation. The resulting data were epoched into 17 s windows of 2 s before and 15 s after each event of interest (described in GLM section below). We applied ordinary least squares GLMs to each time point, resulting in a time course of beta weights for each regressor. All parametric regressors were normalized (mean of 0 and a standard deviation of 1). In all of the following ROI-GLMs, we included constant regressors in addition to the described parametric regressors below.

**Time course GLMs (ROI-GLMs).** We hypothesized that these regions which are involved in processing uncertainty and AD during action generation might also be involved in processing the feedback outcomes. Therefore, our time course analyses were independent of the whole-brain analyses based on which the ROIs were defined. Correlations between regressors for all ROI-GLMs are shown in Supplementary Fig. 13.

Our overall hypothesis is that the brain represents the inferred control belief based on the currently observed feedback difference and the prior inverse self-estimate (Fig. 6a). First, we tested whether SMG and area 7 represent the inferred control estimate and its update. Therefore, we ran the following ROI-GLM1, time-locked to the outcome of a current trial (Fig. 6b):

$$\text{BOLD} \sim \text{control} + \text{control update} \tag{26}$$

where *control* is the control rating from the next trial and is therefore a measure of what control level participants infer from the currently observed feedback. *Control update* is the difference between the next trial's and the current trial's control rating, and is therefore a measure of how much participants change their control rating based on the currently observed outcomes. All trials in the Control-Other phase are included in this ROI-GLM1.

The goal of our next GLM was to examine if ROI activity correlates with the feedback difference and inverse self, which are the components necessary for control inference. Here, we examined two competing hypotheses (see Fig. 6c-d) for how our ROIs might represent the feedback difference at the time of outcome. Firstly, we hypothesized that neural activity might represent the feedback difference as the comparison between normal and AD feedback to infer control (H1, Fig. 6c). This means that neural activity does not depend on whether the currently observed outcome is from an AD trial and the previous was from a normal trial or vice versa. Alternatively, neural activity might track feedback differences between outcomes in a simpler comparison of current trial versus previous trial. In this case, neural activity depends only on the temporal order of current and previously seen feedback, and not on the identity of the trial as normal or AD. We tested these two hypotheses in a single GLM time-locked to the outcome phase (ROI-GLM2):

$$\text{BOLD} \sim \text{feedback diff}_{\text{AD to normal}} + \text{feedback diff}_{\text{normal to AD}} + \text{inverse self} + \text{AD} \tag{27}$$

with

$$\text{feedback diff}_{\text{AD to normal}} = \text{normal feedback}_{\text{current trial}} - \text{AD feedback}_{\text{previous trial}} \tag{28}$$

$$\text{feedback diff}_{\text{normal to AD}} = \text{normal feedback}_{\text{previous trial}} - \text{AD feedback}_{\text{current trial}} \tag{29}$$

Analogously to the behavioural analysis, only switch trials in the Control-Other phase are included in this analysis because these are the trials when participants can observe a feedback difference (between normal and AD trials), which is the critical variable to infer controllability. To test H1 and H2 in one GLM, we defined two regressors for feedback difference which are both computed as the difference of normal minus AD feedback. *Feedback diff*$_{AD\ to\ normal}$ is the feedback difference only on switch trials where the previous trial was AD and the current is normal. In contrast, *feedback diff*$_{normal\ to\ AD}$ is only defined on trials where the previous trial was normal and the current is AD. Defining the feedback difference separately by the type of switch trial allowed us to pit the two hypotheses H1 and H2 against each other (bar plot in Fig. 6e). This is because the average across *feedback diff*$_{AD\ to\ normal}$ and *feedback diff*$_{normal\ to\ AD}$ is the overall effect in favour of H1, while the average across *feedback diff*$_{AD\ to\ normal}$ and the sign-flipped *feedback diff*$_{normal\ to\ AD}$ is the overall effect in favour of H2. By only including switch trials into this analysis, we also ensured that any feedback difference related effects cannot be driven by the feedback difference being higher on switch than stay trials. *Inverse self* is the inverse of the current trial's self rating from the current trial (i.e., before observing the feedback, computed similarly as for the behavioural analysis) to measure what participants' prior for their own performance is. *AD* was included as a regressor of no interest to control for other features that might impact the feedback difference between outcomes.

Next, we investigated the neural correlates of how participants split up the tPE into sPE and oPE according to their known control level in the Self-Other phase. For this, we ran three GLMs. Analogously to the behavioural analysis, we analysed only normal trials during the Self-Other phase and computed sPE and oPE in the same way as for the behaviour. First, we tested whether SMG and area 7 represented the sPE and oPE at the time of outcome. Due to the high correlation between sPE and oPE ($r = 0.55$), we tested their effects in two separate regressions (ROI-GLMs 3 and 4), and jointly within a single regression model (ROI-GLM5) and ensured that similar results were found in all cases.

$$\text{BOLD} \sim \text{sPE} \tag{30}$$

$$\text{BOLD} \sim \text{oPE} \tag{31}$$

$$\text{BOLD} \sim \text{sPE} + \text{oPE} \tag{32}$$

We also tested an alternative approach for how neural activity might represent the splitting up of the tPE in the outcome phase. In ROI-GLM6, we tested whether there are representations of the tPE and control level, which are used to construct the sPE and oPE.

$$\text{BOLD} \sim \text{tPE} + \text{control} \tag{33}$$

where *control* is the true control level and *tPE* is the total prediction error. Analogously to our behavioural analysis, in ROI-GLMs 3 to 6, we analysed only normal trials in the Self-Other phase because this is when the feedback reflects both self and other, and therefore generates a tPE that can be split up into an sPE and oPE in relation to the known control level.

Finally, we tested whether neural activity correlates with feedback in our task, independent of any credit assignment or inference process. A key region that has been implicated in tracking reward is the ventral striatum. To test whether the ventral striatum signals feedback in our task, we extracted activity from an anatomically defined bilateral mask of the nucleus accumbens[91]. We ran the following regression model time locked to the outcome phase (ROI-GLM7):

$$BOLD \sim feedback + feedback\ change \qquad (34)$$

where *feedback* is the parametric feedback observed on the current trial, and *feedback change* is the signed difference between the feedback observed on the current compared to the previous trial. To ensure that any effects related to these two regressors are independent of AD, we tested this only on normal trials following normal trials, across Self-Other and Control-Other phases. In the supplementary materials, we visualise the effects of uncertainty and AD at the time of action using ROI-GLMs (Supplementary Information, fMRI time course analysis on ROIs, Supplementary Fig. 18 and 19).

**Leave-one-out (LOO) procedure.** To test for significance and avoid temporal selection biases in our time course analyses, we used a similar approach to previous studies[17,38,90,91]. For each participant ($n = 31$), we computed the average time course across the remaining participants ($n = 30$). Next, we computed the time point between 4 and 10 s (unless otherwise specified) when this average time course peaks, and extracted the beta value for the excluded participant at this peak time. This was repeated for all participants, resulting in a set of individual peak values that were determined in an unbiased way. These peak values were tested with a two-tailed one-sample t-test against zero.

**Reporting summary**
Further information on research design is available in the Nature Portfolio Reporting Summary linked to this article.

## Data availability
The raw behavioural and preprocessed fMRI data generated in this study have been deposited on Github (https://doi.org/10.5281/zenodo.17602027)[92]. The whole-brain maps of the fMRI data (shown in Fig. 5) can also be found on neurovault (accession code: https://identifiers.org/neurovault.collection:21965).

## Code availability
The code used in this study is available on Github (https://doi.org/10.5281/zenodo.17602027)[92].

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

## Acknowledgements

This work was supported by the Medical Research Council (MRC; MR/N013468/1 to L.S.; Skills Development Fellowship, MR/N014448/1 to J.S.; Career Development Awards, MR/Y010477/1 to M.K.W., MR/T031344/1 to J.X.O; Project Grant, MR/L019639/1 to J.X.O.), Biotechnology and Biological Sciences Research Council (BBSRC; Discovery Fellowship, BB/V004999/1 to J.S.; sLOLA Grant, BB/W003392/1 to J.X.O. and M.F.S.R.), Wellcome Trust (221794/Z/20/Z to MFSR), UK Research and Innovation (UKRI; under the UK government's Horizon Europe funding guarantee UKRI336 to M.K.W. [selected as ERC Starting Grant, EC reference 1011159]), Institut National de la Santé et de la Recherche Médicale (Inserm to J.S. and N.K.) and European Research Council (ERC; grant FORAGINGCORTEX [project number 101076247] to N.K.; EMOBB [101162047] to J.S.). The Wellcome Centre for Integrative Neuroimaging is supported by core funding from the Wellcome Trust (203139/Z/16/Z and 203139/A/16/Z). Views and opinions expressed are those of the authors only and do not necessarily reflect those of the European Union or the European Research Council Executive Agency. Neither the European Union nor the granting authority can be held responsible for them. For the purpose of Open Access, the authors have applied a CC BY public copyright licence to any Author Accepted Manuscript (AAM) version arising from this submission. We thank T. Behrens and Q. Huys for helpful discussions of this work.

## Author contributions

L.S., M.F.S.R. and J.S. conceived and designed the experiment. L.S., H.T. and J.S. programmed the experiment. L.S. collected the data. L.S. and J.S. constructed the computational models. L.S., J.X.O., N.K., M.K.W., M.F.S.R. and J.S. conceived the data analysis. L.S. conducted the data analysis. L.S., M.F.S.R. and J.S. wrote the manuscript. All authors provided expertise and feedback on the write-up. J.S. and M.F.S.R. supervised the research project.

## Competing interests

The authors declare no competing interests.
