## [Transparent Peer Review file · Nature Communications]

Active disambiguation guides inferring controllability and cause in social interactions

Corresponding Author: Dr Lisa Spiering

Version 0:

Reviewer comments:

Reviewer #1

(Remarks to the Author)

This manuscript addresses a longstanding issue of interest in psychology and cognitive neuroscience, and leverages recent advances in the computational understanding and neural study of controllability inference, to study a specific instance of this problem, namely, that of inferring agency in a collaborative game between two people.

Specifically, the study addresses how individuals assign outcome/feedback to their own performance versus that of others in social collaborative contexts, with varying levels of controllability. Controllability, in this study, is defined as how much an outcome is influenced by an individual's actions compared with that of another person, aiming to address the mechanisms by which people infer their control over outcomes in collaborative scenarios. The study makes significant contributions by introducing a clever novel behavioural task, combining computational modelling and fMRI. The authors identify active disambiguation (AD) as an exploratory mechanism for resolving uncertainty in social contexts, and demonstrate the importance of the supramarginal gyrus as a central brain region for agency-related processes during uncertainty. The use of Bayesian models to link the use of AD to intervention-based causal learning and exploration for information acquisition using two alternate behaviour models is interesting and explained well. The study has findings that have implications for understanding the mechanisms of how humans infer agency in different collaborative social interactions, and relevance to conditions such as depression where the estimation of control assignment is impaired.

The results will be of interest to the broader social cognitive neuroscience community, particularly those with affinity for advanced computational approaches to studying control.

We did find the paper a bit dense, and complex, and therefore, hard to evaluate. Many different analyses are presented that it is hard to keep track. The reader might be helped by some sort of overview or bullet point summary of the key results at the start of the results section.

There are a few specific things that we have difficulty wrapping our heads around.

First, to what extent does the current set-up isolate the inference of controllability specifically, instead of uncertainty or unpredictability more generally? It is clear that people updated control ratings as a function of uncertainty-triggered ADs, but does this mean they compute outcome controllability in the strict sense, or do they instead track outcome predictability more generally, and engage in AD to increase predictability when it drops? Another way to ask this question is: What are the (internal vs external) determinants of uncertainty? Only the 'other's' performance level fluctuated to disentangle the variables of interest. Does this mean that uncertainty peaked disproportionately when the others performance level changed? Given that uncertainties were averaged in the models across 'control' and 'other', it is not clear whether uncertainty was driven primarily by uncertainty about control or about the other-performance. And whether subjects were tracking separate estimates of self-, other-, and control-related uncertainties at all? From Figure 2D it appears that they do track these components, albeit less precisely than overall uncertainty. Is there a way to quantify the contribution of 'control' uncertainty versus the uncertainty over other-performance to ADs, and to updating of subjective control ratings? And is there a way to quantify the relative contribution of ADs (which were greatest when uncertainty was greatest, so we assume upon a reversal of other's performance level) to estimating the other's performance level rather than control?

Can the timecourses of self-, other-, control-related uncertainty, as well as averaged uncertainty and ADs be visualized?

Second, and this is related to the previous question, while it is recognized that the process of inferring agency under study here represents an instance of the controllability inference problem that has recently attracted renewed interest, the similarities and differences between the algorithms and neural activity patterns observed here and those observed in those recent previous studies are not addressed explicitly. Specifically to what degree does the computational problem of inferring agency as operationalized here correspond algorithmically to the same problem as that of inferring environmental controllability by comparing actor and spectator models of the world? Previous fMRI, lesion and brain stimulation work has highlighted a role for medial frontal cortex (pgACC and IPFC in particular) in controllability inference. Is the SMG doing what the pgACC is doing for controllability inference when the algorithmic problem is framed as a social problem?

Other comments:

The task description could be clearer right from the start. For example, it describes: “three hidden latent causes: their own performance, another fictive player’s performance, and their own control over the feedback (Figure 1A).” It might help to make more explicit right away that these refer to (i) own action efficacy, (ii) other’s action efficacy and (iii) the degree to which own, or other’s efficacy matters for the outcome. It would also be good to clarify whether participants are told that outcomes always depend on (own or others) actions – in other words, that environmental controllability was fixed.

Please specify how AD errors were defined in the main text (not just methods), as this is key for understanding the main results section.

We missed a clear rationale for testing (and clear conclusions based on the observation of) reward signal in the ventral striatum.

Please clarify whether participants were aware of the fact that other-performance level could change during a game, but control level could not

Participants were asked to report whether they generated an AD trial. What was the exact question they were asked? Although they do report that participants generated an AD trial when uncertainty was high, the participants were also primed in a way by the explicit instructions to explore alternative action strategies to best estimate their control levels. To strengthen the conclusion that AD behaviour is a generalizable phenomenon that does not depend on demand characteristics, it would be good, in future studies, to add a control block without such instructions, allowing isolation of this effect of instruction/demands characteristics.

Why did they decide to have different rating scales for self and other ratings (0-100) as compared to the control rating scale (0-1). Would participants then perceive control to be more of a binary concept as compared to performance which could vary?

How consistent were the reported effects across the 4 different games/blocks completed? For example, did participants tendency to adopt the AD strategy increase over time? Was there any evidence of meta-learning?

Fig 3B – does this refer to the entire sample (MRI + online)?

Typing error in line 555 – SMS instead of SMG

Please clarify this statement, we felt it was unclear: Line 154 – “In the following, we will refer to control from the perspective of the participant (e.g. 20% control means the participant and the other control 20% and 80% of the feedback respectively)”

Just out of curiosity, to what extent do the results and conclusions revealed by the current study generalize to (more naturalistic) collaborative gaming set-ups with more than 2 players, in which multiple sources that can influence the outcomes – similar to real-world scenarios with team/group work with more than a pair of interactions.

Reviewer #2

(Remarks to the Author)

Reviewer #3

(Remarks to the Author)

In this paper, Spiering and colleagues describe a combination of behavioural experiments, computational modelling and neuroimaging aimed at unravelling the processes involved in tracking responsibility during collaborative interactions. To this end, they design a novel task that involves people playing simple games along with a ‘partner’, while they receive ambiguous feedback that signals a mixture of their performance and the performance of the partner. Critically, participants are explicitly tasked with estimating the performance level of each player, and the relative contribution each player makes to the feedback that is received.

First, the authors find a characteristic form of behaviour which they call 'active disambiguation' – where participants seemingly make active errors in the task, but only when such errors would be helpful to disaggregate between self and other performance and control. Moreover, they show that important behavioural signatures from the task can only be captured by a variant of a Bayesian learning model that learns from these active disambiguation episodes appropriately (i.e., which represents them as active mistakes). This allows participants to learn using the difference between the typical feedback, and the feedback that arrives when agents 'hypothesis test' via the errors they make.

Finally, in an fMRI experiment, the authors find a common coding of (model-based) uncertainty and these 'active disambiguation' behaviours in SMG. They show, moreover, that SMG contains a range of signals needed for inferences about responsibility – including estimates of control, the feedback difference and prediction errors about self and other performance.

In general, I thought the paradigm was rich and the paper was technically impressive – with a range of clearly thought out analyses mapping out the component parts of the computational model implied by the behavioural data onto activity in the SMG. However, I was still left with a general sense that – despite these virtues – the theoretical claims made from the task were slight exaggerated. In particular, it was not so obvious to me that these results tell us much about 'control' in the context of 'collaboration'. This may mean significant reframing of the paper is necessary, particular if it is to be situated within the way these concepts are typically studied and understood by other psychologists and neuroscientists.

Major comments

1) As noted above, it is clear that this manuscript is impressive in its technical richness. But from the outset, I had some general concerns about how far these impressive methods can be said to be targeting the cognitive and mental functions implied in the title, introduction and discussion.

First, the paper is largely framed as being about how we determine what we can and cannot 'control' in joint action settings. In psychology, such joint actions have typically been studied in contexts where people collaboratively coordinate to control the same object (e.g., Knoblich and Sebanz's work on controlling the spatiotemporal trajectory of a single object together). Here though, the outcome being controlled is rather more nebulous – as it relates to the 'feedback' that occurs at the end of the trial. Naturally, participants in the task are encouraged to think about the task by asking 'how much did you *control* that feedback?' but any equally plausible framing could be 'how much did that feedback *relate* to your action?' or 'how much were you *responsible* for the feedback that appeared on screen?'

I raise these issues not to try and split terminological hairs, but to drill into the connection between the processes studied here and the broader literature on 'control' and 'controllability'. For instance, many theoretical models (including those that implicate neural regions like SMG) suggest that personal and joint feelings of control are generated by monitoring sensorimotor processes that are engaged as we create actions and outcomes ensue – and it is very hard to see how such processes could be deployed in the very clever but rather artificial design used here. And this may become relevant to some of the connections the authors make to sense of agency, schizophrenia etc. where a sensorimotor monitoring deficit seems implicated.

It may be that the cognitive and neural mechanisms described here tell us more about cognitions of 'responsibility' rather than feelings of 'control' – which may be important for adjusting the paper's framing and/or considering which literatures the paper does and does not seek to connect to. I would be interested to know what the authors think.

2) I have similar concerns about framing this experimental task as a study of 'collaboration'. Most real and experimental studies of collaborative decision making necessarily involve two or more agents working towards the same goal (e.g., work from Bahrami and colleagues examining how two people judge the same stimulus, and arrive at an accurate answer together). However, in this case the participant's task is explicitly *not* to maximise joint outcomes in the task, but to make accurate ratings about the self and other. I feel this somewhat weakens how far this study tells us much about how we 'collaborate' and 'cooperate' with others. For instance, the paper claims in the abstract that that 'people engage in specific patterns of behaviour' in ambiguous social settings – and this is mediated through a particular set of brain regions. But it seems this may be a little overstated, since such 'active disambiguation' might only occur when we are given the task 'work out what you are responsible for' rather than this being a common feature of our collaborative joint interactions. It strikes me it would be stronger if these behavioural, computational or neural signatures could also be generated in a variant of the task where participants have an incentive to work towards the joint outcome (i.e., maximizing feedback outcomes). I would be interested to know if the authors think these effects tell us much about 'collaboration' or whether they tell us more straightforwardly about mechanisms we use to assess responsibility when this is the task we have been given.

3) In the ROI-based analyses, I did not understand the rationale behind limiting analyses to small spheres drawn around the peak effects in the whole brain analysis rather than e.g., all the voxels active in the whole-brain cluster, or those voxels passing some other activation threshold. Are any effects changed if the regions of interest are drawn less narrowly?

Minor comments

4) Page 19 lists N = 66 online participants in the final sample, while p.4 lists N = 36. Which is correct?

5) There are regular references to the Bayesian learner models being 'optimal'. Then, in other cases, the 'passive' and 'ignorant' learner models are described as being 'suboptimal'. I know what the authors mean when they talk about Bayes optimality, but I worry that this has the potential confuse a reader given that (from a Bayesian perspective) even the

suboptimal learners are optimal, given their architecture! For instance, the paper contains sentences like "...this suggests that participants, like a Bayesian optimal observer, attribute social PEs in accordance with their controllability" (p.7). This is potentially confusing, as to a general reader this may imply that the observer is optimal *because* it is doing this kind of controllability weighting. When in practice the authors evaluate other models that are equally (Bayes) optimal which don't show this signature.

It didn't strike me that any of the paper's conclusions really rest on human behaviour being Bayes optimal in any case, so the paper may be more easily digested if the authors simply speak of 'Bayesian learners' rather than 'optimal Bayesian learners' throughout. e.g. sentences like that above would become "this suggests that participants attribute social PEs in accordance with controllability, just as the winning model does".

6) In some ways, this work reminds me of work by Pescetelli & Yeung (2021) on the advice-taking and trust. Here the authors show that agents sometimes seek advice from advisors even when they are very confident in their answers – which may seem suboptimal, but in fact allows agents to learn about the quality of the advice they are being given (as they know they know the correct answer). It seems an essential feature of the present task is to disrupt possible introspective metacognitive feelings by distorting the feedback that participants receive so it is not on a straightforwardly objective scale. To what extent do the authors see any connection between the kind of 'pointless' advice taking behaviour described by Pescetelli & Yeung (2021) and the 'purposive errors' described in their work?

7) Related to the last point – do the authors think it would be possible to solve the present task without active information seeking 'errors' if introspection was not actively disrupted? I.e., can metacognitive appraisals of personal correctness play the same role of disaggregating responsibility in settings where these signals are not actively corrupted?

8) The authors may consider making reference to work by Wen Wen on the active sensing of control difference (e.g., Wen et al (2020), iScience). This work has already established that agents actively manipulate the actions they perform to exaggerate the differences between (and thus discriminate between) the things they can and cannot control – analogously to what is achieved in the present work (though not in a social setting).

Version 1:

Reviewer comments:

Reviewer #1

(Remarks to the Author)

The authors have gone out of their way to address our comments in great detail and have addressed some of our comments. For example, we agree that the term action efficacy might create further confusion.

There is a lot to like about this paper, and the demonstration that participants recruit an Active Disambiguation strategy and the supramarginal gyrus to resolve uncertainty is valuable and the evidence for this is solid.

However, we remain puzzled by key parts of the revised paper and the rebuttal.

The first point we struggle with is the assertion that feedback can be highly predictable, even if there is great uncertainty about the causes of the feedback. We can see from Fig 2d that estimated feedback/rating accuracy increases over the course of a phase. Accordingly, feedback predictability seems to increase in a manner that is proportional to the reduction of uncertainty about who is in charge, over the course of a phase. In line with this, the new paragraph regarding distinguishing between uncertainty and unpredictability is hard to parse. If something is uncertain, it is by definition also unpredictable. The key thing that I think the authors want to highlight is the nature of the thing that is uncertain/unpredictable. I think the authors would like to state: "In our task, we distinguish between uncertainty about (=unpredictability of) which agent is causing the outcomes and uncertainty about (= unpredictability of) the outcomes themselves (i.e., about how good the joint performance is)."

Importantly, I am not sure there is evidence for this distinction. Indeed, the additional supplementary analyses presented in the rebuttal (point 1, page 11) demonstrate that ADs are not uniquely predicted by uncertainty about control, but also by uncertainty about other-performance. This is not surprising, because participants exhibit high AD rates in the first self-other phase, in which the level of control is instructed. A key implication of this observation is that the conclusion that 'active disambiguation enables inference of controllability' does not hold in the unconditional/generalized form that is suggested by this paper. What the paper shows is that AD enables inference of controllability if and only if controllability is the thing that is ambiguous, as is the case in the third phase of the current experiment. Conversely, if self- and/or other-performance is the thing that is ambiguous, as is the case in the first phase of the current experiment, then ADs are triggered by uncertainty about performance (about the feedback itself). In fact, the data from the first phase might well be analyzed to establish that ADs predict belief updates about e.g. other-performance in the 1st phase, just like they predict belief updates about controllability in the 3rd phase. By analogy, the conclusion that the 'SMG carries neural signals for learning controllability' is likely also invalid in its generalized/unconditional form, as demonstrated by the analyses of data from the first phase of the current experiment, in which the SGM tracks a learning signal that is not used for controllability inference (because the level of control is already instructed) but instead for inferring the quality of own or other's performance levels.

We also note that the more conditional inference that AD enables controllability inference in the third phase where controllability is the thing that is ambiguous seems to depend on the additional analysis reported in the rebuttal and the supplement, showing that AD predicts the difference between uncertainty about control and that about other-performance.

Based on these observations, we recommend that the authors revise their paper to abstain from concluding a key role for AD and SMG in controllability inference specifically. Instead, we suggest that the authors highlight that, based on the present data/results, they can conclude that people recruit an active disambiguation strategy in proportion to uncertainty (I assume also in proportion to tPE?) and associated uncertainty- (and tPE-?)related activity in the SMG, for resolving uncertainty. Crucially, this AD strategy and associated SMG activity seem to be recruited not only for resolving uncertainty about who is in charge (controllability), but also for resolving uncertainty about the quality of (e.g. joint, other, self) performance, and probably also for other types of uncertainty depending on the task at hand. The paper seems to capture a more general uncertainty resolution construct than the one that is implied by the current version of the paper. Note that this still represents a very valuable contribution.

Regarding our second major comment, we did not fully understand the speculation about the putative role of exploratory actions to infer control in the neural activity revealed by the current study vs Ligneul et al. In that prior work (which highlights a role for medial frontal cortex), controllability inference is equally dependent on exploratory, 'counterfactual' actions (see also Ligneul, 2021). Indeed, without such exploration, spectator and actor predictions are perfectly correlated and controllability cannot be inferred. What is the essential difference then between the computations that are studied here, activating SMG, and those non-social computations?

Minor comments:

The figure of the evolution of the various uncertainties across trials is insightful. We suggest to include this in the main paper, and to consider including a line for ADs.

Was the paper pre-registered? The task instructions suggest that the project was set up to assess effects of mood. If such analyses performed (and perhaps reported separately), then perhaps report them?

The rebuttal highlighted that 30 out of 69 online participants never made any ADs, and that, for the new, larger online study, again, only 147 of the 352 participants ended up making ADs. I recommend to highlight this observation that around half of the group makes ADs explicitly in the results (not just methods) section, and to address the implication of this finding in the discussion section of the paper.

There is a lot of redundancy in the figure legends and the text in the results section. This is a matter of taste, but I suspect the legends can be compressed to increase reading load/efficiency.

Reviewer #2

(Remarks to the Author)

Reviewer #3

(Remarks to the Author)

I am grateful to the authors for their work on this revision. In general, I think it is clear that the authors have done a significant amount of work to improve the manuscript, both in response to my queries and those raised in other reviews.

That said, I remain concerned about the unhedged claim that this is a study of 'human collaboration' - which I don't think the reviewers have appropriately addressed.

In their reply to me, the authors have explained very convincingly how this kind of 'social credit assignment' is likely to be an important part of enabling genuinely collaborative behaviours (e.g., I find the example of the student getting negative feedback and needing to diagnose the cause very compelling).

But I don't think it follows that just because a process is important for a behaviour in some settings, that studies of that process are therefore studies of that behaviour (e.g., spatial navigation is an important part of buying groceries in a store, but it doesn't follow that all studies of core navigation mechanisms are studies of buying groceries).

I agree with the authors that their social cover story makes it very likely that agents are engaged in some kind of social cognition and causal inference. But as they note, the cover story is not telling participants 'you will be working with each other to achieve a joint goal' (i.e. collaboration). The cover story is that of being a 'team captain' and estimating the performance of different players to form a team. As far as I can see, the only element of the task design that approaches collaboration is the fact that participants (believe they are) performing the same trial of the same game with another participant. But I don't think this kind of 'action in parallel' is strong enough to constitute collaboration (e.g., students sitting an exam in the same hall are solving the same problems at the same time, but that does not mean they are collaborating with each other).

These conceptual issues have a long history in both the cognitive science and philosophy of joint action - where a key defining feature of collaboration is usually taken to be the fact that agents hold a common collaborative intention, and act towards the same end. This is not the case anywhere in this task, where the task is explicitly to try and do causal inference over a feedback signal that can belong to you or somebody else.

As per my original review, I think the study here is very clever, the approach technically sophisticated and the statistical results are clear and persuasive. I also agree that this kind of social causal inference is an important element of genuine collaborative cognition when it does occur. But I don't see that the contributions of this paper *require* over-egging of the message to say this *is* a study of collaboration. In my view, it would suffice (and still be extremely interesting) to say that this is a study of 'social credit assignment', and this is an important process in some real life situations where the causal influence of different agents on an outcome is (like collaboration towards a joint goal). But I would worry about misleading readers, or adding 'noise' to the literature, if this study is literally titled as a study of 'human collaboration' - when it plainly isn't.

I think mitigating against this worry could be done with relatively simple changes to the paper (e.g., by not calling it a study of 'human collaboration' in the title, and by describing the process of social credit assignment or causal inference properly and how it is *relevant to* collaborative examples, rather than describing the task here as 'collaboration' or 'collaborative' per se). But such simple changes would be important for situating the paper in the literature appropriately, and giving readers a clear sense of what the paper contributes (and what it does not). I say all this, ultimately, as an enthusiast for what the paper does contribute to a range of important timely issues - and I hope that authors will take this suggestion in that spirit.

Version 2:

Reviewer comments:

Reviewer #1

(Remarks to the Author)

My residual comments have been addressed very well in the second revision of this impressive paper. The link between the results and the conclusions is now compelling. Congratulations to the authors for their important contribution to the literature: an intriguing question, an innovative procedure, advanced data analyses, and a carefully written manuscript.

Reviewer #2

(Remarks to the Author)

Reviewer #3

(Remarks to the Author)

I am grateful to the authors for their thoughtful work in their revisions and their responses. My previous concerns are addressed perfectly and I would be very happy to see this paper published in its present form.

REVIEWER COMMENTS

Reviewer #1 (Remarks to the Author):

This manuscript addresses a longstanding issue of interest in psychology and cognitive neuroscience, and leverages recent advances in the computational understanding and neural study of controllability inference, to study a specific instance of this problem, namely, that of inferring agency in a collaborative game between two people.

Specifically, the study addresses how individuals assign outcome/feedback to their own performance versus that of others in social collaborative contexts, with varying levels of controllability. Controllability, in this study, is defined as how much an outcome is influenced by an individual's actions compared with that of another person, aiming to address the mechanisms by which people infer their control over outcomes in collaborative scenarios. The study makes significant contributions by introducing a clever novel behavioural task, combining computational modelling and fMRI. The authors identify active disambiguation (AD) as an exploratory mechanism for resolving uncertainty in social contexts, and demonstrate the importance of the supramarginal gyrus as a central brain region for agency-related processes during uncertainty. The use of Bayesian models to link the use of AD to intervention-based causal learning and exploration for information acquisition using two alternate behaviour models is interesting and explained well. The study has findings that have implications for understanding the mechanisms of how humans infer agency in different collaborative social interactions, and relevance to conditions such as depression where the estimation of control assignment is impaired.

The results will be of interest to the broader social cognitive neuroscience community, particularly those with affinity for advanced computational approaches to studying control.

First, we thank the reviewer for their interest, detailed comments and their positive evaluation. In particular, we would like to thank them for saying that our study “makes significant contributions by introducing a clever novel behavioural task, combining computational modelling and fMRI”. In the following, we address the reviewer's points and highlight our changes to the manuscript in red.

We did find the paper a bit dense, and complex, and therefore, hard to evaluate. Many different analyses are presented that it is hard to keep track. The reader might be helped by some sort of overview or bullet point summary of the key results at the start of the results section.

We thank the reviewer for pointing this out and their suggestion. We concur with the reviewer's recommendation to enhance readability. Since at present, *Nature Communications* does not allow graphical abstracts, we addressed the reviewer's suggestion in the following two ways:

(1) Firstly, we included the following new figure into our manuscript which summarises our main findings. We hope that the readability of our manuscript will be improved by this new Figure 8. In the figure, panel a summarises the basic credit assignment problem humans face when working together with others. Panel b summarises the three main behavioural findings in our study; that participants performed active disambiguation (AD, top panel), that they use AD and normal outcomes to infer their level of control (middle panel), and finally that participants use their knowledge about control to determine how to split up and assign ambiguous social prediction errors to themselves and others (bottom panel). Panel c shows corresponding neural activity in the supramarginal gyrus (SMG) which is associated with each of these behavioural findings.

Results, “Supramarginal Gyrus tracks prediction errors assigned to self and others”, lines 614 and 630-642

[...] Figure 8 shows a result summary of this study.

Figure 1. Controllability in social credit assignment and the role of the SMG. **a)** People often work together to achieve their joint goals. Therefore, whenever they experience an outcome for their joint efforts, they face the fundamental credit assignment problem of knowing how much to attribute ambiguous outcomes to themselves, others and their level of control. **b)** Participants varied their behaviour by making deliberate mistakes (“active disambiguation”, AD) when it was useful for credit assignment. They had insight into AD, and it allowed them to reduce uncertainty and assign credit more accurately. They employed this behavioural strategy to infer their level of control, by comparing outcomes obtained through AD and normal performance. When participants knew their level of control, they used this knowledge to assign ambiguous PEs to themselves and others appropriately. **c)** Neural activity in the supramarginal gyrus tracked each of these processes. At the time of action, it signalled AD (shown are whole-brain cluster-corrected maps of the effect of AD at the time of action, see also Figure 5b). At the time of outcome, it tracked both the inferred level of control and the components necessary for this learning process. Finally, SMG activity represented the portions of the PE attributed to oneself versus another person.

(2) Secondly, to enhance readability even more, we included short paragraphs summarising the main findings in the result subsections. We hope that these changes will help the reader keep track of our findings.

Results, “Estimates of controllability modulate social prediction errors”, lines 258-261

Taken together, participants tracked the feedback they observed in the task but attributed it to the correct causes less well. To assign social PEs, they adjusted their beliefs about themselves and others in relation to their controllability beliefs. This behaviour was similar to that of the Active learning model.

Results, “Active disambiguation enables inference of controllability”, lines 427-430

In sum, participants used the information afforded from AD in order to infer their level of control over the feedback. They did so by varying their behaviour and observing the resulting feedback changes. Comparing their behaviour with three computational models suggests that this was a deliberate strategy which requires insight into the nature of AD.

Results, “Supramarginal Gyrus tracks prediction errors assigned to self and others”, lines 612-614

Overall, in situations where control is explicit, SMG tracked outcomes as social PEs which are modulated by individuals’ control beliefs. In contrast, the ventral striatum signalled the observed feedback.

We hope that these changes improve readability of our manuscript. Finally, we would like to highlight a paragraph at the end of our introduction which summarises our main findings (see Introduction, lines 97-111).

There are a few specific things that we have difficulty wrapping our heads around.

First, to what extent does the current set-up isolate the inference of controllability specifically, instead of uncertainty or unpredictability more generally? It is clear that people updated control ratings as a function of uncertainty-triggered ADs, but does this mean they compute outcome controllability in the strict sense, or do they instead track outcome predictability more generally, and engage in AD to increase predictability when it drops?

We thank the reviewer for this question. The reviewer raises an important issue about the relationship between predictability, uncertainty and controllability. As pointed out by Ligneul (2021), it is important to distinguish task controllability from predictability and in some definitions of control, these two are confused. For example, while a weather forecaster can accurately predict the weather, they do not control it even though their action (weather forecast) typically predicts the weather (Ligneul, 2021).

In our study, we distinguish between causation, control, uncertainty and predictability. In our manuscript, when we refer to causation, we mean how different latent causes (such as one's own or another's actions) bring about outcomes in the environment. With control, we refer to the degree to which one's actions cause changes in the environment. In our study, we examine interpersonal control which is how people come to understand their level of responsibility over outcomes in social interactions. This means that by knowing one's level of control, people can understand the degree to which their actions cause changes in the environment. Notably, there is a certain degree of ambiguity to the concept of control and it is closely linked to the sense of agency (Farrer & Frith, 2022; Haggard, 2017; Moscarello & Hartley, 2017; Ohata et al., 2020) and perceived self-efficacy (Bandura, 1982; Zorowitz et al., 2020) (see also our response to a later question from the reviewer about action efficacy). When we refer to uncertainty in our task, we refer to how uncertain the feedback is with respect to its underlying causes (self, other and control), rather than how predictable it is. In our manuscript, we show that AD reduces uncertainty but we believe that uncertainty is not the same as (un)predictability. While the feedback in our task fluctuated according to the trial-wise performances of the participant and the other player, it was still relatively stable and, therefore, predictable (except for when the other player's performance changed unexpectedly once during the task phase, as pointed out by the reviewer in their question below). This means that, assuming the participant keeps performing normal trials, then after the first trial, the feedback was fairly predictable because similar performance (normal trials) resulted in similar feedback. In contrast and importantly, the hidden causes of the feedback were uncertain and had to be inferred. Indeed, Figure 2d shows that participants could predict the feedback better than their own performance, the other player's performance and their level of control, see Figure 2d in the main text:

As pointed out by the reviewer, performing AD reduced uncertainty but only because on those AD trials, participants knew their own performance on that particular trial as being zero. This means that on that trial, they should have no uncertainty about how well they did on that trial. In other words, the number of uncertain or ambiguous latent causes determining the feedback reduced by one (self performance is now no longer a determinant of outcome). In this way, this did indeed reduce the ambiguity of the feedback. Notably, however, we would argue that this is not the same as reducing the predictability of the feedback because before performing the first AD trials, participants were not able to accurately predict what the AD feedback will be. Once they have observed an AD feedback, however, they are able to form an accurate prediction about how high or low the feedback is on an AD trial.

Since we believe that the reviewer raises an important issue here, we have made the following changes to our manuscript which aim to clarify the distinctions between these concepts in our introduction and methods sections.

Introduction, lines 83-89

Controllability is closely linked to constructs such as sense of agency (Farrer & Frith, 2002; Haggard, 2017; Moscarello & Hartley, 2017; Ohata et al., 2020) and perceived self-efficacy (Bandura, 1982; Zorowitz et al., 2020). Here, we examine interpersonal control, that is how people come to understand their level of responsibility over outcomes in social interactions. A key aspect of establishing control is establishing whether outcomes are contingent on our actions over and above any contingency that already exists between outcomes and states of the environment (Ligneul, 2021; Ligneul et al., 2022). Analogously, establishing control in social situations requires the establishment of each individual's degree of agency relative to one another. [...]

Methods, "Active disambiguation as a behavioural strategy", lines 1218-1229

In our task, we distinguish between uncertainty and unpredictability. Uncertainty refers not to how predictable the feedback is, but to the ambiguity of its underlying causes. This means the degree to which it is caused by one's own performance, another's performance, and each agent's level of control. While the feedback fluctuated based on the participants' and the other's trial-wise performance, it was generally stable and therefore predictable. In other words, similar performance (normal trials) resulted in similar feedback. The core challenge for participants was to infer the hidden, uncertain causes of this feedback. We hypothesized that participants used AD to reduce the uncertainty rather than unpredictability of the feedback. This is because on an AD trial, participants knew their own performance was zero and this reduced one uncertain factor influencing the feedback. However, this is different from making the feedback more predictable because participants could not accurately predict AD feedback before doing AD for the first time. Therefore, in the above analysis we predict how trial-wise uncertainty is modulated by AD.

Another way to ask this question is: What are the (internal vs external) determinants of uncertainty? Only the ‘other’s’ performance level fluctuated to disentangle the variables of interest. Does this mean that uncertainty peaked disproportionately when the others performance level changed? Given that uncertainties were averaged in the models across ‘control’ and ‘other’, it is not clear whether uncertainty was driven primarily by uncertainty about control or about the other-performance. And whether subjects were tracking separate estimates of self-, other-, and control-related uncertainties at all? From Figure 2D it appears that they do track these components, albeit less precisely than overall uncertainty.

We thank the reviewer for raising this. Below, we examine in depth the factors contributing to uncertainty and relationships between self and other uncertainty. Before, we just wanted to briefly clarify our definitions of the various relevant variables. We discriminate between uncertainty about the causes of the overall feedback, and the degree to which the overall feedback can be predicted from previous trials (see also our response to the previous question). The feedback might be highly predictable (because it is relatively stable within a block), but it can still be highly uncertain because its latent causes are unknown.

In our task, participants needed to learn about the latent causes of feedback, specifically about their own level of control, their own performance and someone else’s performance. This means they needed to figure out (and rate) the *true levels* of these three unknowns. This also means that each of these three unknowns (self, other, control) is estimated and there is an *uncertainty* attached to each of these three estimates. In the Self-Other phase, self and other performances were uncertain while control was known and certain. In the Control-Other phase, other and control were uncertain while the self was known. Importantly, the true level and the uncertainty of, for example, control are independent (for example, the participant might estimate their level of control is high but they might be either very certain or very uncertain about that estimate; equally they might estimate that their level of control is low but they might be either very certain or uncertain about that estimate). We measured or derived each of these estimates and the uncertainty about the estimate independently. We obtained participants’ estimates/beliefs about the true levels from their ratings, and the uncertainties about the true levels from the Bayesian observer model (“Active learner”). We used this model to estimate trial-wise uncertainty because we did not ask participants to rate their uncertainty but only their beliefs about the true levels of self, other and control. Uncertainties were extracted as the 95% credible interval around the estimates from the Active learner model. Additionally, we manipulated the true levels of self, other and control in each task block and phase according to predetermined schedules. In contrast, the uncertainties around the true levels of self, other and control were not explicitly manipulated but estimated from the Active learner model. Below, we show a correlation matrix of the true levels, uncertainties and, for reference, the feedback participants observed (added as Supplementary Figure 7).

This correlation matrix shows that the true levels of control, self and other are unrelated to their respective uncertainties (absolute $r_s < 0.2$, combined data from MRI and online samples). Additionally, this plot might also answer the reviewer's question about whether self, other or control uncertainty mainly drove the overall uncertainty: Since uncertainty was computed as the average across other and control uncertainties in the Control-Other phase, and self and other uncertainties in the Self-Other phase, each of these two uncertainties contributed equally to the overall, averaged uncertainty. The correlation matrix shows that each individual uncertainty correlates highly with the average uncertainty ($r_s > 0.8$).

To clarify, in this Figure 2d (see also reply to first question above), we do not plot the uncertainties reported by the participants, but their beliefs about the true levels of control, other and self. Below is the equivalent plot but rather than plotting participants' rating accuracies, here we plotted the uncertainties about each estimate extracted from the Active learner model (added as Supplementary Figure 8a).

Note that in this plot, the individual data points correspond to the Active learner estimates for each participant. At the start of each phase, the Active learner had the same wide uncertainty for each unknown variable and each participant. Therefore, each individual data point for 'Start' has the same uncertainty value in the plot and is covered by the bigger data point denoting the mean. Additionally, this plot and Figure 2d show the standard error of the mean, which in both cases is so small that the bigger data point denoting the mean hides the error bars.

To clarify, in our task, we did not just manipulate the other's performance level between task phases and blocks but also participants' own performance and control level. This allowed us to disentangle all our variables of interest. Based on the reviewer's comment, we clarified this in the manuscript methods as follows:

Methods, Experimental design, lines 975-977

Schedule. [...] Each task block was divided into distinct task phases signalled to participants (see more information above). In both Self-Other and Control-Other phases, participants played 16 trials together with new other players and new control levels. Changing the true self level from task block to task block, and the true control level and other performance level between phases allowed us to decorrelate our variables of interest. To further decorrelate them, on the 12th trial of each phase, the other's performance level changed unannounced. [...]

While each to-be-learned variable (self, other and control) changed between task blocks and phases, it is true that the other's performance is the only one that changed unannounced within a phase. Indeed, other uncertainty and therefore also the averaged uncertainty peaked when the other's performance changed unannounced. The following plot shows the uncertainties (averaged, control, other, self) for each task block, averaged across participants (added as Supplementary Figure 8b):

Note that while it is clear that these changes in the other's performance lead to increased uncertainty, the highest uncertainty is still in the beginning of each phase.

In Figure 3e, we show that (averaged) uncertainty reduces after AD. To illustrate other factors influencing the dynamics driving uncertainty, we expanded our existing regression analysis by another binary regressor coding whether the other player's performance changed unannounced during the phase and in below plot show all the other regressors of this analysis (added as Supplementary Figure 8c). Note that the main finding from this regression, that AD reduces uncertainty, does not change by including this additional regressor (the updated Figure 3e looks almost identical to before). In addition to the effects of AD on uncertainty (also shown in main Figure 3e), we found that uncertainty reduces over time within a phase (*trial*, $F(1,65)=5813.88$, $p<0.001$, $\eta^2=0.99$) and increases when the other's performance changed unannounced during the task phase (*other changed within phase*, $F(1,65)=1124.28$, $p<0.001$, $\eta^2=0.95$). Uncertainty was not different between task phases (*task phase*, $F(1,65)=2.65$, $p=0.11$, $\eta^2=0.04$). Note that trial number has the largest effect on uncertainty, confirming that uncertainty is highest in the beginning of the phase.

Regarding the reviewer’s question about whether participants were tracking the individual uncertainties at all: Since they did not self-report the individual uncertainties, we can only investigate whether their behaviour is driven by the individual uncertainties. We did so in our replies to the reviewer’s next two questions, below.

We included the updated analyses and results in our manuscript as follows:

Results, “Active disambiguation as a behavioural strategy”, lines 333-335

We found that uncertainty was indeed reduced in the trials directly after AD (Effect on uncertainty: *1 trial after AD*, $F(1,65)=301.54$, $p<0.001$, $\eta^2=0.82$; *2 trials after AD*, $F(1,65)=282.17$, $p<0.001$, $\eta^2=0.81$; Figure 3e), [...]

Results, Figure 3 legend, line 365

Figure 3. Active disambiguation (AD). [...] **e** Trial-wise uncertainty, extracted from the active Bayesian learning model, is reduced if the previous trial or the one before was an AD trial. The effects of other variables influencing uncertainty are shown in Supplementary Figure 8c. [...]

Methods, “Active disambiguation as a behavioural strategy”, lines 1205-1216

Next, we tested whether AD reduces the trial-wise uncertainty (Figure 3e). The following regression model was fit:

$$\text{uncertainty} \sim AD_{t-1} + AD_{t-2} + \text{trial number} + \text{other changed} + \text{phase} \quad (1)$$

where *uncertainty* is the current trial’s average uncertainty. AD_{t-1} and AD_{t-2} are binary indicator variables for whether there was an AD one or two trials before the current trial. *Trial number* is time in the current phase, and *phase* is a binary variable for whether the current trial is in the Self-Other phase or Control-Other phase. *Other changed* is a binary variable for whether the other player’s performance has changed (0 until 11th trial, 1 from 12th to 16th trial, see also “Schedule”) and accounts for the uncertainty changing when the other player’s performance changed unannounced during a phase.

Methods, Experimental design, lines 967-978

Schedule. We used a pre-determined schedule to manipulate the true self-performance level, other performance level and control levels per task block and phase. We used a within-participant design and each participant experienced the same schedule of self, other and control levels. Supplementary Figure 3a-b shows the schedule of true values participants had to learn in the four task blocks. Within each task block, the true self level stayed constant and only changed between blocks. On each trial within each task block, participants played one game. Each task block was divided into distinct task phases signalled to participants (see more information above). In both Self-Other and Control-Other phases, participants played 16 trials together with new other players and new control levels. Changing the true self level from task block to task block, and the true control level and other performance level between phases allowed

us to decorrelate our variables of interest. To further decorrelate them, on the 12th trial of each phase, the other's performance level changed unannounced. [...]

Methods, “Bayesian observer models”, “Model parameters”, lines 1094-1096

[...] For reference, Supplementary Figure 7 shows the correlation of the extracted uncertainties of the Active learner model with the task schedule.

Supplementary figures

Supplementary Figure 7. Correlation matrix between the task schedule of true self, other and control levels, the feedback observed by participants, and the uncertainties extracted from the Active learning model. The true levels of control, self and other are unrelated to their respective uncertainties (absolute $r_s < 0.2$). Data from both MRI and online samples is included.

Supplementary Figure 8. The dynamics of uncertainty (extracted from the Active learner model). A) From the start to the end of a phase, uncertainty reduced (averaged uncertainty, $F(1,65)=7103.74$, $p<0.001$; control uncertainty, $F(1,65)=3807.87$, $p<0.001$; other uncertainty, $F(1,65)=6249.37$, $p<0.001$; self uncertainty, $F(1,65)=10184.59$, $p<0.001$). To test whether uncertainties reduced over time, here we plot the equivalent of Figure 2d but for uncertainties rather than rating accuracies. For each type of uncertainty (averaged, control, other and self) as dependent variable, we ran an ANOVA with *time in phase* ('Start' or 'End') as within, and *sample* (MRI or online) as between participant factors. At the start of each phase, the Active learner had the same wide uncertainty for each unknown variable and each participant. Therefore, each individual data point for 'Start' has the same uncertainty value in the plot and is covered by the bigger data point denoting the mean. Additionally, this plot and Figure 2d show the standard error of the mean, which in both cases is so small that the bigger data point denoting the mean hides the error bars. **B)** Uncertainties plotted for each game block and by task phase, averaged across Active learner model fit to MRI and online sample. While all uncertainties are highest at the start of each task phase, other and therefore averaged uncertainties increased when the other player changed performance unannounced. **C)** Full results plot of the regression shown in Figure 3e in the main text. (Averaged) uncertainty was reduced on trials following AD (as reported in main text), and uncertainty reduced over time within a phase (trial, $F(1,65)=5813.88$, $p<0.001$, $\eta^2=0.99$). Uncertainty increased when the other's performance changed unannounced during the task phase (other changed within phase, $F(1,65)=1124.28$, $p<0.001$, $\eta^2=0.95$). Averaged uncertainty did not differ between task phases (task phase, $F(1,65)=2.65$, $p=0.11$, $\eta^2=0.04$).

Is there a way to quantify the contribution of ‘control’ uncertainty versus the uncertainty over other-performance to ADs, and to updating of subjective control ratings? And is there a way to quantify the relative contribution of ADs (which were greatest when uncertainty was greatest, so we assume upon a reversal of other’s performance level) to estimating the other’s performance level rather than control?

We thank the reviewer for these suggestions. These are interesting analyses to follow-up on our findings that participants used AD to learn their control (Figure 4), and that participants reported finding AD trials most helpful for inferring their level of control (Supplementary Figure 9f). We addressed the reviewer’s comments using two analyses:

(1) Firstly, we examined whether AD trials were driven more by the uncertainty about control than about the other player in the Control-Other phase. Since control and other uncertainties are highly correlated ($r=0.75$, Supplementary Figure 7), it is difficult to disentangle their effects when they are included as two separate regressors in one regression model. Therefore, we predicted the occurrence of AD trials by the difference between control and other uncertainty to quantify the relative contribution of control versus other uncertainty to AD trials. Trial number was included as a regressor of no interest. Intuitively, if the difference between control vs other uncertainty has a positive effect on AD, then AD trials were more driven by control than other uncertainty. If control vs other uncertainty has a negative effect, then AD trials were more driven by uncertainty about the other player. We did not find a significant effect of the relative control vs other uncertainty on AD (control vs other uncertainty, $\beta=0.10$, 95% CI=(0.00,0.19)). However, it is noteworthy that control vs other uncertainty appears to have a (non-significant) positive trend effect on AD which is present in all participants but two. This means that AD tended to be more driven by uncertainty about control than other uncertainty. However, since this effect was not significant, we cannot draw firm conclusions from this analysis.

(2) We addressed the next two suggestions by the reviewer in one joint analysis. We examined which variables contributed more to learning about control compared to the other player. We hypothesized that participants updated their beliefs about their control more than about the other player when control uncertainty was higher than other uncertainty, and following switch trials from AD to normal or vice-versa. Based on two of our previous findings, we also expected that AD trials contributed more to learning about control than the other player. These findings were that participants reported AD trials as more helpful to learn about control than other (Supplementary Figure 9F), and that they did not learn about the other player from AD trials (Supplementary Figure 9E).

To test these hypotheses, we analysed all trials from the Control-Other phase and predicted the difference between the absolute belief changes of control and the absolute belief changes of other. This measure allowed us to investigate whether participants updated their beliefs about the control more than the other player. We predicted this relative belief update by the difference between control vs other uncertainty (similar as in analysis 1 above). In addition, we included whether on the trial before, an AD trial occurred, a switch trial from AD to normal or vice-versa occurred, whether the other player changed performance unexpectedly. The regressor ‘Other changed’ denoted whether the trial occurred when or after the other player changed performance unexpectedly. Furthermore, including interaction terms between other performance changes and AD and switch trials allowed us to investigate whether participants’ relative learning about other and control changed before and after the other player changed performance unexpectedly. Finally, we included trial number as a regressor of no interest.

As expected, we found that participants updated their beliefs about control more than the other player if the control uncertainty was higher than the other uncertainty (control vs other uncertainty, $\beta=0.06$, 95% CI=(0.02,0.10), see Figure below). Next, we hypothesized that both AD and switch trials might be particularly informative to update beliefs about control. Indeed, we found that participants updated their beliefs about their control more strongly than about the other following both AD and switch trials (AD trials, $\beta=0.14$, 95% CI=(0.04,0.24); Switch trial, $\beta=0.17$, 95% CI=(0.07,0.27)). We did not find that participants learned more about the other when the other's performance changed unannounced (non-significant main effect Other changed, and the interaction terms Other changed * AD trial and Other changed * Switch trial).

Overall, these results suggest that throughout the Control-Other phase, participants used AD and switch trials more to learn about their level of control than to learn about the other player. Furthermore, participants updated their beliefs about their control particularly when the uncertainty about control was higher.

We included these analyses in the manuscript as follows.

Results, Active disambiguation enables inference of controllability, lines 420-425

We considered other ways in which participants might have used equivalent strategies to infer self and other, and show that participants did not use these strategies to the same degree to infer control (Supplementary Figures 9 and 10). After finishing the task, participants also reported that they found AD trials most helpful for inferring their control (Supplementary Figure 9f). This suggests that participants used AD efficiently for learning about their control and that they had insight into this.

Methods, Inferring control from active disambiguation, lines 1304-1306

We examined factors driving AD trials and relative control vs other updating in the supplementary information (Supplementary Methods, section 'Effects driving AD trials and learning in the Control-Other phase', Supplementary Figure 8).

Supplementals, Behavioural analyses, "Effects driving AD trials and learning in the Control-Other phase", lines 421-476

Effects driving AD trials and learning in the Control-Other phase. We examined the factors driving participants to perform AD and learn about their control rather than about the other in the Control-Other phase. We performed two analyses to address this (results are shown in Supplementary Figure 10). Firstly, we examined whether AD trials were driven more by the uncertainty about control than about the other player in the Control-Other phase. We ran the following hierarchical regression model across MRI and online samples to predict whether the current trial in the Control-Other phase is an AD trial or not:

$$AD \sim \text{control vs other uncertainty} + \text{trial number} + (\text{control vs other uncertainty} + \text{trial number} | ID) \quad (S2)$$

with

$$\text{control vs other uncertainty} = 100 * \text{control uncertainty} - \text{other uncertainty} \quad (S3)$$

Where *control vs other uncertainty* is the difference between control and other uncertainty. Since control and other uncertainties are highly correlated ($r=0.75$), it is difficult to disentangle their effects when they are included as two separate regressors in one regression model. Therefore, we predicted the occurrence of AD trials by the difference between control and other uncertainty. This allowed us to quantify the relative contribution of control versus other uncertainty to AD trials. Intuitively, if the difference between control vs other uncertainty has a positive effect on AD, then AD trials were more driven by control than other uncertainty. If control vs other uncertainty has a negative effect, then AD trials were more driven by uncertainty about the other player. *Trial number* was included as a regressor of no interest. All trials in the Control-Other phase were included in this analysis and the data from online and MRI samples were combined.

Secondly, we investigated which factors contributed more to learning about control compared to the other player. We hypothesized that participants updated their beliefs about their control more than about the other player when control uncertainty was higher than other uncertainty, and following switch trials from AD to normal or vice-versa. Based on two of our previous findings, we also expected that AD trials contributed more to learning about control than the other player. These findings were that participants reported AD trials as more helpful to learn about control than other and that they did not learn about the other player from AD trials (Supplementary Figure 9E-F). We used the following regression analysis to test this:

$$\begin{aligned} \text{Control vs other updating} \sim & \text{control vs other uncertainty} + \text{trial number} + \text{other changed} \\ & * (\text{AD} + \text{switch trial}) + (\text{control vs other uncertainty} + \text{trial number} + \text{other changed} \\ & * (\text{AD} + \text{switch trial}) | ID) \end{aligned} \quad (S4)$$

with

$$\text{control vs other updating} = 100 * |\text{control update}| - |\text{other update}| \quad (S5)$$

where *control vs other updating* is the difference between the absolute belief changes of control and the absolute belief changes of other. This measure allowed us to investigate whether participants updated their beliefs about the control more than the other player. We predicted this relative belief update by the difference between *control vs other uncertainty* (similar as in analysis above). In addition, we included whether the previous trial was an *AD trial*, or was a *switch trial* from AD to normal or vice-versa. The regressor *Other changed* denoted whether the current trial occurred when or after the other player changed performance unexpectedly. Furthermore, including interaction terms between other performance changes and AD and switch trials allowed us to investigate whether participants' relative learning about other and control changed before and after the other player changed performance unexpectedly. Finally, we included *trial number* as a regressor of no interest. All trials except for the first phase trial of the Control-Other phase were included because on those trials, we could extract rating updates. Data from online and MRI sample were combined in this analysis.

Supplementary figures

Supplementary Figure 10. Effects driving AD trials and learning in the Control-Other phase. A) Firstly, we examined whether AD trials were driven more by the uncertainty about control than about the other player in the Control-Other phase. Intuitively, if the difference between control vs other uncertainty has a positive effect on AD, then AD trials were more driven by control than other uncertainty. If control vs other uncertainty has a negative effect, then AD trials were more driven by uncertainty about the other player. We did not find a significant effect of the relative control vs other uncertainty on AD (control vs other uncertainty, $\beta=0.10$, 95% CI=(0.00,0.19)). However, it is noteworthy that control vs other uncertainty appears to have a (non-significant) positive trend effect on AD which is present in all participants but two. This means that AD tended to be more driven by control than other uncertainty. However, since this effect was not significant, we cannot draw firm conclusions from this analysis. **B)** As expected, we found that participants updated their beliefs about control more than the other player if the control uncertainty was higher than the other uncertainty (control vs other uncertainty, $\beta=0.06$, 95% CI=(0.02,0.10)). Next, we hypothesized that both AD and switch trials might be particularly informative to update beliefs about control. Indeed, we found that participants updated their beliefs about their control more strongly than about the other following both AD and switch trials (AD trials, $\beta=0.14$, 95% CI=(0.04,0.24); Switch trial, $\beta=0.17$, 95% CI=(0.07,0.27)). We did not find that that participants learned more about the other when the other's performance changed unannounced (non-significant main effect Other changed, and the interaction terms Other changed * AD trial and Other changed * Switch trial). Overall, these results suggest that throughout the Control-Other phase, participants used AD and switch trials more to learn about their level of control than to learn about the other player. Furthermore, participants updated their beliefs about their control particularly when the uncertainty about control was higher.

Can the timecourses of self-, other-, control-related uncertainty, as well as averaged uncertainty and ADs be visualized?

We performed a set of five new time course regression analyses to address this point. First, we visualised the time courses of averaged uncertainty and AD time-locked to the time of action for SMG and area 7, the ROIs we used throughout in our manuscript. The results of this new time course analysis (ROI-GLM 8) are now included in Supplementary Figure 18a for the SMG and Supplementary Figure 19a for area 7.

In this time course analysis, we included the same regressors of no interest as in the whole-brain analysis for the time of action. For completeness, the correlation matrix of the regressors of this new ROI-GLM is shown in the new panel D of Supplementary Figure 13:

D

ROI-GLM 8

AD	0.08	-0.06	-0.05	0.06	0.38	0.03	0.01	1
Uncertainty	-0.09	0.08	-0.01	0.15	-0.03	-0.24	1	0.01
Trial	0	0	0	-0.1	0.02	1	-0.24	0.03
Switch trial	0.05	-0.06	-0.02	-0.01	1	0.02	-0.03	0.38
RT	-0.2	0.48	0.19	1	-0.01	-0.1	0.15	0.06
Game type 1	-0.33	-0.33	1	0.19	-0.02	0	-0.01	-0.05
Game type 2	-0.33	1	-0.33	0.48	-0.06	0	0.08	-0.06
Game type 3	1	-0.33	-0.33	-0.2	0.05	0	-0.09	0.08

Game type 3
Game type 2
Game type 1
RT
Switch trial
Trial
Uncertainty
AD

Please note that this new time course analysis is for visualisation purposes only and no statistical tests were performed on the resulting time courses. The reason for this is that in our manuscript, we selected SMG and area 7 as ROIs because in the whole-brain analysis, they showed overlapping cluster-corrected effects for averaged uncertainty and AD at the time of action. This means that the new time courses shown above re-visualise effects from the whole-brain results, and performing a statistical test on the time-courses of ROIs would be circular. The whole-brain effects of uncertainty and AD are shown in Figure 5 in the main text. Supplementary Table 1 lists the peak coordinates of AD and uncertainty from the whole-brain analysis, including the peaks in SMG and area 7.

Having re-established that our time course analysis recovers our whole-brain effects, next we visualised the time courses of self, other and control uncertainties in the different task phases. Again, we did not perform statistical tests on these time courses because the individual uncertainties are highly correlated with their average ($r > 0.8$, see new panels E and F in Supplementary Figure 13), which is the contrast that was used initially to define the ROIs (Figure 5 in main text). First, we examined the time courses of uncertainty about the other and control in the Control-Other phase. As noted in our response above, control and other uncertainties showed high correlations with each other ($r = 0.79$) and with trial number ($r < -0.6$, see also new panel E in Supplementary Figure 13).

Therefore, we visualised their time-courses in separate ROI-GLMs as well as together in one ROI-GLM (similar to how we tested the effects of sPE and oPE in ROI-GLMs 3-6).

In the Control-Other phase, we found that when tested in separate ROI-GLMs, both control and other uncertainties showed strong effects in SMG. When we examined their time courses in one ROI-GLM, SMG appeared to only track control uncertainty. For completeness, we also conducted the same analyses in area 7.

SMG:

Area 7:

We included the methods and figures of these new ROI-GLMs in the Supplementals as follows:

Methods, “Time course GLMs (ROI-GLMs)”, line 1512-1514

In the supplementary materials, we visualise the effects of uncertainty and AD at the time of action using ROI-GLMs (Supplementary Information, “fMRI time course analysis on ROIs”, Supplementary Figures 18 and 19).

Supplementary methods, “fMRI time course analysis on ROIs”, lines 493-553

Effects of AD and uncertainties at the time of action. We used time course analyses to follow-up on the whole-brain effects we found related to AD and uncertainty during the action phase (main Figure 5). Similar to the ROI-GLMs reported in the main text, we focussed our analyses on SMG and area 7 as ROIs. Firstly, we visualised the effects of AD and uncertainty across Self-Other and Control-Other phase as time courses. Importantly, note that this ROI-GLM uses the same regressors of interest and time-locking as the whole-brain analysis based on which SMG and area 7 were defined. Therefore, no statistical tests were performed on the resulting time courses and they are shown for visualisation purposes only. The results of the following ROI-GLMs are shown in Supplementary Figures 18b-d and 19b-d. We ran the following ROI-GLM 8 to the action phase across Self-Other and Control-Other phases:

$$\text{BOLD} \sim \text{AD} + \text{uncertainty} + \text{trial} + \text{switch trial} + \text{RT} + \text{game type 1} + \text{game type 2} + \text{game type 3} (S6)$$

where *AD* is a binary variable referring to whether the current trial is AD or not. *Uncertainty* is the averaged uncertainty across self, other and control. We included the same regressors of no interest as were included in the action phase of the whole brain analysis. *Trial* number was included to account for time in block. *Switch trial* was included to control for any neural effects related to switching from AD to normal or vice-versa. To control for reaction-time related effects, we included *RT* as a regressor of no interest. Finally, binary regressors for three of the four game types were included (*game type 1*, *game type 2* and *game type 3*) to account for any differences between the games that participants played during the action phase. Note that here, we included three binary game type regressors so that the intercept of this ROI-GLM captured the fourth game type. The result of this ROI-GLM is shown in Supplementary Figures 18a and 19a. Regressor correlations are displayed in Supplementary Figure 13d.

Next, we examined whether SMG and area 7 represent the individual components of uncertainty. Specifically, we tested whether these ROIs tracked control and other uncertainties in the Control-Other phase, and self and other uncertainties in the Self-Other phase. Note that again, these regressors are not independent of the whole-brain effects based on which the ROIs were defined. Uncertainty was

computed as the average across self and other uncertainties in the Self-Other phase, and control and other uncertainties in the Control-Other phase. While the regressors in the following ROI-GLMs are not identical to the whole-brain regressors, they correlate highly with each other ($r > 0.8$, Supplementary Figure 13E-F). Therefore, we did not test the time courses for significance and the resulting time course plots are for illustration purposes only.

First, we investigated whether our ROIs represent control and other uncertainties at the time of action during the Control-Other phase. Since other and control uncertainties correlated highly with each other and with trial number (absolute $r > 0.6$, Supplementary Figure 13e), we tested their time courses in separate regressions (ROI-GLM 9 and 10) and jointly in one regression (ROI-GLM 11).

$$\text{BOLD} \sim \text{control uncertainty} + \text{AD} + \text{switch trial} + \text{RT} + \text{game type 1} + \text{game type 2} + \text{game type 3} \quad (\text{S7})$$

$$\text{BOLD} \sim \text{other uncertainty} + \text{AD} + \text{switch trial} + \text{RT} + \text{game type 1} + \text{game type 2} + \text{game type 3} \quad (\text{S8})$$

$$\text{BOLD} \sim \text{control uncertainty} + \text{other uncertainty} + \text{AD} + \text{trial} + \text{switch trial} + \text{RT} + \text{game type 1} + \text{game type 2} + \text{game type 3} \quad (\text{S9})$$

where *control uncertainty* and *other uncertainty* are the control and other uncertainty, extracted from the Active learner model. The other regressors included in these ROI-GLMs were parameters of no interest, similar to ROI-GLM 8.

Secondly, we investigated whether our ROIs represent the uncertainty about self and other at the time of action during the Self-Other phase. We tested their time courses in the following regression (ROI-GLM 12).

$$\text{BOLD} \sim \text{self uncertainty} + \text{other uncertainty} + \text{AD} + \text{trial} + \text{switch trial} + \text{RT} + \text{game type 1} + \text{game type 2} + \text{game type 3} \quad (\text{S10})$$

where *self uncertainty* and *other uncertainty* are the uncertainties around self and other, extracted from the Active learner model. The other regressors were of no interest, similar to the ROI-GLMs above.

Supplementary Figures

Supplementary Figure 2 (relating to Figures 6-7, Supplementary Figures 14-19). Correlation matrices for regressors included in time course analyses (ROI-GLMs). **A)** Correlation of regressors shown in main Figure 6. Regressors for the ROI-GLMs 1 and 2 are shown in one correlation matrix to highlight that the regressors of control belief (ROI-GLM 1) and feedback differences (ROI-GLM 2) show low correlations ($r < 0.3$). **B)** Correlations of regressors shown in main Figure 7. Since sPE and oPE showed a correlation of $r = 0.55$, we tested the time courses of these regressors first separately (ROI-GLMs 3 and 4), and then together (ROI-GLM 5). We also tested the effects of tPE and control separately in ROI-GLM 6. **C)** Regressor correlations for Supplementary Figure 17 (ROI-GLM 7). **D)** Regressor correlations for Supplementary Figures 18a and 19a (ROI-GLM 8). **E)** Correlations of regressors shown in Supplementary Figures 18b-c and 19b-c. Due to the high correlations between other and control uncertainties, we tested their time courses separately (ROI-GLMs 9 and 10) and together in one ROI-GLM 11. **F)** Correlations of regressors shown in Supplementary Figures 18d and 19d (ROI-GLM 12).

Supplementary Figure 3. SMG time courses of AD, (average) uncertainty and the uncertainties about self, other and control in the action phase. **A)** Time courses of uncertainty and AD across both Self-Other and Control-Other phases (ROI-GLM 8). Note that the SMG ROI was defined based on the overlapping clusters of the same regressors (uncertainty and AD) from the whole-brain analysis (shown in main Figure 5). Therefore, these time courses are purely for visualising this whole-brain effect in another way and no statistics have been performed on these regressors. **B-C)** Time courses of control and other uncertainties in the Control-Other phase. Since the regressors of control and other uncertainties showed high correlations ($r=0.79$), we visualised their time courses in separate ROI-GLMs first (panel B). When control and other uncertainties were tested in one ROI-GLM, due to their high correlations, their representations are weaker (panel C). **D)** Time courses of self and other uncertainties in the Self-Other phase. Regressor correlations are shown in Supplementary Figure 13D-F. $n=31$ MRI, mean beta weights are plotted as lines with s.e.m. as shaded intervals.

Supplementary Figure 4. Area 7 time courses of AD, (average) uncertainty and the uncertainties about self, other and control in the action phase. Since the area 7 ROI was defined based on whole-brain effects of uncertainty and AD, we did not perform statistics on these time courses to avoid circularity. This means these time courses are for illustration purposes only. **A)** Time courses of uncertainty and AD across Self-Other and Control-Other phases (ROI-GLM 8). **B-C)** Time courses of control and other uncertainties in the Control-Other phase, examined in separate and a joint ROI-GLM due to their high correlation. **D)** Time courses of self and other uncertainties in the Self-Other phase. Regressor correlations are displayed in Supplementary Figure 13D-F. $n=31$ MRI, mean beta weights are plotted as lines with s.e.m. as shaded intervals.

Second, and this is related to the previous question, while it is recognized that the process of inferring agency under study here represents an instance of the controllability inference problem that has recently attracted renewed interest, the similarities and differences between the algorithms and neural activity patterns observed here and those observed in those recent previous studies are not addressed explicitly. Specifically to what degree does the computational problem of inferring agency as operationalized here correspond algorithmically to the same problem as that of inferring environmental controllability by comparing actor and spectator models of the world? Previous fMRI, lesion and brain stimulation work has highlighted a role for medial frontal cortex (pgACC and LPFC in particular) in controllability inference. Is the SMG doing what the pgACC is doing for controllability inference when the algorithmic problem is framed as a social problem?

Thank you for raising this. We address this comment by firstly, including a more explicit definition of the instance of control we study in this manuscript (see also our response to an earlier comment by the reviewer). Specifically, we examine interpersonal control which is how people come to understand their level of responsibility over outcomes in social interactions. In our introduction, we highlight that previous work by Ligneul et al (2021, 2022) showed that in non-social contexts, to infer control, people need to establish whether outcomes are caused by our actions over and above general contingencies present in the environment. Similarly, in our work, we show that in social situations, people need to establish whether their actions cause outcomes independent from the actions of other people. However, there are also some distinctions between Ligneul’s framework and ours. Ligneul and colleagues studied controllability in non-social contexts where state transitions are either determined by one’s own actions or dynamics in the environment. We examined controllability in social environments when people need to discern the influence their actions have over parametric outcomes that are always influenced to some degree by their and someone else’s actions. Our findings highlight the importance of making varied exploratory actions to infer control, while Ligneul and colleagues highlight the importance of tracking outcomes when an action is present vs considering the environmental dynamics. These differences might indeed explain why Ligneul’s findings on more prefrontal regions, while our neural findings are in the SMG (and consistent with some previous work involving the SMG in agency attributions (Sirigu et al., 1999; Farrer & Frith, 2002; Ohata et al., 2020)).

We made the following changes to our manuscript to incorporate the reviewer’s comment, together with some of our other answers to their comments.

Discussion, lines 687-700

At the core of control inference studied here is a comparison process and prior work highlighted that control can be inferred through comparing either observed and predicted action outcomes (Frith et al., 2000) or the likelihoods of outcomes in the presence or absence of one’s actions (Allan, 1980; Ligneul, 2021; Ligneul et al., 2022; Maier & Seligman, 1976). Ligneul et al. studied how people track their control over categorical outcomes by comparing a ‘spectator’ and ‘actor’ model in non-social contexts. Here, we show that a similar comparison process is at play in social situations with ambiguous and parametric joint outcomes, but that it additionally requires comparing outcomes caused by a range of actions, including the strategic use of poor or self-defeating actions. These differences likely account for our divergent neural findings, with our results implicating the SMG, in contrast to the prefrontal regions reported by Ligneul. Engaging in exploratory actions also allows sensing one’s control over different objects in non-social contexts (Wen et al., 2020). On an abstract level, there are some similarities of this AD-guided control inference to prior work showing that people take advice despite already being highly confident (Pescetelli & Yeung, 2021). While initially appearing suboptimal, both behaviours are beneficial for long-term learning about the quality of advice (Pescetelli & Yeung, 2021) or controllability.

Introduction, lines 83-85

Controllability is closely linked to constructs such as sense of agency (Farrer & Frith, 2002; Haggard, 2017; Moscarello & Hartley, 2017; Ohata et al., 2020) and perceived self-efficacy (Bandura, 1982; Zorowitz et al., 2020). Here, we examine interpersonal control, that is how people come to understand their level of responsibility over outcomes in social interactions. A key aspect of establishing control is establishing whether outcomes are contingent on our actions over and above any contingency that already exists between outcomes and states of the environment (Ligneul, 2021; Ligneul et al., 2022). Analogously, establishing control in social situations requires the establishment of each individual's degree of agency relative to one another. [...]

Other comments:

The task description could be clearer right from the start. For example, it describes: “three hidden latent causes: their own performance, another fictive player’s performance, and their own control over the feedback (Figure 1A).” It might help to make more explicit right away that these refer to (i) own action efficacy, (ii) other’s action efficacy and (iii) the degree to which own, or other’s efficacy matters for the outcome. It would also be good to clarify whether participants are told that outcomes always depend on (own or others) actions – in other words, that environmental controllability was fixed.

To address this, we included more information on what we mean by performance and control. We refrain from explaining performance as action efficacy because this might lead to confusion because here, we understand performance more as ability or how well each agent does. In contrast, control is the degree of influence each agent exerts and in prior work, this has also been related to perception of self-efficacy (Bandura, 1982; Zorowitz et al., 2020). We added a more detailed explanation of performance and control in our results section (see below).

In the instructions, participants were told that the feedback is based on their own and the other player’s performance on that trial, and that therefore, they control (or influence) only part of the feedback. They were also given schematic pictures and example scenarios to clarify what it means when they influence (or control) the feedback more, the same, or less than the other player. Furthermore, we checked their understanding of the concepts of performance and control using task understanding questions which they had to complete successfully prior to starting the main task. To clarify this in our manuscript, we included our instructions relating to this in the supplementary information.

Results, Experimental structure, lines 118-125

Participants were asked to assign a single, overall outcome (feedback) to three hidden latent causes: their own performance, another fictive player’s performance, and their own control over the feedback (Figure 1a). Here, performance refers to the ability or how well the participant and the other player do. In contrast, control refers to each agent’s influence or responsibility, i.e. how much each agent’s performance matters for the feedback. The feedback varied parametrically over trials and was the sum of the participant’s own and the other player’s performance, weighted by their relative control levels over the outcome.

Methods, Experimental procedure, line 807

After providing written informed consent, they were instructed about the task and performed a task practice outside of the scanner (details can be found in the supplementary methods, ‘Task instructions’).

Supplementals, Supplementary Methods, lines 363-367

Task instructions. Prior to the task, participants clicked through a series of task instruction screens including practice trials. This was then followed by a multiple choice quiz to check their understanding. In below screenshots, we obscured any face stimuli used with icons.

Our aim is to understand how people learn and how this relates to mood.

In this study you are a team captain and your task is to put a good team together. For this, you will play four games, together with other fictive players. For each game, you need to figure out how good you and the other players are to put the best team together.

You can earn extra money for doing this well.

In the Slingshot Game, the aim is to hit the target with the ball.

Next, we will check that this game works. Please do your best in this system check.

[practice trials for Slingshot game]

The next game is the Bouncing Ball Game:

Press LEFT to release the ball. As soon as you think it has passed the target flag, press LEFT again. Next, we will again check that this game works. Please do your best in this system check.

[practice trials for Bouncing Ball game]

The Rock Slide Game: Press LEFT to release the ball. Press LEFT when you think the ball hits the target. Please do your best in the next system check.

[practice trials for Rock Slide Game]

Finally, the Light Game: After you press LEFT, the circles light up one after another. Press LEFT when you think the target circle lights up. Please do your best in the following system check.

[practice trials for Light game]

You will play every game round together with another player

You will both play at the same time, but you will never see the other play.

Your aim as team captain is to pick good teammates for each game! Therefore, you will need to assess your own and the other player's performance in each game.

After each game round, a colored bar will show you how well you both did together.

> How well you do can change from game to game: In some, you will do better than in others.

> The players are computer-generated and based on real people. Therefore, they may get better or worse at the games over time. Also, some players will do better or worse than others.

The feedback depends on how well you and the other do. Therefore, you influence only a part of the feedback.
 > Sometimes, you and the other influence the feedback equally: How well you do matters as much as how well the other does.

> In other cases, you have more influence than the other.

> You can also have less influence than the other.

Example: A participant does well in the game and the other player does not. If the other has more influence, they will still get a low feedback together. (see below)

Another example: In football, you might do much worse than the rest of your team. Your team can still make many goals because you as a single player don't matter as much as the rest of the team.

To put together your team, your task is to find out: (1) your influence over the feedback, (2) how well you do, and (3) how well the other player does.

You can do this by paying attention to the feedback after each game round. You can also try to change how you play the game to see how this affects the feedback.

Before each game round, you will make three ratings. Doing these ratings well is a crucial part of the study – you will get a bonus payment for doing them well and NOT for how well you play the games. The ratings will also help you to put a good team together after each game by selecting the best players and influence levels. Press MIDDLE to practice one game round with ratings. Your ratings don't matter for now.

Press LEFT and RIGHT to rate. Press MIDDLE to submit

[practice trial]

At the end of each game, you will be asked to choose the team mates and distribute the influence within your team. The image below shows you what this will look like.

Sometimes, we will ask you about your mood. Then, please take a moment to think about how you feel. This is an important part of our study. Well done, that's it for the instructions.

Please specify how AD errors were defined in the main text (not just methods), as this is key for understanding the main results section.

Thank you for this suggestion. We included the following information into our main text to highlight how we categorised AD trials.

Results, “Active disambiguation as a behavioural strategy”, lines 309-317

As hypothesised, we found evidence of AD; participants sometimes made errors in the games so extreme and obvious that their objective performance was transformed into a self-performance score of zero on that trial (see Methods and Supplementary Figure 4 for details). For example, in the rockslide game, this meant responding while the ball was still visible, either before or after it entered the black box and was close to the target cross. On those trials, participants’ objective game performance was transformed into a self-performance of zero, meaning they did not contribute to the feedback. Participants’ behaviour exhibited high levels of AD under high uncertainty when feedback was caused by multiple sources, and AD was therefore informative [...]

We missed a clear rationale for testing (and clear conclusions based on the observation of) reward signal in the ventral striatum.

The reason for us including the reward analysis in the ventral striatum was a sanity check whether we can find more traditional reward signals using our novel task. We added the following to our manuscript to explain our motivation more clearly.

Results, “Supramarginal Gyrus tracks prediction errors assigned to self and others”, lines 603-610

Finally, we tested whether neural activity tracked reward in our task, irrespective of any inference process. The aim of this analysis was to check whether in our task, we can find more classic reward signals. The ventral striatum has been implicated in representing reward in past studies. In an anatomically defined ROI of the ventral striatum, we found it indeed tracked the shown feedback, as well as trial-wise changes of the feedback at the time of the outcome (feedback, $t(30)=2.23$, $p=0.03$; feedback change, $t(30)=2.48$, $p=0.02$; ROI-GLM7, Supplementary Figure 17). This suggests that our novel tasks also allows investigating more traditional reward-related signals in the human brain.

Please clarify whether participants were aware of the fact that other-performance level could change during a game, but control level could not

During task instructions (see also response above including the precise instruction text), participants were instructed that the other “*players are computer-generated and based on real people. Therefore, they may get better or worse at the games over time. Also, some players will do better or worse than others.*”. In contrast, the level of control was held constant during a task phase and at the beginning of each task phase, participants were informed whenever the control level changed. We did so by including information screen prior to the first trial of each task phase (Self-Other phase, Self-only trials and Control-Other phase). In the Self-Other phase, this screen stated that they will now start playing a new game with a new other player, and displayed them their level of control. Their level of control was also shown on each rating screen and therefore it was explicit that the control could not change during a task phase; see also Figure 1c:

In contrast, at the beginning of each Control-Other phase, the information screen stated that their control “over the feedback has reset” and that from now on, they “might have more, equal or less influence than the other player.” To clarify this in our manuscript, we included the task instruction in the supplementary material (see also response above).

Participants were asked to report whether they generated an AD trial. What was the exact question they were asked? Although they do report that participants generated an AD trial when uncertainty was high, the participants were also primed in a way by the explicit instructions to explore alternative action strategies to best estimate their control levels. To strengthen the conclusion that AD behaviour is a generalizable phenomenon that does not depend on demand characteristics, it would be good, in future studies, to add a control block without such instructions, allowing isolation of this effect of instruction/demands characteristics.

In the post-task debrief questionnaire, participants answered two questions asking them whether they performed AD in the task:

1. “In the study, how were you able to know how well you and the other players did, and how much influence you had? For example, did you do badly on purpose to find this out?” This question could be answered as a free text.
2. “In the study, how many times per game did you do badly on purpose?” This could be answered on a scale from “Never”, “Less than once per game”, 1, 2, 3, 4, 5, 6, 7, 8, 9, 10, “More than 10 times per game”

In our manuscript, we analysed data from the second debrief question (see Figure 3d). We added this information to the relevant figure and supplementary methods and refer to it in our main text.

Next, we would like to address the second comment about whether our findings about AD behaviour are restricted to participants who have received a task hint. As part of another study (findings to be

reported elsewhere), we collected a large online sample of participants (n=352 included after data quality checks) who completed the same task as presented in this manuscript. However and importantly, they did not receive a hint in the instructions that they can change their behaviour in the games to see how this affects the feedback. This means that this sample allows us to examine whether participants employ AD without being primed to do so. Firstly, we found that these participants who were not primed to do AD still performed AD but they showed a lower proportion of AD compared to the primed participants reported in our manuscript (proportion of AD, averaged across Self-Other and Control-Other phases: unprimed online sample, 9.28%; [primed] online sample, 19.62%; [primed] MRI sample, 27.39%). Consistent with the findings reported in our manuscript, we found that the proportion of AD shows a strong drop when the feedback was not ambiguous and only reflected participants' own performance (panel A below). Secondly, we again found that the unprimed participants had insight into AD because the reported number of AD trials correlated with the number of AD trials we detected in their behaviour (Spearman's $\rho=0.62$, $p<0.001$; panel B below). This suggests that even though not priming participants to do AD indeed reduces AD behaviour in our task, participants still did AD and we found that this is modulated by uncertainty and that they had insight into using this strategy.

We refer to the exact wording of the debrief question and included the results from the new online sample in our manuscript as follows.

Results, Figure 3 legend, lines 355-362

[...] c) We indeed found that participants generated AD in phases when the feedback was ambiguous (Self-Other and Control-Other phases). Whenever the feedback was certain (Self-only trials), AD dropped strikingly. We found a similar pattern in another online sample who did not receive a task hint to change their game behaviour (Supplementary Figure 5a). Bars and dots show the mean proportion of AD. **d)** After finishing the task, participants reported how many AD trials they generated per game (debrief question: “In the study, how many times per game did you do badly on purpose?”). We found that their subjective reports of AD trials were significantly correlated with the objective frequency of AD trials identified in their behaviour. Again, we found that these results replicate in another online sample without the task hint to change their game behaviour (Supplementary Figure 5b).

Methods, Behavioural analyses, Active disambiguation as a behavioural strategy, lines 1193-1196

[...] For this, we extracted participants' responses to the debrief question "In the study, how many times per game did you do badly on purpose?" (see supplementary methods, "Experimental design", "post-task debrief questionnaire" for more details). [...]

Supplementals, Supplementary Figure 5

Supplementary Figure 5. AD behaviour in a large online sample (n=352 included after data quality checks) without the task hint to change game behaviour. We collected a version of our task online in which participants did not receive the hint that they can change their game behaviour during the instructions. This was part of another study and the findings will be reported elsewhere. This sample is included here to address a reviewer question about the degree to which AD behaviour is a generalizable phenomenon that does not depend on task instructions. **A)** Firstly, we found that these participants who were not primed to do AD still performed AD but they showed a lower proportion of AD compared to the primed participants reported in our manuscript (proportion of AD, averaged across Self-Other and Control-Other phases: unprimed online sample, 9.28%; [primed] online sample, 19.62%; [primed] MRI sample, 27.39%). Consistent with the findings reported in our manuscript, we found that the proportion of AD shows a strong drop when the feedback was not ambiguous and only reflected participants' own performance. Similar to our previous analysis (Figure 3c), we only included online participants for who we detected AD trials in their behaviour and who reported doing AD in the debrief questionnaire. This left us with n=147 online participants for this analysis. **B)** Secondly, we again found that the unprimed participants had insight into AD because the reported number of AD trials correlated with the number of AD trials we detected in their behaviour (Spearman's $\rho=0.62, p<0.001$). Here, we analysed the full online sample including participants who reported never having done AD and for who we did not detect AD (i.e. n=352 participants included). Overall, these results suggest that even though not priming participants to do AD indeed reduces AD behaviour in our task, participants still did AD and we found that this is modulated by uncertainty and that they had insight into using this strategy.

Why did they decide to have different rating scales for self and other ratings (0-100) as compared to the control rating scale (0-1). Would participants then perceive control to be more of a binary concept as compared to performance which could vary?

All three rating sliders were continuous and participants could move the handle to adjust their self, other and control ratings with the same degree of granularity. This means that control could be rated on 101 possible values between (and including) 0 and 1. The reason why we transformed control ratings into values between 0 and 1 was to aid analyses where it was beneficial to understand control as a

weight or percentage value which can be multiplied with either performance. To clarify that control was rated continuously, we added the following to our manuscript:

Methods, “Experimental Design”, lines 876-881

Ratings. [...] Self and other was rated between 0 and 100, labelled as ‘very bad’ and ‘very good’ respectively. Control was rated (or, in the Self-Other phase, shown) on a scale between 0 and 1, ranging from ‘only other’ to ‘only me’. All three rating sliders were continuous with no numerical values shown on the scales. Participants could position the rating handle on any value between ‘very bad’ and ‘very good’ for self and other, and ‘only other’ and ‘only me’ for control.

How consistent were the reported effects across the 4 different games/blocks completed? For example, did participants tendency to adopt the AD strategy increase over time? Was there any evidence of meta-learning?

We addressed this set of questions by examining whether participants’ rating accuracies at the end of each task phase and the proportion of AD trials changed across task blocks. For this, we extracted participants’ rating accuracies for self, other and control on the final trial for each task block (self only for Self-Other phase, control only for Control-Other phase). We then ran a set of ANOVAs predicting self, other and control accuracies by task block (1 to 4) as within and sample (online or MRI) as between-participant factors. For AD, we computed the proportion of AD trials for each task block per participant and then ran an ANOVA predicting the proportion of AD trials by task block (1 to 4) as within and sample (online or MRI) as between-participant factors. We found a significant main effect of task block on the rating accuracies of control, self and other (effect of task block on: control rating accuracy, $F(3,195)=11.36$, $p<0.001$; other rating accuracy, $F(3,195)=35.98$, $p<0.001$; self rating accuracy, $F(3,195)=59.50$, $p<0.001$; panel A below). Post-hoc paired t-tests comparing the accuracies from the first to the last task block revealed that participants got better at inferring their control and their own performance, but worse at learning about the other player (control, $t(66)=4.27$, $p<0.001$; self, $t(66)=10.38$, $p<0.001$; other, $t(66)=-5.57$, $p<0.001$). In particular, the apparent steady increase of control rating accuracies from task block to task block, also compared to the noisier fluctuations of other and self accuracies, might suggest that control inference improved over the course of the task and hint at some sort of meta-learning. We did not find that the proportion of AD differed between task blocks (effect of task block on proportion of AD trials, $F(3,195)=0.15$, $p=0.93$, panel B below), suggesting that participants did not change their AD strategy use over time.

We included these results in our manuscript as follows.

Supplementary Figure 11

Supplementary Figure 11. Rating accuracies and AD over the course of the experiment. A) We investigated whether participants got better at learning about their own or the other player’s performances over the course of the experiment. We found a significant main effect of task block on the rating accuracies of control, self and other (effect of task block on: control rating accuracy, $F(3,195)=11.36$, $p<0.001$; other rating accuracy, $F(3,195)=35.98$, $p<0.001$; self rating accuracy, $F(3,195)=59.50$, $p<0.001$). Post-hoc paired t-tests comparing the accuracies from the first to the last task block revealed that participants got better at inferring their control and their own performance, but worse at learning about the other player (control, $t(66)=4.27$, $p<0.001$; self, $t(66)=10.38$, $p<0.001$; other, $t(66)=-5.57$, $p<0.001$). In particular, the apparent steady increase of control rating accuracies from task block to task block, also compared to the noisier fluctuations of other and self accuracies, might suggest that control inference improved over the course of the task. Data from MRI and online sample are combined here. **B)** We did not find that the proportion of AD differed between task blocks (effect of task block on proportion of AD trials, $F(3,195)=0.15$, $p=0.93$), suggesting that participants did not change their AD use over time. Data from online and MRI samples are pooled here.

Supplementary Methods, Behavioural analyses, lines 482-490

Rating accuracies and AD over the course of the experiment. We examined whether people got better at the task over the course of the experiment. For this, firstly, we extracted their rating accuracies from the final trials of each task phase and task block (self from Self-Other phase only, control from Control-Other phase only, other averaged across both task phases). We then ran a set of ANOVAs predicting self, other and control accuracies by task block (1 to 4) as within and sample (online or MRI) as between-participant factors. Secondly, we computed the proportion of AD trials for each person across Self-Other and Control-Other phases in each task block. We then ran an ANOVA predicting the proportion of AD trials by task block (1 to 4) as within and sample (online or MRI) as between-participant factors.

Fig 3B – does this refer to the entire sample (MRI + online)?

Figure 3 includes data from both MRI and online sample, highlighted with different visuals. Panel B only shows a schematic of our hypothesis. In panel C, the online sample is plotted as dots and the MRI sample as bars. Panel D differentiates the two samples in different colours (MRI in yellow, online in grey). Panels E and F shows the online sample in light yellow and the MRI sample in dark yellow. To clarify the included samples and following formatting instructions, we added information to the included participant numbers in the legends of each figure.

Typing error in line 555 – SMS instead of SMG

Thank you for highlighting this. We have corrected the typo in the text.

Please clarify this statement, we felt it was unclear: Line 154 – “In the following, we will refer to control from the perspective of the participant (e.g. 20% control means the participant and the other control 20% and 80% of the feedback respectively)”

We rephrased the statement as follows to improve clarity (lines 157-162)

Note that control was a single parameter reflecting the balance between participants' own and the other's influence over outcomes (other control=1-self control). In the following, when we refer to control, we refer to the level of control the participant exerts (self control). For example, a level of control of 20% means that the participant's performance determines 20% of the feedback, while the remaining 80% of the feedback is determined by the other player's performance.

Just out of curiosity, to what extent do the results and conclusions revealed by the current study generalize to (more naturalistic) collaborative gaming set-ups with more than 2 players, in which multiple sources that can influence the outcomes – similar to real-world scenarios with team/group work with more than a pair of interactions.

This is an interesting idea and this would be an interesting follow-up study. We hypothesize that in such social groups, people might still engage in AD – and potentially, they might do so even more because if more than 2 people work towards one joint goal, there are more other people to 'pick up the slack' if one person is not contributing. In addition, with more than one collaborator, it becomes more difficult to discern one's own contribution to a joint project than it is with just one collaborator (as is the case in our study). We included these considerations into our discussion as potential follow-up work, as follows.

Discussion, lines 757-761

Furthermore, future studies should test whether our findings on AD behaviour generalise beyond dyadic interactions to larger social groups. It might be that AD-guided control inference is even more prevalent in larger groups, where the diffusion of responsibility can be greater and disentangling one's own contribution from that of others is more difficult.

Reviewer #2 (Remarks to the Author):

We thank the reviewer for their evaluation of our manuscript. We have revised our manuscript and responded to their combined comments above.

Reviewer #3 (Remarks to the Author):

In this paper, Spiering and colleagues describe a combination of behavioural experiments, computational modelling and neuroimaging aimed at unravelling the processes involved in tracking responsibility during collaborative interactions. To this end, they design a novel task that involves people playing simple games along with a ‘partner’, while they receive ambiguous feedback that signals a mixture of their performance and the performance of the partner. Critically, participants are explicitly tasked with estimating the performance level of each player, and the relative contribution each player makes to the feedback that is received.

First, the authors find a characteristic form of behaviour which they call ‘active disambiguation’ – where participants seemingly make active errors in the task, but only when such errors would be helpful to disaggregate between self and other performance and control. Moreover, they show that important behavioural signatures from the task can only be captured by a variant of a Bayesian learning model that learns from these active disambiguation episodes appropriately (i.e., which represents them as active mistakes). This allows participants to learn using the difference between the typical feedback, and the feedback that arrives when agents ‘hypothesis test’ via the errors they make.

Finally, in an fMRI experiment, the authors find a common coding of (model-based) uncertainty and these ‘active disambiguation’ behaviours in SMG. They show, moreover, that SMG contains a range of signals needed for inferences about responsibility – including estimates of control, the feedback difference and prediction errors about self and other performance.

In generally, I thought the paradigm was rich and the paper was technically impressive – with a range of clearly thought out analyses mapping out the component parts of the computational model implied by the behavioural data onto activity in the SMG. However, I was still left with a general sense that – despite these virtues – the theoretical claims made from the task were slight exaggerated. In particular, it was not so obvious to me that these results tell us much about ‘control’ in the context of ‘collaboration’. This may mean significant reframing of the paper is necessary, particular if it is to be situated within the way these concepts are typically studied and understood by other psychologists and neuroscientists.

We thank the reviewer for their insightful and considerate feedback, especially for highlighting that our “paradigm was rich” and that our manuscript “was technically impressive – with a range of clearly thought out analyses”. We understand that the reviewer has some concerns regarding the framing our findings in the context of control and collaboration and we aim to address these in our responses below. We highlight our changes to the manuscript in red.

Major comments

1) As noted above, it is clear that this manuscript is impressive in its technical richness. But from the outset, I had some general concerns about how far these impressive methods can be said to be targeting the cognitive and mental functions implied in the title, introduction and discussion.

First, the paper is largely framed as being about how we determine what we can and cannot 'control' in joint action settings. In psychology, such joint actions have typically been studied in contexts where people collaboratively coordinate to control the same object (e.g., Knoblich and Sebanz's work on controlling the spatiotemporal trajectory of a single object together). Here though, the outcome being controlled is rather more nebulous – as it relates to the 'feedback' that occurs at the end of the trial. Naturally, participants in the task are encouraged to think about the task by asking 'how much did you *control* that feedback?' but any equally plausible framing could be 'how much did that feedback *relate* to your action?' or 'how much were you *responsible* for the feedback that appeared on screen?'

I raise these issues not to try and split terminological hairs, but to drill into the connection between the processes studied here and the broader literature on 'control' and 'controllability'. For instance, many theoretical models (including those that implicate neural regions like SMG) suggest that personal and joint feelings of control are generated by monitoring sensorimotor processes that are engaged as we create actions and outcomes ensue – and it is very hard to see how such processes could be deployed in the very clever but rather artificial design used here. And this may become relevant to some of the connections the authors make to sense of agency, schizophrenia etc. where a sensorimotor monitoring deficit seems implicated.

It may be that the cognitive and neural mechanisms described here tell us more about cognitions of 'responsibility' rather than feelings of 'control' – which may be important for adjusting the paper's framing and/or considering which literatures the paper does and does not seek to connect to. I would be interested to know what the authors think.

Thank you for your constructive comment and for highlighting the distinction between the concept of control in our manuscript and how it links to some of the other literature. In this study, we studied interpersonal control and we agree with the reviewer that this can also be understood as the level of responsibility one has over joint outcomes with other people. To make our understanding of the term control and how we use it throughout the manuscript clearer for the reader, we have now included a clearer definition of the control problem we studied in our introduction (as also detailed in a response to the other reviewer).

In addition to the comparator model (Frith et al., 2000), it has been highlighted that sense of agency or control can be understood on different levels such as the sensorimotor and the cognitive level (referred to as implicit 'feeling of agency' and explicit 'judgement of agency', respectively, by Synofzik et al., 2008, 2013). Indeed, this can be translated from individual to joint actions in social interactions (Zapparoli et al., 2022). While some research focussed more on the cognitive mechanisms of feeling in control (e.g. Ligneul et al., 2022; Moscarello & Hartley, 2017), others examined the intersection with the sensorimotor level (e.g. Dewey & Knoblich, 2014; Sebanz & Knoblich, 2006). In psychiatric conditions, control is also used as a term to study how responsible people feel for outcomes. For example, negative attributional styles in depression describe that depressed people tend to feel a heightened perceived control for negative life events, and are more sensitive to having no or a low level of control over neutral events (depressive realism; Alloy & Abramson, 1979; Benassi & Mahler, 1985). In the learned helplessness theory, Maier and Seligman (1976) explicitly proposed that being in control

means a higher outcome likelihood in the presence of an action than in the absence of an action (a cognitive framework which has since been extended in various ways, e.g. by Ligneul 2021).

In our task, we might be tapping into both the cognitive and sensorimotor aspects of control because while we ask participants to judge or rate their level of control, we also disrupt their sensorimotor experience by withholding direct perceptual feedback for their game performance (by hiding the ball's path and by transforming their objective performance into a self-performance score; as also highlighted by the reviewer in a comment below). As highlighted by the reviewer, our neural results appear coherent with this notion because they highlight the role of the SMG, a region not just involved in motor attention (e.g. Rushworth et al., 2001) and representing movements in terms of their purpose and meaning (e.g. Desmurget et al., 2009), but also action attributions and agency judgments in social contexts (e.g. Farrer & Frith, 2002; Ohata et al., 2020). For these reasons, we believe that our study is relevant for researchers working on the sensorimotor and the more cognitive level of controllability.

We believe that the reviewer highlights an important aspect and we made the following changes to our manuscript introduction and discussion to address their concerns about the framing of our study.

Introduction, lines 83-89

Controllability is closely linked to constructs such as sense of agency (Farrer & Frith, 2002; Haggard, 2017; Moscarello & Hartley, 2017; Ohata et al., 2020) and perceived self-efficacy (Bandura, 1982; Zorowitz et al., 2020). Here, we examine interpersonal control, that is how people come to understand their level of responsibility over outcomes in social interactions. A key aspect of establishing control is establishing whether outcomes are contingent on our actions over and above any contingency that already exists between outcomes and states of the environment (Ligneul, 2021; Ligneul et al., 2022). Analogously, establishing control in social situations requires the establishment of each individual's degree of agency relative to one another. [...]

Discussion, lines 731-747

In this study, we studied interpersonal control of joint task outcomes – how well is a task performed together by two people. A complementary approach has been to ask how people coordinate the actions they each make with one another in order to move a particular physical object (Sebanz and Knoblich, 2021). Typically the focus in such studies has been on how people learn to predict each others' actions so that they can be coordinated together. However, a crucial element of such coordination is ascertaining whether joint goals are accomplished and the focus of our current study can be thought of as understanding how we learn the nature and degree of our contribution to the accomplishment of a joint goal. It has been noted that the sense of agency or control can be understood on different levels such as the sensorimotor and the cognitive level (referred to as implicit 'feeling of agency' and explicit 'judgement of agency', respectively, by Synofzik et al, 2008, 2013) and this can be translated from individual to joint actions in social interactions (Zapparoli et al., 2022). Again different research strands have tended to focus on either the cognitive mechanisms of feeling in control (e.g. Ligneul et al., 2022; Moscarello & Hartley, 2017) or control at a more sensorimotor level (e.g. Dewey & Knoblich, 2014; Sebanz & Knoblich, 2006). In psychiatric conditions, control is also used as a term to study how responsible people feel for outcomes. For example, negative attributional styles in depression describe how depressed people tend to feel a heightened perception of have exerted control over negative life events but no or a low level of control over neutral events (depressive realism; Alloy & Abramson, 1979; Benassi & Mahler, 1985).

2) I have similar concerns about framing this experimental task as a study of ‘collaboration’. Most real and experimental studies of collaborative decision making necessarily involve two or more agents working towards the same goal (e.g., work from Bahrami and colleagues examining how two people judge the same stimulus, and arrive at an accurate answer together). However, in this case the participant’s task is explicitly *not* to maximise joint outcomes in the task, but to make accurate ratings about the self and other. I feel this somewhat weakens how far this study tells us much about how we ‘collaborate’ and ‘cooperate’ with others. For instance, the paper claims in the abstract that that ‘people engage in specific patterns of behaviour’ in ambiguous social settings – and this is mediated through a particular set of brain regions. But it seems this may be a little overstated, since such ‘active disambiguation’ might only occur when we are given the task ‘work out what you are responsible for’ rather than this being a common feature of our collaborative joint interactions. It strikes me it would be stronger if these behavioural, computational or neural signatures could also be generated in a variant of the task where participants have an incentive to work towards the joint outcome (i.e., maximizing feedback outcomes). I would be interested to know if the authors think these effects tell us much about ‘collaboration’ or whether they tell us more straightforwardly about mechanisms we use to assess responsibility when this is the task we have been given.

Thank you for this comment. We indeed believe that our task measures collaboration because participants had to work with the various other players towards the joint outcomes (feedback). This is a fundamental feature of our task design and participants are indeed not able to freely decide when to work in isolation and when to cooperate with the other player. The results we find (occurrence of AD, assigning prediction errors to self and other, learning about control; Figures 2-4) are present at the times in our task when participants need to work together with the other players. We concur with the reviewer that we did not give participants monetary rewards (bonus payment) for maximising the feedback, but instead for assigning credit to themselves, others and their control (i.e. doing the ratings well). However, we believe that this is a strength rather than a weakness of our study design because it allows disentangling the process of credit assignment from the process of maximising collaborative outcomes. Against this background, it is very interesting that we found that the ventral striatum, a classic reward structure in the brain, tracks the feedback on every trial, and whether it is better or worse than on the trial before (Supplementary Figure 17). We concur with the reviewer that for future studies, it would be an interesting extension of our current work to test whether participants engage in AD when they are being rewarded for both their ratings and their game performance. We hypothesize that in such circumstances, people will employ AD more sparsely and efficiently, in order to maximise reward for game performance and correct ratings.

Furthermore, in our task, we used a cover story which instructed participants that they were team captains and they had to find out their own and the other player's performances and their level of control, so that at the end of each game (task block), they could select a teammate and distribute their control between themselves and their teammates. Note that we did not analyse participants’ team selection ratings at the end of each task block because they did not relate to our main research questions. However, we believe that this framing makes our findings relevant for collaborative behaviours. Coming back the example of our introduction, if a student works together with a collaborator on a research project and receives worse feedback than she expected, then she first needs to find out the reason for the suboptimal feedback in order to know how to improve the project. This means that causal inference is an important feature of collaborative behaviours and in our study, we investigate how humans do so in a lab experiment. It is true that for the purpose of our lab experiment, we detected AD as extreme and obvious performance deviations which might appear artificial. In real-life, while not repeatedly not turning up for a project meeting would probably lead to severe negative effects, we believe that strategically and slightly varying how much effort to exert towards joint outcomes, can help establish how much one’s own actions matter in social contexts.

We incorporate this comment by the reviewer in our discussion as follows (together with another comment made by the other reviewer about extending our research to group settings)

Discussion, lines 749-761

Here, we studied how humans assign credit and infer their control in collaborative settings. We rewarded participants only for how well they assigned credit. This allowed us to delineate the process of credit assignment from that of maximising collaborative outcomes. It is interesting that nevertheless, we found feedback-related signals in the ventral striatum reflecting how well the games were performed on each trial. Previous work, however, has investigated how humans decide to collaborate to do better (Bahrami et al., 2010). Future work can investigate the degree to which our findings change when participants are rewarded not just for assigning credit correctly, but also for maximising the joint feedback they receive as a result of game performance. This might lead to a sparser but more efficient use of AD to guide learning. Furthermore, future studies should test whether our findings on AD behaviour generalise beyond dyadic interactions to larger social groups. It might be that AD-guided control inference is even more prevalent in larger groups, where the diffusion of responsibility can be greater and disentangling one's own contribution from that of others is more difficult.

3) In the ROI-based analyses, I did not understand the rationale behind limiting analyses to small spheres drawn around the peak effects in the whole brain analysis rather than e.g., all the voxels active in the whole-brain cluster, or those voxels passing some other activation threshold. Are any effects changed if the regions of interest are drawn less narrowly?

We thank the reviewer for this comment. In the past, most studies using fMRI time course analyses have been based on small spherical ROIs (e.g. Khalighinejad et al., 2021; Murphy et al., 2018; Scholl et al., 2015, 2017; Trudel et al., 2021; Wittmann et al., 2016). The rationale for using a sphere centred around the peak activation (obtained from the whole-brain analysis) is that such an area shows the most consistent effect across participants. With increasing distance from this area, interindividual differences in anatomy and registration increase, making effects become more variable and unreliable. The location with the strongest effect in most people, is therefore usually a good candidate region to test effects especially when they are orthogonal to the whole-brain contrast that was employed when the area of activity was first employed in the initial whole brain analysis. Indeed, spherical ROIs with a 3 voxel radius (like in our study) are typically used to study subcortical regions such as the amygdala (e.g. Scholl et al., 2015, 2017) and cortical regions such as vmPFC or pgACC (e.g. Trudel et al., 2021; Wittmann et al., 2025). The SPM RFX toolbox suggests a similar ROI sphere size of 6 mm for fMRI time course analyses (Gläscher, 2009; 336 citations to date; used by e.g. Paret et al., 2014). For these reasons and to avoid confusion about introducing a novel analysis approach, we followed convention and would prefer not to deviate from it in our study.

We included this information about this convention in the methods as follows.

Methods, fMRI time course analysis on ROIs, lines 1395-1396

ROI selection and time series extraction. Similar to previous work (Khalighinejad et al., 2021; Scholl et al., 2015; Scholl et al., 2017; Trudel et al., 2021; Wittmann et al., 2016), spherical ROIs with a radius of three voxels (5 mm) were constructed based on significant activation clusters in the whole-brain analysis. [...]

Minor comments

4) Page 19 lists N = 66 online participants in the final sample, while p.4 lists N = 36. Which is correct?

Thank you for highlighting this inconsistency. We clarified this as follows in our methods section.

Methods, Participants, lines 778-794

36 healthy participants took part in the fMRI experiment (sample taken from a larger study, see supplementary methods), and 69 participants took part in the online experiment. 2 MRI participants were excluded because they misunderstood how they should rate their own performance in the games (they rated their previous trial's performance rather than their overall best attempt, meaning that they rated their performance lower after an AD trial; one of these two participants reported having changed how they did their self-ratings midway through the experiment). These misunderstandings were prevented in the participants collected afterwards by an additional task-understanding question administered by the experimenter. 3 additional MRI participants were excluded because they performed fewer than 3 active disambiguation trials in the Control-Other phase. For the online sample, 3 participants were excluded because they took breaks longer than 5 min during the task blocks, and 9 participants were excluded because they misunderstood how to rate their own performance. In addition and for almost all of our analyses (except the correlation analysis shown in Figure 3d), we excluded 30 online participants who did not perform AD in the task or reported never having done AD in the post-task debrief questionnaire. The final samples included 31 MRI participants (self-reported gender: 22 female, aged 18–33 years), and 36 online participants (self-reported gender: 21 female, 1 diverse, aged 18–40 years).

5) There are regular references to the Bayesian learner models being 'optimal'. Then, in other cases, the 'passive' and 'ignorant' learner models are described as being 'suboptimal'. I know what the authors mean when they talk about Bayes optimality, but I worry that this has the potential confuse a reader given that (from a Bayesian perspective) even the suboptimal learners are optimal, given their architecture! For instance, the paper contains sentences like "...this suggests that participants, like a Bayesian optimal observer, attribute social PEs in accordance with their controllability" (p.7). This is potentially confusing, as to a general reader this may imply that the observer is optimal *because* it is doing this kind of controllability weighting. When in practice the authors evaluate other models that are equally (Bayes) optimal which don't show this signature.

It didn't strike me that any of the paper's conclusions really rest on human behaviour being Bayes optimal in any case, so the paper may be more easily digested if the authors simply speak of 'Bayesian learners' rather than 'optimal Bayesian learners' throughout. e.g. sentences like that above would become "this suggests that participants attribute social PEs in accordance with controllability, just as the winning model does".

Thank you for highlighting this. We addressed this by substituting our mentions of e.g. optimal Bayesian learning models by Active learner or Active learner model throughout. Furthermore, we removed mentions of the ignorant and passive learning models being suboptimal to avoid confusion. As suggested by the reviewer, we also removed mentions of 'optimal Bayesian learners' and instead say 'Bayesian learners'. Below, we highlight some of our changes that hopefully clarify our findings.

Results, "Estimates of controllability modulate social prediction errors", lines 252-256

We found consistent effects in simulated data from the **Active learning model** [...]. This suggests that participants, **consistent with an Active learning model**, attribute social PEs in accordance with their controllability.

Results, “Active disambiguation enables inference of controllability”, lines 396-400

To test whether the behaviour observed is indeed the behaviour expected from a learner that does perform AD trials and that knows that AD trials were performed (rather than assumes that a trial reflects its best performance), we compared participants’ behaviour to the **Active learner** (introduced before) and two **alternative Bayesian learning models** we refer to as the “Ignorant” and the “Passive” learners (**Supplementary Figure 6**).

Methods, “Behavioural analyses”, “Credit assignment in the Self-Other phase”, lines 1174-1175

This was fit separately to the updates of self and other ratings of the participant, as well as the respective estimates from the **Active learning model**.

Supplementals, Caption text of Supplementary Figure 9

B) For inferring self performance in the Self-Other phase, we found that participant performance neither resembled the **Active learner** (main effect of inverse control), nor the **Passive or Ignorant learning models** (inverse control and interaction effect). [...] **C)** In contrast, participants inferred the other’s performance similarly to the **Active learner** in the Self-Other phase. Here, the interaction effect AD feedback * Inverse other’s control differentiated between, on one hand, participants and the active learner, and on the other hand, the two **alternative learning models (which make wrong assumptions about AD)**. [...] **D)** The interaction effect from C shows that both participants and the **Active learner** inferred that the other had a low performance if the AD feedback is low, irrespective of the other’s control. [...] **E)** In the Control-Other phase, we found that participants were not able to infer the other’s performance like the **Active learner** (see interaction effect). [...]

6) In some ways, this work reminds me of work by Pescetelli & Yeung (2021) on the advice-taking and trust. Here the authors show that agents sometimes seek advice from advisors even when they are very confident in their answers – which may seem suboptimal, but in fact allows agents to learn about the quality of the advice they are being given (as they know they know the correct answer). It seems an essential feature of the present task is to disrupt possible introspective metacognitive feelings by distorting the feedback that participants receive so it is not on a straightforwardly objective scale. To what extent do the authors see any connection between the kind of ‘pointless’ advice taking behaviour described by Pescetelli & Yeung (2021) and the ‘purposive errors’ described in their work?

We thank the reviewer for highlighting this work and the potential parallels. We agree that on an abstract level, there are some similarities between AD (or deliberate mistakes) in our study and the pointless advice taking in Pescetelli & Yeung (2021). Both behaviours appear to be suboptimal at first glance because in our study, AD leads to worse feedback, and in Pescetelli & Yeung, taking advice when already being confident appears unnecessary. However, both can actually help people learn on the long run; AD helps reduce uncertainty and learn about controllability (our study) while pointless advice taking enables learning about the quality of the advice given (Pescetelli & Yeung). We incorporated this in our discussion in a new paragraph in which we address the parallels of AD-guided control inference to other prior work.

Discussion, lines 687-700

At the core of control inference studied here is a comparison process and prior work highlighted that control can be inferred through comparing either observed and predicted action outcomes (Frith et al., 2000) or the likelihoods of outcomes in the presence or absence of one’s actions (Allan, 1980; Ligneul,

2021; Ligneul et al., 2022; Maier & Seligman, 1976). Ligneul et al. studied how people track their control over categorical outcomes by comparing a ‘spectator’ and ‘actor’ model in non-social contexts. Here, we show that a similar comparison process is at play in social situations with ambiguous and parametric joint outcomes, but that it additionally requires comparing outcomes caused by a range of actions, including the strategic use of poor or self-defeating actions. These differences likely account for our divergent neural findings, with our results implicating the SMG, in contrast to the prefrontal regions reported by Ligneul. Engaging in exploratory actions also allows sensing one’s control over different objects in non-social contexts (Wen et al., 2020). On an abstract level, there are some similarities of this AD-guided control inference to prior work showing that people take advice despite already being highly confident (Pescetelli & Yeung, 2021). While initially appearing suboptimal, both behaviours are beneficial for long-term learning about the quality of advice (Pescetelli & Yeung, 2021) or controllability.

7) Related to the last point – do the authors think it would it be possible to solve the present task without active information seeking ‘errors’ if introspection was not actively disrupted? I.e., can metacognitive appraisals of personal correctness play the same role of disaggregating responsibility in settings where these signals are not actively corrupted?

This is an interesting suggestion. In our task, we disrupted participants’ introspection into their performance in two ways, 1) by changing the mapping of their objective game performance to a self-performance point score (see Supplementary Figure 2), and 2) by placing a big black box around the target which hides the ball’s path (in the example of the rock slide game, see Figure 1b and Supplementary Figure 1). Notably, these two manipulations disrupted participants’ insight into their own performance, not their control. Indeed, control only acted on the feedback because it determined the weighting between self and other performance. We found that participants used AD most to learn about their level of control over the feedback and also reported doing so (Figure 4, Supplementary Figure 9f). Therefore, we hypothesize that if the task didn’t involve changing their performance introspection, people would still use AD to solve the task.

In the Discussion and Methods we have now included the following comments:

Discussion, lines 702-706

In the current task, participants’ metacognitive introspection was disrupted by the manner in which objective performance was mapped to a point score and by features of the stimulus display that obscured aspects of the ball’s path. However, in more natural settings it is likely that metacognitive assessment of performance might also contribute to a person’s estimate of control.

Methods, Experimental design, Mapping of objective game performance to self-performance scores, lines 938-941

This mapping procedure means that we disrupted participants’ metacognitive assessment of their performance. This mirrors real-world situations where, for example, taking a certain number of seconds to run 50 m can only be interpreted as ‘good’ or ‘bad’ performance when compared to others (or one’s own previous performance).

8) The authors may consider making reference to work by Wen Wen on the active sensing of control difference (e.g., Wen et al (2020), iScience). This work has already established that agents actively manipulate the actions they perform to exaggerate the differences between (and thus discriminate between) the things they can and cannot control – analogously to what is achieved in the present work (though not in a social setting).

We thank the reviewer for raising this. There are indeed some parallels between the exploratory actions studied by Wen and colleagues, and the AD behaviour we identified, although our AD process highlights the need for apparently drastic and ‘self-sabotaging’ exploration. We refer to Wen et al.’s study in the discussion as follows, together with the previously mentioned work by e.g. Pescetelli & Young (see above).

Discussion, line 687-700

At the core of control inference studied here is a comparison process and prior work highlighted that control can be inferred through comparing either observed and predicted action outcomes (Frith et al., 2000) or the likelihoods of outcomes in the presence or absence of one’s actions (Allan, 1980; Ligneul, 2021; Ligneul et al., 2022; Maier & Seligman, 1976). Ligneul et al. studied how people track their control over categorical outcomes by comparing a ‘spectator’ and ‘actor’ model in non-social contexts. Here, we show that a similar comparison process is at play in social situations with ambiguous and parametric joint outcomes, but that it additionally requires comparing outcomes caused by a range of actions, including the strategic use of poor or self-defeating actions. These differences likely account for our divergent neural findings, with our results implicating the SMG, in contrast to the prefrontal regions reported by Ligneul. Engaging in exploratory actions also allows sensing one’s control over different objects in non-social contexts (Wen et al., 2020). On an abstract level, there are some similarities of this AD-guided control inference to prior work showing that people take advice despite already being highly confident (Pescetelli & Yeung, 2021). While initially appearing suboptimal, both behaviours are beneficial for long-term learning about the quality of advice (Pescetelli & Yeung, 2021) or controllability.

REVIEWER COMMENTS

Reviewer #1 (Remarks to the Author):

The authors have gone out of their way to address our comments in great detail and have addressed some of our comments. For example, we agree that the term action efficacy might create further confusion.

There is a lot to like about this paper, and the demonstration that participants recruit an Active Disambiguation strategy and the supramarginal gyrus to resolve uncertainty is valuable and the evidence for this is solid.

We thank the reviewer for their positive feedback and highlighting that we “have gone out of our way to address the comments in great detail”. We indeed did our best to incorporate the reviewer feedback and feel that the manuscript is much improved. We understand that the reviewer has follow-up comments on our response to their previous comments. Below, we address these in a point-by-point response. New changes in our manuscript are highlighted in blue, while previous changes from the last revision are kept in red.

However, we remain puzzled by key parts of the revised paper and the rebuttal.

The first point we struggle with is the assertion that feedback can be highly predictable, even if there is great uncertainty about the causes of the feedback. We can see from Fig 2d that estimated feedback/rating accuracy increases over the course of a phase. Accordingly, feedback predictability seems to increase in a manner that is proportional to the reduction of uncertainty about who is in charge, over the course of a phase. In line with this, the new paragraph regarding distinguishing between uncertainty and unpredictability is hard to parse. If something is uncertain, it is by definition also unpredictable. The key thing that I think the authors want to highlight is the nature of the thing that is uncertain/unpredictable. I think the authors would like to state: “In our task, we distinguish between uncertainty about (=unpredictability of) which agent is causing the outcomes and uncertainty about (= unpredictability of) the outcomes themselves (i.e., about how good the joint performance is).“

Importantly, I am not sure there is evidence for this distinction. Indeed, the additional supplementary analyses presented in the rebuttal (point 1, page 11) demonstrate that ADs are not uniquely predicted by uncertainty about control, but also by uncertainty about other-performance. This is not surprising, because participants exhibit high AD rates in the first self-other phase, in which the level of control is instructed. A key implication of this observation is that the conclusion that ‘active disambiguation enables inference of controllability’ does not hold in the unconditional/generalized form that is suggested by this paper. What the paper shows is that AD enables inference of controllability if and only if controllability is the thing that is ambiguous, as is the case in the third phase of the current experiment. Conversely, if self- and/or other-performance is the thing that is ambiguous, as is the case in the first phase of the current experiment, then ADs are triggered by uncertainty about performance (about the feedback itself). In fact, the data from the first phase might well be analyzed to establish that ADs predict belief updates about e.g. other-performance in the 1st phase, just like they predict belief updates about controllability in the 3rd phase. By analogy, the conclusion that the ‘SMG carries neural signals for learning controllability’ is likely also invalid in its generalized/unconditional form, as demonstrated by the analyses of data from the first phase of the current experiment, in which the SGM tracks a learning signal that is not used for controllability inference (because the level of control is already instructed) but instead for inferring the quality of own or other’s performance levels.

We also note that the more conditional inference that AD enables controllability inference in the third phase where controllability is the thing that is ambiguous seems to depend on the additional analysis reported in the rebuttal and the supplement, showing that AD predicts the difference between uncertainty about control and that about other-performance.

Based on these observations, we recommend that the authors revise their paper to abstain from concluding a key role for AD and SMG in controllability inference specifically. Instead, we suggest that the authors highlight that, based on the present data/results, they can conclude that people recruit an active disambiguation strategy in proportion to uncertainty (I assume also in proportion to tPE?) and associated uncertainty- (and tPE-?)related activity in the SMG, for resolving uncertainty. Crucially, this AD strategy and associated SMG activity seem to be recruited not only for resolving uncertainty about who is in charge (controllability), but also for resolving uncertainty about the quality of (e.g. joint, other, self) performance, and probably also for other types of uncertainty depending on the task at hand. The paper seems to capture a more general uncertainty resolution construct than the one that is implied by the current version of the paper. Note that this still represents a very valuable contribution.

We concur that the new paragraph about uncertainty and unpredictability, which we included in the last revision in order to add clarity, ended up leading to more confusion. We have therefore now removed this paragraph (previously in lines 1218-1229) to avoid making claims about uncertainty vs unpredictability which might not have been as helpful as we had hoped.

We agree with the reviewer that in our study, AD is triggered not just by uncertainty about control but also by uncertainty about the performances of self and other, which is why we observed AD not just in the Control-Other phase but also the Self-Other phase (Figure 3c).

However, with regard to the reviewers' next question as to whether SMG activity reflects not just uncertainty but also tPE processing, we believe that this is not the case. While we found SMG activity reflects uncertainty (Figure 5a), we did not find that it tracks the tPE in our task (Figure 7c). Instead, when control is known and performance is uncertain (Self-Other phase), SMG carries PE signals in an agent-based manner; it does so for both the self-related PE and other-related PE (Figure 7a-b). We believe that together with our results on SMG involvement in control inference when control is unknown (Figure 6), this is a very interesting set of analyses.

In summary, we agree with the reviewer that these results suggest that, depending on the current set of uncertain latent causes, the SMG is not just involved in AD-guided control inference but also involved in learning about self and other performance. In other scenarios, the SMG might also be involved in tracking uncertainties about features not captured by the task we developed. We understand that the reviewer suggests that we change our conclusions to acknowledge this more explicitly and highlight the role of SMG in uncertainty tracking, and we aim to do so with the various changes to our manuscript below.

Abstract, page 1

A pattern of activity in the supramarginal gyrus that emerges during and after active disambiguation is linked to tracking uncertainty and establishing controllability. We show that activity in this brain region signals a second learning mechanism, by which individuals attribute outcomes to themselves versus others, in proportion to their perceived control.

Introduction, page 3

AD was driven by uncertainty, and in turn reduced uncertainty and led to better credit assignment. Thirdly, we established that when controllability was uncertain, AD mediated inference about control.

Results, pages 10-11

Active disambiguation supports inference of controllability

[...]

We reasoned that AD trials provided information important for inferring controllability when it is ambiguous, in the following way.

[...]

This suggests that when control was uncertain, participants used AD efficiently for learning about their control and that they had insight into this.

[...]

In sum, when control was ambiguous, participants used the information afforded from AD in order to infer their level of control over the feedback.

Results, page 14

In summary, when control was ambiguous, SMG signalled all component variables needed for control inference, including the observed feedback difference between AD and normal trials, the self-performance estimate, and the updating of the control estimate at the time of the outcome.

Results, page 17

They had insight into AD, and it allowed them to reduce uncertainty and assign credit more accurately. When control was unknown, they employed this behavioural strategy to inform their control estimates, by comparing outcomes obtained through AD and normal performance.

Discussion, page 18

Our findings suggest that AD is recruited as a general strategy to resolve uncertainty and aid credit assignment, and that SMG activity reflects this process. In the current task, AD supported inference of controllability specifically when control was ambiguous, but it was also active when there was uncertainty was about self- or other-performance levels. Thus, our results point to a broader contribution of both AD and SMG to uncertainty resolution across different dimensions of social credit assignment.

Regarding our second major comment, we did not fully understand the speculation about the putative role of exploratory actions to infer control in the neural activity revealed by the current study vs Ligneul et al. In that prior work (which highlights a role for medial frontal cortex), controllability inference is equally dependent on exploratory, ‘counterfactual’ actions (see also Ligneul, 2021). Indeed, without such exploration, spectator and actor predictions are perfectly correlated and controllability cannot be inferred. What is the essential difference then between the computations that are studied here, activating SMG, and those non-social computations?

Thank you for highlighting this. We agree that our findings build on and complement the findings on the role of exploratory actions in control inference by Ligneul et al. (2022). We realise that our previous phrasing was not clear enough with regard to the differences between our work and the study by Ligneul and colleagues. However, we believe that there are at least three main differences that could explain them. Firstly, Ligneul and colleagues investigated control inference in a non-social setting while we investigated control inference in a social setting. Secondly, the studies differ in ‘what’ the participants could control. Ligneul used an explore-and-predict task to study how humans estimate their control over categorical and geometric stimuli and their transitions. We studied how humans estimate their control over parametric feedback that reflects their own performance and someone else’s performance. Thirdly, the nature of the exploratory actions differs

between studies. Exploratory actions in Ligneul's study were choices between options while exploratory actions in our task consisted of deliberately performing poorly. We have updated our discussion to more clearly describe the differences that might explain the diverging neural results of our compared to Ligneul's study.

Discussion, page 19

Ligneul et al.²⁷ studied how people track their control over categorical outcomes by comparing a 'spectator' and 'actor' model in non-social contexts. Here, we show that a similar comparison process involving exploratory actions is at play in social situations. Several differences likely account for our divergent neural findings, with our results implicating the SMG, in contrast to the prefrontal regions reported by Ligneul et al. First, Ligneul et al. studied controllability inference in non-social settings, where the challenge was to establish whether one's own actions influenced state transitions. In contrast, our paradigm involved social contexts, in which participants had to disambiguate their own versus another person's contribution to shared outcomes. Second, Ligneul et al. examined control over categorical, geometric stimuli and their transitions, whereas our task focused on control over parametric feedback reflecting self- and other-performance that also varied in a similarly continuous manner. Third, in Ligneul's study, exploration meant choosing between options to generate informative transitions, while in our task it consisted of deliberately reducing one's own performance to reveal one's own versus someone else's contribution. Taken together, the two lines of work suggest that exploratory behaviour is a general mechanism supporting controllability inference, but that the neural substrates recruited may differ depending on whether the challenge is arbitration between actor-spectator models in non-social contexts or disambiguating of self-other agency in social contexts.

Minor comments:

The figure of the evolution of the various uncertainties across trials is insightful. We suggest to include this in the main paper, and to consider including a line for ADs. Was the paper pre-registered? The task instructions suggest that the project was set up to assess effects of mood. If such analyses performed (and perhaps reported separately), then perhaps report them?

As suggested by the reviewer, we included an example block from Supplementary Figure 8, showing the evolution of uncertainty to the main Figure 3 (new panel e). The mean proportion of AD trials is now included in both the Supplementary Figure 8 and the new panel e of Figure 3. To further illustrate how AD affects trial-wise uncertainty, we additionally included another new panel to our main Figure 3, panel f. This panel shows for an example participant and during the Control-Other phase of one example task block, how the uncertainties are strikingly reduced following AD.

We did not pre-register the study. We have not yet analysed mood ratings as they did not pertain to the research questions of this project. This data is part of a larger data collection effort and the mood effects will be reported in the future once that data is analysed (as mentioned in lines 328-331 in Supplementary materials).

We included the updated Supplementary Figure 8 and the new panels of main Figure 3 as follows. We have also updated the text to reflect the updated panel labels of Figure 3 throughout the manuscript.

Results, Figure 3

Figure 3. Active disambiguation (AD). **a**) We hypothesized that participants made deliberate mistakes in the games to assign credit better. In the game shown here, they could press the button too late to see the difference this makes to the feedback. Whenever participants made such extreme errors, these were transformed into a zero self-performance and detected as AD trials (see also Methods and Supplementary Figure 4). **b**) We hypothesized that participants generated more AD trials under high uncertainty, and that AD trials helped them reduce uncertainty and assign credit better. **c**) Participants indeed generated AD when the feedback was ambiguous. Whenever the feedback was certain, AD dropped strikingly. We found a similar pattern in another online sample who did not receive a task hint to change their game behaviour (Supplementary Figure 5a). Bars and dots show the mean AD proportion. **d**) After the task, participants reported how many AD trials they generated per game (“In the study, how many times per game did you do badly on purpose?”). Their subjective reports of AD trials were significantly correlated with the objective frequency of AD trials identified in their behaviour. Again, these results replicate in another online sample (Supplementary Figure 5b). Dots show individual participants, line and shaded intervals are fitted regression curves with confidence intervals. **e**) Uncertainty is highest when a new phase starts (example block 3). Within each phase, uncertainty increases when the other’s performance changes. **f-g**) Trial-wise uncertainty is reduced if the previous trial or the one before was an AD trial (f shows data from example participant; see Supplementary Figure 8c for full regression results of g). **h**) Crucially, participants’ rating errors are reduced after AD, showing that they can use the additional information afforded by AD (similar to the Active learner). In g-h we did not find significant group differences between samples (all $p > 0.05$). In g-h, bars show mean beta estimates with error bars indicating s.e.m., and grey dots for individual participants. Panel d: $n=29$ MRI and $n=64$ online with otherwise excluded participants (see Methods); Panels c, e, g, h: $n=31$ MRI, $n=36$ online; n.s., not significant; **, $p < 0.01$; ***, $p < 0.001$.

Supplementary Materials, Supplementary Figure 8

Supplementary Figure 8. The dynamics of uncertainty (extracted from the Active learner model). **A)** From the start to the end of a phase, uncertainty reduced (averaged uncertainty, $F(1,65)=7103.74$, $p<0.001$; control uncertainty, $F(1,65)=3807.87$, $p<0.001$; other uncertainty, $F(1,65)=6249.37$, $p<0.001$; self uncertainty, $F(1,65)=10184.59$, $p<0.001$). To test whether uncertainties reduced over time, here we plot the equivalent of Figure 2d but for uncertainties rather than rating accuracies. For each type of uncertainty (averaged, control, other and self) as dependent variable, we ran an ANOVA with *time in phase* ('Start' or 'End') as within, and *sample* (MRI or online) as between participant factors. At the start of each phase, the Active learner had the same wide uncertainty for each unknown variable and each participant. Therefore, each individual data point for 'Start' has the same uncertainty value in the plot and is covered by the bigger data point denoting the mean. Additionally, this plot and Figure 2d show the standard error of the mean, which in both cases is so small that the bigger data point denoting the mean hides the error bars. **B)** Uncertainties plotted for each game block and by task phase, averaged across Active learner model fit to MRI and online sample. While all uncertainties are highest at the start of each task phase, other and therefore averaged uncertainties increased when the other player changed performance unannounced. For reference, yellow dots indicate the proportion of AD trials, averaged across MRI and online samples. **C)** Full results plot of the regression shown in Figure 3g in the main text. (Averaged) uncertainty was reduced on trials following AD (as reported in main text), and uncertainty reduced over time within a phase (trial, $F(1,65)=5813.88$, $p<0.001$, $\eta^2=0.99$). Uncertainty increased when the other's performance changed unannounced during the task phase (other changed within phase, $F(1,65)=1124.28$, $p<0.001$, $\eta^2=0.95$). Averaged uncertainty did not differ between task phases (task phase, $F(1,65)=2.65$, $p=0.11$, $\eta^2=0.04$).

The rebuttal highlighted that 30 out of 69 online participants never made any ADs, and that, for the new, larger online study, again, only 147 of the 352 participants ended up making ADs. I recommend to highlight this observation that around half of the group makes ADs explicitly in the results (not just methods) section, and to address the implication of this finding in the discussion section of the paper.

We agree with the reviewer that this is an interesting feature of the online sample. However, we would like to highlight that this indeed only pertains to the online but not the MRI sample: Every MRI participant performed AD trials. We acknowledge this more explicitly in our results and discussion in the following ways.

Results, page 9

Note that we detected AD in all 31 MRI participants but only in a subset of the online sample (n=36 out of 69 had usable AD data), suggesting differences between lab-based and online data collection.

Discussion, page 20

While all our MRI participants exhibited AD behaviour, only about half of the online sample did so. This could point towards differences between lab-based and online data collection. Alternatively, it could indicate that some individuals may rely on more passive observation to disambiguate causes. It would be an interesting future avenue to characterise the factors influencing why some individuals adopt AD readily while others do not.

There is a lot of redundancy in the figure legends and the text in the results section. This is a matter of taste, but I suspect the legends can be compressed to increase reading load/efficiency.

We have substantially shortened our figure legends following this suggestion (see below). This means that all our figure legends now also follow Nature Communications' formatting guidelines of being below 350 words.

Figure 1, page 5

Figure 1. Structure of the credit assignment task. **a)** In our task, participants assigned outcomes (overall feedback) to three causes: their own performance, another fictive player's performance, and their respective control over the feedback. These causes were hidden and latent, and there was only one piece of feedback on each trial. The feedback parametrically varied over trials. **b)** Each trial started with a rating screen on which participants rated their beliefs about their own control, their own performance, and the other player's performance. Then, they played one round of a simple game (action phase): In the example here, a ball rolled down a slide and they had to press a button when they believed it had passed the green target cross. A large black box hid the ball's path which made it impossible to know exactly when the ball passed the cross. This meant that participants had to rely on the feedback shown to them to infer their own performance, alongside the other player's performance and their own control. At the end of the trial, the overall feedback was shown which was the sum of the participants' own and the other player's performance, weighted by their respective control over the feedback. After participants pressed a button to continue, an ITI followed. **c)** Which ratings participants completed depended on the phase: In the beginning of each task block, participants entered the 'Self-Other phase' (left). This task phase allowed us to determine how participants assign credit to themselves and a new other player while knowing their control. Note that in the task, real faces were shown for the other players on each rating screen (here substituted with icons). In the Control-Other phase (right), participants inferred the new other player's performance and their new control over the feedback. In this phase, participants knew their own true performance level because of the preceding Self-only trials they had played between Self-Other and Control-Other phase. **d)** Participants completed four blocks, each of which was divided into three phases, signalled to participants (example block is shown here). Each block started with the Self-Other phase, after which followed Self-only trials, ending with the Control-Other phase.

Figure 2, page 7

Figure 2. Credit assignment behaviour. **a)** Hypothesized cognitive process and behaviour. Participants' beliefs (measured as their ratings) together form an estimate of the feedback they expect to receive (estimated feedback). At the time of the outcome, the discrepancy between their estimated feedback and the actual feedback results in a total prediction error (tPE). In the Self-Other phase, they use their knowledge about their control to split up the tPE into a PE assigned to themselves (sPE) and the other (oPE). **b)** Participants' estimated feedback followed the observed mean feedback throughout the task (example block 3 out of 4, mean±SD plotted as bold lines with shaded intervals). **c)** Participants' mean ratings (bold lines) and their SD (shaded interval) for the same example block as in **b**. Supplementary Figure 3 shows all blocks. **d)** Over time, participants improved the accuracy, both of the estimated feedback and the individual component ratings of self-performance, other performance, and control, from observing the feedback. Overall, the estimated feedback improved more from the start of a phase to the end than did the individual component ratings individually. Mean±s.e.m. accuracy, small dots show participants. Accuracy is defined as the inverse of the absolute rating errors, normalised by the highest error possible. **e)** Participants updated their beliefs about self and other based on the previous trial's tPE. As expected if participants assign credit appropriately, they here also take into account their control level over the feedback, and attribute more of the tPE to themselves (and less to the other) if they themselves have more control. Simulated data from a Bayesian observer model ("Active learner") show the same result pattern as participants. We used ANOVAs (with sample, fMRI versus online, as between-participant factor) and evaluated the p-value of the intercept term to assess the significance of participants' beta estimates (i.e. standardized regression coefficients) irrespective of sample. For completeness, effects of Control on self and 1-Control on other are described in the supplementary notes ('Control effect on self and other ratings'). Bars show mean beta estimates, error bars indicate s.e.m., grey dots show individual participants. n=31 MRI, n=36 online; *, p<0.05; **, p<0.01; ***, p<0.001.

Figure 3, page 10

Figure 3. Active disambiguation (AD). **a)** We hypothesized that participants made deliberate mistakes in the games to assign credit better. In the game shown here, they could press the button too late to see the difference this makes to the feedback. Whenever participants made such extreme errors, these were transformed into a zero self-performance and detected as AD trials (see also Methods and Supplementary Figure 4). **b)** We hypothesized that participants generated more AD trials under high uncertainty, and that AD trials helped them reduce uncertainty and assign credit better. **c)** Participants indeed generated AD when the feedback was ambiguous. Whenever the feedback was certain, AD dropped strikingly. We found a similar pattern in another online sample who did not receive a task hint to change their game behaviour (Supplementary Figure 5a). Bars and dots show the mean AD proportion. **d)** After the task, participants reported how many AD trials they generated per game ("In the study, how many times per game did you do badly on purpose?"). Their subjective reports of AD trials were significantly correlated with the objective frequency of AD trials identified in their behaviour. Again, these results replicate in another online sample (Supplementary Figure 5b). Dots show individual participants, line and shaded intervals are fitted regression curves with confidence intervals. **e)** Uncertainty is highest when a new phase starts (example block 3). Within each phase, uncertainty increases when the other's performance changes. **f-g)** Trial-wise uncertainty is reduced if the previous trial or the one before was an AD trial (f shows data from example participant; see Supplementary Figure 8c for full regression results of g). **h)** Crucially, participants' rating errors are reduced after AD, showing that they can use the additional information afforded by AD (similar to the Active learner). In g-h we did not find significant group differences between samples (all p>0.05). In g-h, bars show mean beta estimates with error bars indicating s.e.m., and grey dots for individual participants. Panel d: n=29 MRI and n=64 online with otherwise excluded participants (see Methods); Panels c, e, g, h: n=31 MRI, n=36 online; n.s., not significant; **, p<0.01; ***, p<0.001.

Figure 4, page 12

Figure 4. Inferring control. **a)** On normal trials, $Feedback = Self \times Control + Other \times (1 - Control)$. On AD trials, the participant plays so badly that $Self = 0$, so that the feedback only reflects the other player, weighted by their control. Therefore, control can optimally be inferred by observing the difference between normal and AD trials, accounting for self-performance ($Control = Feedback\ difference / Self$). **b)** Intuitively, low control can be inferred if there is barely any difference between feedback on a normal versus an AD trial. **c)** As expected and similar to the Active learner, after 'switch' trials (either AD following normal trials, or vice-versa), participants inferred their control from the feedback difference between AD and normal trials, taking into account their own overall game performance. As control analyses, we ran the same analyses on simulated data from two computational models that made wrong

assumptions about AD trials. Importantly, all agents (participants, Active, Passive and Ignorant Learners) **observed the same** feedback **participants received**. We found that indeed the passive and ignorant learner showed different learning patterns: the interaction effect, *Feedback difference*Inverse Self*, differentiated between, on the one hand, participants and the Active learner, and, on the other hand, the Passive and Ignorant learners. While participants and the Active learner had a significant positive interaction effect, the Passive learner had no significant effect, and the Ignorant learner had a significantly negative effect. **Dots with error bars show mean±CI of beta estimates. Grey dots are individual participants.** **d)** Both participants and the Active learner infer a low control when there is a lower difference between normal and AD feedback. If there is a large difference, they infer a high control – but this depends on the level of their own performance. If low self-performance is assumed and there is still a high feedback difference, then a higher control is inferred to account for a high feedback difference. The ignorant and passive learner show markedly different behaviour. **For easier interpretation, we relabelled 'inverse self' as 'self' and flipped its levels accordingly (e.g. 'low self' is 'high inverse self'). Lines with shaded areas show the mean estimates±CI.** n=31 MRI, n=36 online; n.s., 95% CI includes 0; ***, 99.9% CI excludes 0.

Figure 6, page 15

Figure 6. Representations of control, feedback difference and inverse self in the SMG at the time of outcome in the Control-Other phase. **a)** We used time-course analyses to identify the neural correlates of how participants infer their control from feedback difference and the inverse self. **b)** We indeed found that SMG tracks the inferred control estimate and the degree of control level updating at the outcome (ROI-GLM1). These activity patterns remained unchanged whether control and control updated were examined in the same or separate GLMs. **c)** We next tested whether SMG also represents the components necessary for inferring control. We compared two neural hypotheses for how neural activity can track the difference in feedback between trials. **Our first hypothesis (H1)** is that neural activity **tracks the** difference between normal and AD trials. This means that the neural activity does not depend on order: whether a normal trial followed an AD trial, or *vice-versa*. **d)** Alternatively (H2), neural activity might track the feedback difference in a simpler manner, by comparing the currently observed feedback to the feedback from the previous trial. We used a single GLM (ROI-GLM2) to test these two hypotheses. For this, we computed feedback difference as normal vs AD (according to H1) in all panels. However, we split trial transitions between AD to normal trial transitions and *vice versa*, because H1 and H2 make different predictions between those trial types. **e)** **At the outcome (ROI-GLM2), we found a significant effect of feedback difference in accordance with H1 rather than H2, suggesting that the SMG represents the feedback difference as normal vs AD feedback. The right panel shows participants' aggregated peak amplitudes using a LOO procedure. Note that feedback difference and control (panels e and b left) have low regressor correlations ($r < 0.3$, Supplementary Figure 9A), and are run on different trial selections (control on all trials, feedback difference on switch trials only).** **f)** In the same GLM (ROI-GLM2), we found that SMG also tracks participants' prior inverse self-estimate. n=31 MRI; **lines/bars show mean, error bars/shaded intervals show s.e.m.**; n.s.; not significant; ***, $p < 0.001$; effects are time-locked to the outcome phase; **panel a includes all trials from the Control-Other phase; panels e and f include only switch trials in the Control-Other phase.**

Reviewer #2 (Remarks to the Author):

Reviewer #3 (Remarks to the Author):

I am grateful to the authors for their work on this revision. In general, I think it is clear that the authors have done a significant amount of work to improve the manuscript, both in response to my queries and those raised in other reviews.

That said, I remain concerned about the unhedged claim that this is a study of 'human collaboration' - which I don't think the reviewers have appropriately addressed.

In their reply to me, the authors have explained very convincingly how this kind of 'social credit assignment' is likely to be an important part of enabling genuinely collaborative behaviours (e.g., I find the example of the student getting negative feedback and needing to diagnose the cause very compelling).

But I don't think it follows that just because a process is important for a behaviour in some settings, that studies of that process are therefore studies of that behaviour (e.g., spatial navigation is an important part of buying groceries in a store, but it doesn't follow that all studies of core navigation mechanisms are studies of buying groceries).

I agree with the authors that their social cover story makes it very likely that agents are engaged in some kind of social cognition and causal inference. But as they note, the cover story is not telling participants 'you will be working with each other to achieve a joint goal' (i.e. collaboration). The cover story is that of being a 'team captain' and estimating the performance of different players to form a team. As far as I can see, the only element of the task design that approaches collaboration is the fact that participants (believe they are) performing the same trial of the same game with another participant. But I don't think this kind of 'action in parallel' is strong enough to constitute collaboration (e.g., students sitting an exam in the same hall are solving the same problems at the same time, but that does not mean they are collaborating with each other).

These conceptual issues have a long history in both the cognitive science and philosophy of joint action - where a key defining feature of collaboration is usually taken to be the fact that agents hold a common collaborative intention, and act towards the same end. This is not the case anywhere in this task, where the task is explicitly to try and do causal inference over a feedback signal that can belong to you or somebody else.

As per my original review, I think the study here is very clever, the approach technically sophisticated and the statistical results are clear and persuasive. I also agree that this kind of social causal inference is an important element of genuine collaborative cognition when it does occur. But I don't see that the contributions of this paper *require* over-egging of the message to say this *is* a study of collaboration. In my view, it would suffice (and still be extremely interesting) to say that this is a study of 'social credit assignment', and this is an important process in some real life situations where the causal influence of different agents on an outcome is (like collaboration towards a joint goal). But I would worry about misleading readers, or adding 'noise' to the literature, if this study is literally titled as a study of 'human collaboration' - when it plainly isn't.

I think mitigating against this worry could be done with relatively simple changes to the paper (e.g., by not calling it a study of 'human collaboration' in the title, and by describing the process of social credit assignment or causal inference properly and how it is *relevant to* collaborative examples, rather than describing the task here as 'collaboration' or 'collaborative' per se). But such simple changes would be important for situating the paper in the literature appropriately, and giving readers a clear sense of what the paper contributes (and what it does not). I say all this, ultimately, as an enthusiast for what the paper does contribute to a range of important timely

issues - and I hope that authors will take this suggestion in that spirit.

We thank the reviewer for their positive feedback on our manuscript, and for stating that we have done “a significant amount of work to improve the manuscript”. We understand that they suggest we take more care with the specific choices of language that we make. As a result, we have changed the title of our manuscript, abstract, introduction and discussion to address their comments.

We agree that ‘social credit assignment’ is an accurate description of the process under study. We rather like that phrase ourselves and have wondered about using it as it makes a link with other work from our laboratories on credit assignment outside the social setting. However, we refrained from using this term in the title to avoid potential confusion with the Chinese Social Credit System (https://en.wikipedia.org/wiki/Social_Credit_System). Therefore, we exchanged ‘collaboration’ by the broader term ‘social interactions’ in our title. Below we detail the various changes we made to the manuscript in response to the reviewers’ comments.

Title: Controllability and cause in social interactions

Abstract, page 1

There has been considerable interest in how we ascertain whether an environment is controllable and the neural mechanisms mediating this process. An especially acute version of this problem occurs when multiple people work together. [...]

Introduction, page 2

It might initially seem difficult to imagine how such a principle might operate in social contexts but one possibility is that people effectively remove their own contribution by performing badly on purpose to determine what, if any, are the consequences.

Discussion, page 18

We investigated how in cooperative two-person games, individuals assign credit to either themselves or others, while also estimating their control over the outcomes (Figure 1).

Discussion, page 19

Instead, the focus is on comparison of the degree of action-dependent contingency exhibited by each actor – the participant themselves and the other person who they work together with in the task.

Discussion, page 20

The current results suggest that a similar process of inference and information seeking is inherent in social contexts.

[...]

Here, we studied how humans assign credit and infer their control in social settings. [...]

REVIEWERS' COMMENTS

Reviewer #1 (Remarks to the Author):

My residual comments have been addressed very well in the second revision of this impressive paper. The link between the results and the conclusions is now compelling. Congratulations to the authors for their important contribution to the literature: an intriguing question, an innovative procedure, advanced data analyses, and a carefully written manuscript.

Reviewer #2 (Remarks to the Author):

Reviewer #3 (Remarks to the Author):

I am grateful to the authors for their thoughtful work in their revisions and their responses. My previous concerns are addressed perfectly and I would be very happy to see this paper published in its present form.

We would like to thank all three reviewers for taking the time to review our manuscript. We appreciate their thoughtful comments and constructive feedback, and believe that our manuscript is much improved in its current form.